# Negotiated Reasoning: On Provably Addressing Relative Over-Generalization

**Junjie Sheng**[1], **Yantian Wang**[1], **Bo Jin**[1], **Hongyuan Zha**[2], **Jun Wang**[3],
**Wenhao Li**[1,✉], **Xiangfeng Wang**[4,5,✉]

1. School of Computer Science and Technology, Tongji University
2. School of Data Science, The Chinese University of Hong Kong, Shenzhen
3. School of Computer Science and Technology, East China Normal University (ECNU)
4. Key Laboratory of Mathematics and Engineering Applications (MoE), ECNU
5. Shenzhen Loop Area Institute (SLAI)
`{whli,bjin,wyt2024}@tongji.edu.cn, zhahy@cuhk.edu.cn`
`{jarvis@stu,jwang@cs,xfwang@cs}.ecnu.edu.cn`

## Abstract

Over-generalization is a thorny issue in cognitive science, where people may become overly cautious due to past experiences. Agents in multi-agent reinforcement learning (MARL) also have been found suffering *relative over-generalization* (RO) as people do and stuck to sub-optimal cooperation. Recent methods have shown that assigning *reasoning* ability to agent can mitigate RO algorithmically and empirically, but there has been a lack of theoretical understanding of RO, let alone designing provably RO-free methods. This paper first proves that RO can be avoided when the MARL method satisfies a consistent reasoning requirement under certain conditions. Then we introduce a novel reasoning framework, called negotiated reasoning, that first builds the connection between reasoning and RO with theoretical justifications. After that, we propose an instantiated algorithm, Stein variational negotiated reasoning (SVNR), which uses Stein variational gradient descent to derive a negotiation policy that provably avoids RO in MARL under maximum entropy policy iteration. The method is further parameterized with neural networks for amortized learning, making computation efficient. Numerical experiments on many RO-challenged environments demonstrate the superiority and efficiency of SVNR compared to state-of-the-art methods in addressing RO.

## 1 Introduction

Multi-agent reinforcement learning (MARL) has been successfully applied in multiplayer games (Rashid et al., 2019; Kurach et al., 2020; Li et al., 2026), robotics (Ding et al., 2020), and traffic control (Calvo & Dusparic, 2018). This paper addresses relative over-generalization (RO), a critical pathology in fully cooperative MARL settings where agents pursue team-optimal outcomes. RO is analogous to over-generalization in cognitive science (Rand et al., 2014; Laufer et al., 2016), where limited experiences lead to broad, often inaccurate generalizations—as in the "once bitten, twice shy" idiom, where a person bitten by a snake develops fear of rope-like objects. This cognitive phenomenon has been documented across language acquisition (Gershkoff-Stowe et al., 2006), social learning (Rand et al., 2014), and decision-making (Laufer et al., 2016).

In MARL, relative over-generalization (RO) poses a significant challenge to optimal cooperation (Palmer, 2020), as agents overfit their policies to others' exploration behaviors. This is evident in *Particle Gather*, where particles aiming to reach a landmark synchronously become risk-averse after experiencing penalties from uncoordinated visits, causing methods like (Lowe et al., 2017; Wei et al., 2018; Wen et al., 2019) to converge to suboptimal strategies (see §6). Two major approaches address RO: credit assignment methods (evolving from early lenient learning (Panait et al., 2006b; Wei & Luke, 2016; Palmer et al., 2017) to sophisticated value decomposition (Li et al., 2021; Peng et al., 2021; Zhang et al., 2021; Gupta et al., 2021; Siu et al., 2021; Huang et al., 2022; Kang et al., 2022; Yang et al., 2022; Shi et al., 2024; Hu & Ying, 2024) and shaped values (Wan et al., 2022; Shi & Peng, 2022; Zhao et al., 2023; Li et al., 2024a; Toquebiau et al., 2024)), and reasoning-endowed

methods (Wen et al., 2019; Ma et al., 2022; Tian et al., 2019; Wei et al., 2018) that adopt an ego-agent perspective, equipping agents with capabilities to model others' behavior—like recursive reasoning in (Wen et al., 2019) inspired by human cognition (Von Der Osten et al., 2017). Despite empirical successes, both approaches lack solid theoretical foundations. Some works prove algorithm convergence (Peng et al., 2021; Li et al., 2024a; Hu & Ying, 2024) or optimality in matrix games (Wan et al., 2022), but none formally define RO. This raises two key questions: (1) **can RO be provably avoided?** and if it can, (2) **how to design a method that provably addresses RO?**

This paper answers the first question with theoretical justifications and introduces new concepts to analyze Relative Over-generalization (RO) in Multi-Agent Reinforcement Learning (MARL). The current RO is defined on empirical converged joint policy, which makes it difficult to analyze MARL methods before training. To address this issue, we introduce *Perceived Relative Over-generalization* (PRO) and *Executed Relative Over-generalization* (ERO), which define RO for each joint policy update and policy execution, respectively. The RO is guaranteed to be addressed when ERO is avoided at convergence. With the basis, we prove that RO can be provably avoided when the MARL method satisfies a *consistent reasoning* condition at convergence. This condition requires each agent to model the behaviors of others consistently with their updated/executed behaviors.

For the second question, we propose a novel negotiated reasoning framework that satisfies the consistent reasoning condition, inspired by human negotiation processes (Kim, 1996; Carnevale & Lawler, 1986) and graphical model message-passing inference (Pearl, 1988). Our framework enables explicit reasoning through negotiation policies during training and decision-making based on negotiated agreements. We prove that agents achieve consistent reasoning when they reach action selection agreements through appropriate negotiation, and introduce Stein Variational Negotiated Reasoning (SVNR), which derives negotiation policies via Stein variational gradient descent and employs a strict nested negotiation structure. With maximum entropy policy iteration, SVNR provably achieves consistent reasoning and optimal cooperation at convergence under mild conditions. We further parameterize SVNR with neural networks and implement amortized learning to address computational complexity, distilling negotiation dynamics into network updates and approximating multiple negotiation rounds with single forward passes. Experiments in challenging differential games, particle world and multi-agent MuJoCo environments demonstrate SVNR's superiority in addressing RO compared to state-of-the-art reasoning methods.

The main contributions are threefold: 1) We confirm the existence of provably addressing relative over-generalization (RO) methods; 2) We propose a novel framework called negotiated reasoning (NR) and specify the Stein variational NR method, which is the first MARL method that can provably address relative over-generalization (RO); 3) We propose a practical implementation of SVNR that demonstrates superior performance in achieving global optimal cooperation in RO-challenged tasks.

**Remark 1.** Our work adopts an autoregressive conditional policy factorization. This approach has been used in several multi-agent policy factorization works (Ding et al., 2022; Wang et al., 2023a; Fu et al., 2022; Ye et al., 2022; Li et al., 2024b) and decision-making foundation models (Wen et al., 2022), supporting SVNR's effectiveness. Unlike these works, which address general multi-agent cooperative tasks, we focus specifically on the RO problem. The autoregressive policy factorization (the strictly nested negotiation set in SVNR) is just one optimal form. The optimal negotiation set covers a broader range of factorizations. Moreover, in contrast to some previous negative results on autoregressive policy factorization—such as the inability to leverage other agents' optimal actions (Ding et al., 2022), sensitivity to the autoregressive order (Li et al., 2024b), and the requirement for centralized execution (Fu et al., 2022)—we provide a theoretical proof of the optimality of any strictly nested negotiation set. We also achieve decentralized execution through an amortized negotiation mechanism.

**Remark 2.** In addition to autoregressive policy factorization, another class of methods in MARL sequentially updates agents' local, independent policies (Wang et al., 2023b; Kuba et al., 2022; Feng et al., 2023; Zhang et al., 2024). These methods are closely related to SVNR, though they mainly address non-stationarity rather than RO. Unlike autoregressive factorization, where agents exchange current policies in the negotiation process, sequential update methods convey the impact of one agent's policy update on the environment and subsequent agents.

## 2 RELATIVE OVER-GENERALIZATION

This section defines RO under CTDE MARL contexts. Specifically, we propose two concepts, perceived RO (PRO) and executed RO (ERO), that distinguish different RO in CTDE. Then, we bridge the two concepts to RO and prove that RO can be avoided when PRO and ERO are addressed under mild conditions. Prior to introducing formal definitions, we first establish the problem formulation and associated mathematical notation.

**Cooperative Stochastic Game.** A Cooperative Stochastic Game (CSG) is commonly used to model cooperation in multi-agent systems (Petrosjan, 2006). It is defined by a tuple $(\mathcal{S}, \{\mathcal{U}_i\}_{i=1}^N, P, \mathcal{R}, \gamma)$, where $N$ is the number of agents; $\mathcal{S}$ is the state space; $\mathcal{U}_i$ represents the action space for agent $i$ with $\mathcal{U} = \times_i \mathcal{U}_i$ representing the joint action space; $P(\boldsymbol{s}' \mid \boldsymbol{s}, \boldsymbol{u})$ representing the probability that environment transit to $\boldsymbol{s}'$ when taking joint action $\boldsymbol{u}$ at state $\boldsymbol{s}$; $\mathcal{R} : \mathcal{S} \times \mathcal{U} \to \mathbb{R}$ is the team reward[1] function; $\gamma \in [0,1]$ is the discount factor. The goal for the CSG is to find policies $\{\pi_i\}_{i=1}^N$ that make accumulative reward the highest. The $\pi_i : \mathcal{S} \to \mathcal{U}_i$ maps the state to agent $i$'s action and the objective of CSG can be formulated as $\max_{\pi_1, \ldots, \pi_N} \mathcal{E} \left[ \sum_{t=1}^{\infty} \gamma^t \mathcal{R}(\boldsymbol{s}_t, \boldsymbol{u}_t) \right]$, where $\boldsymbol{u}_t$ is sampled from the policies as $\boldsymbol{u}_t^i \sim \pi_i(\cdot \mid \boldsymbol{s}_t)$.

**Multi-Agent Reinforcement Learning.** MARL methods are popular for solving the cooperative stochastic game. This paper considers the mainstream of MARL schemes: centralized training decentralized execution (CTDE). Each agent $i$ holds an execution policy $\bar{\pi}_i(u^i \mid s)$ to make execution in a decentralization manner and a *perceived* joint policy $\hat{\pi}_i(\boldsymbol{u} \mid s)$ to do centralized training. The *perceived* joint policy can be factorized as $\hat{\pi}_i = \pi_i \rho_i$, where $\pi_i$ is the individual policy and $\rho_i$ is the perceived opponent policy. Following MaxEnt MARL (Tian et al., 2019; Wen et al., 2019; Wei et al., 2018), each agent $i$ optimizes its policy by minimizing the KL-divergence between perceived joint policy and the induced optimal joint policy: $\min_{\pi_i} D_{KL} \left( \hat{\pi}_i \| \pi_\alpha^* \right)$ where $\alpha$ is the factor that balances the reward and entropy. The $\pi_\alpha^*$ is induced by the Boltzmann optimal policy:

$$\pi_\alpha^*(\boldsymbol{u} \mid \boldsymbol{s}) := \exp \left( \tfrac{1}{\alpha} \left( Q_{\text{soft}}^* (\boldsymbol{s}_t, \boldsymbol{u}_t) - V_{\text{soft}}^* (\boldsymbol{s}_t) \right) \right), \tag{1}$$

where $Q_{\text{soft}}^*$, $V_{\text{soft}}^*$ denote optimal, soft state-action and state value function, respectively (Haarnoja et al., 2017). After that, each agent $i$ obtains decentralized execution policy as $\bar{\pi}_i(u^i \mid \boldsymbol{s}) := \int \hat{\pi}_i d\boldsymbol{u}^{-i}$ and the utility of the decentralized execution is: $U^{\bar{\pi}} := \sum_t \mathbb{E}_{(\boldsymbol{s}_t, \boldsymbol{u}_t) \sim \beta_{\bar{\pi}}} \mathcal{R}(\boldsymbol{s}_t, \boldsymbol{u}_t)$, where $\bar{\pi} := \prod_i^N \bar{\pi}_i$ is the executed joint policy, and $\beta_{\bar{\pi}}$ is the state-action marginals of the trajectory distribution induced by $\bar{\pi}$.

Relative over-generalization is a critical game pathology in MARL. It occurs when agents prefer a sub-optimal Nash Equilibrium over an optimal Nash Equilibrium because each agent's individual policy in the sub-optimal equilibrium has a higher utility when paired with arbitrary policies from opponents (Wei et al., 2018). This definition assumes MARL methods directly select the joint policy from multiple Nash Equilibriums while these methods make a comparison between the current joint policy and updated joint policy for each updating. Thus we extend RO by considering each update. Besides that, the current CTDE scheme in MARL motivates us to decompose RO to perceived relative over-generalization (PRO) in the training phase and executed relative over-generalization (ERO) in the execution phase. First, we define the ERO, which extends RO at each execution step and identifies whether the optimal cooperation is disturbed due to not knowing the behaviors of opponents.

**Definition 2.1** (Executed Relative Over-generalization). Agent $i$ suffers executed relative over-generalization if and only if the utility of executed joint policy can be improved by letting agents know others' actions: $\max_{\pi_i} \{ U^{\pi_i(u^i \mid s, \boldsymbol{u}^{-i})} \prod_{j \neq i} \bar{\pi}_j^*(u^j \mid s) \} > U^{\prod_j \bar{\pi}_j^*(u^j \mid s)}$ where $\pi_i^* = \arg\min_{\pi_i} D_{KL}(\pi_i \rho_i \| \pi_\alpha^*)$ is the $i$'s optimal policy with $\rho_i$ and $\bar{\pi}_i^* = \int \pi_i^* \rho_i d\boldsymbol{u}^{-i}$ is the executed policy for each agent $i$.

It is straightforward that agents do not suffer from RO if all agents are free from ERO at convergence. Besides that, agents also suffer from RO during their training phase, and we further propose the definition of *Perceived Relative Over-generalization*.

**Definition 2.2** (Perceived Relative Over-generalization). Agents suffer perceived relative over-generalization iff. there exists an agent $i$ whose optimal perceived joint policy can be closer to the optimal joint policy when knowing the optimal opponent policy: $\min_{\pi_i} D_{KL}(\pi_i \rho_i \| \pi_\alpha^*) >$

---

[1]The utility, reward and payoff are not distinguished.

$\min_{\pi_i} D_{KL}(\pi_i \pi_\alpha^*(\boldsymbol{u}^{-i}) \| \pi_\alpha^*)$ where $\pi_\alpha^*$ is the optimal joint policy with entropy factor $\alpha$, and $\pi_\alpha^*(\boldsymbol{u}^{-i}) := \int_{u^i} \pi_\alpha^* du^i$ is the optimal opponent policy.

The perceived optimal joint policy for each agent is equal to the optimal joint policy for the case that the agents are free from PRO. When each agent $i$ reasons others' behaviors consistent with their optimal policy $\rho_i = \pi_\alpha^*(\boldsymbol{u}^{-i})$ in the training phase, others' exploration will not impact the agent's policy updating and the PRO is avoided. If PRO is avoided and $\alpha \to 0$, all agents execute deterministically, the agent's execution will not be impacted by others' exploration stochastic in the execution phase, and ERO is avoided. These conditions are denoted as consistent reasoning, and we define them below.

**Definition 2.3** (Consistent Reasoning). Agents meet consistent reasoning if and only if all agents reason others' behaviors consistent with their optimal policy $\rho_i = \pi_\alpha^*(\boldsymbol{u}^{-i})$ in the training phase and reason others' behaviors consistent with their executed actions during execution.

When the requirement is met at convergence, agents are free from ERO, and they do not suffer from RO. Existing reasoning methods are unable to reach consistent reasoning. We take Figure 1 as an example to better illustrate how these methods suffer from PRO and ERO respectively. It is a single-stage, cooperative game and contains two agents "A" and "B". The action space of each agent is $\{0, 1\}$. In Figure 1 (Left), MADDPG (Lowe et al., 2017) usually suffers from PRO due to agents reason others through their historical behaviors. For agent A, if $\rho_A(0) = \rho_A(1) = 0.5$, it will obtain $\hat{\pi}_A'(1, 0) = 1$ which is sub-optimal. MASQL (Wei et al., 2018) usually suffers from ERO in Figure 1 (Right). If $\hat{\pi}_A'(1, 0) = \hat{\pi}_A'(0, 1) = 0.5$ and

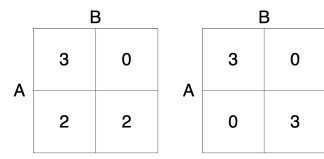

Figure 1: The PRO and ERO payoff functions examples.

$\hat{\pi}_B'(1, 0) = \hat{\pi}_B'(0, 1) = 0.5$, then PRO is avoided. However when making decentralized execution based on $\hat{\pi}'$ for each agent, $\bar{\pi}(1, 1) = \bar{\pi}(0, 0) = \bar{\pi}(1, 0) = \bar{\pi}(0, 1) = 0.25$, which are sub-optimal and suffer from ERO.

**Intuitive Interpretations of Theoretical Concepts.** To connect the formal definitions with practice, we summarize PRO and ERO via variational inference and distributional factorization; a full version is provided in Appendix E.8. **(1) PRO as Variational Bias.** In MaxEnt MARL, agent $i$ minimizes $D_{\text{KL}}(\pi_i \rho_i \| \pi_\alpha^\star)$, where $\rho_i$ is the *perceived* opponent policy. If $\rho_i$ deviates from the optimal conditional $\pi_{-i}^\star$, the objective becomes biased, so optimization steers toward a safe local optimum rather than the cooperative global one. Thus, PRO is a training-time estimation error caused by imperfect beliefs about others. This mirrors misspecified variational families: the learner optimizes a KL against a target distribution filtered through a wrong conditional, so the fixed point is self-consistent with $\rho_i$ but not globally optimal. **(2) ERO as Factorization Loss.** Even if training converges to an optimal joint policy $\hat{\pi}$, decentralized execution requires a product form $\bar{\pi}(u) = \prod_i \pi_i(u_i)$. When $\hat{\pi}$ is correlated or multimodal, projecting it onto independent marginals incurs factorization loss, yielding miscoordination and lower return. Geometrically, $\bar{\pi}$ is the closest point in the independence manifold, so correlation structure in $\hat{\pi}$ is flattened, removing the signals needed to pick a common mode. **(3) Consistent Reasoning as Closing the Loop.** Consistent reasoning enforces both training consistency ($\rho_i \to \pi_{-i}^\star$) and execution consistency (negotiation collapses to a single agreed mode so $\hat{\pi}(u) \approx \prod_i \pi_i(u_i)$ as $\alpha \to 0$), aligning planned and executed actions. One distribution is optimized, shared, and executed, turning centralized optimality into decentralized realizability.

## 3 NEGOTIATED REASONING FRAMEWORK

Inspired by the critical role of negotiation for consistent reasoning in social cooperation, we introduce negotiation in the reasoning process to avoid PRO and ERO with theoretical justifications and propose a novel reasoning framework, NR. In NR, agents take $M$ particles $\{\boldsymbol{u}^{\ell,0}\}_{\ell=1}^M$ to represent the initial perceived joint policy distribution $p(\boldsymbol{u}^0) := \frac{1}{M} \sum_{\ell=1}^M \delta_{\boldsymbol{u}^{\ell,0}}(\boldsymbol{u})$ for a state $s$. Moreover, each agent $i$ holds a negotiation (*i.e.*, perturb) policy $f_i(u_i \mid \boldsymbol{u}_{C_i}, s)$ that updates its action when knowing the $C_i$'s action selection. Here $C_i \subseteq 1, \ldots, N$ is the negotiated set for agent $i$, which determines whom to negotiate, $f_i := \{f_i^1, \ldots, f_i^K\}$ where $f_i^k$ is the negotiation policy of agent $i$ in iteration $k$, and $K$ is the number of negotiation rounds which is often large enough. Then every agent $i$ makes negotiated reasoning as $u_i^{\ell,k} = f_i^k(u_i \mid s, \boldsymbol{u}_{C_i}^{\ell,k-1}), \forall i \leq N, \ell \leq M, k \leq K$. Such a

negotiation process can be interpreted as agents starting from initial action beliefs and negotiating with each other based on their negotiation policies. When $f_i^k$ converges to an identity map for each agent, the perceived joint policy converges to a steady perceived joint policy (*i.e.*, agreement): $\lim_{k \to K} p(\boldsymbol{u}^k \mid \boldsymbol{s}) := \frac{1}{M} \sum_{\ell=1}^M \delta_{\boldsymbol{u}^{\ell,k}}(\boldsymbol{u}) \to \pi^s(\boldsymbol{u} \mid \boldsymbol{s}), \forall \boldsymbol{u} \in \mathcal{U}$. Negotiated reasoning avoids PRO when it meets certain conditions.

It is crucial to distinguish this framework from communication-based MARL methods that exchange messages to resolve partial observability (i.e., approximating global state). In contrast, Negotiated Reasoning operates on the *probability measure space*. The "negotiation" is a functional gradient descent process in a Reproducing Kernel Hilbert Space (RKHS) that aligns the joint policy distribution with the global value landscape. This addresses equilibrium selection rather than state estimation.

**Theorem 3.1** (PRO-free Negotiated Reasoning). *For any environment state $s$ where the optimal joint policy is defined as $\pi_\alpha^*$, consider each agent $i$ takes a negotiated reasoning defined on a compact action space $\mathcal{U}_i$, they are PRO-free with $K$ steps negotiated reasoning if $\lim_{k \to K} p(\boldsymbol{u}^k \mid \boldsymbol{s}) = \pi^*(\boldsymbol{u}^k \mid \boldsymbol{s}), \quad \forall \boldsymbol{u}^k \in \mathcal{U}$.*

This motivates us to learn negotiation policy $f_i$ satisfying the following conditions:

$$\lim_{k \to K} f_i^k(u_i \mid \boldsymbol{s}, \boldsymbol{u}_{C_i}^{\ell,k-1}) = u_i^{\ell,k-1}, \ \lim_{k \to K} p(\boldsymbol{u}^k \mid \boldsymbol{s}) = \pi^*(\boldsymbol{u}^k \mid \boldsymbol{s}), \ \forall \, i \le N, \ell \le M, \boldsymbol{u}^k \in \mathcal{U}, . \tag{2}$$

The first condition requires the negotiation policies to converge to the identity map, and the second one requires the perceived joint policy to be identical to the optimal joint policy when the negotiation policy converges. We will specify the negotiated policy learning in the following two sections.

As for ERO-free in decentralized execution, we prove that setting $\bar{\pi}_i = u_i^{0,K}$ with annealing $\alpha \to 0$ ensures ERO-free in decentralized execution (see proof in Appendix E.2).

**Theorem 3.2** (ERO-free Negotiated Reasoning). *For any environment state $s$, when agents are PRO-free with $K$ reasoning steps, they achieve ERO-free with annealing $\alpha \to 0$ if each agent $i$ sample action $\bar{\pi}_i = u_i^{0,K}$.*

When all the conditions are met, it is straightforward that consistent reasoning is obtained. Up to this point, we have established a theoretical connection between reasoning and RO. The next step is to design a negotiation policy that satisfies the condition in equation 2 and integrate this negotiated reasoning into existing multi-agent reinforcement learning.

## 4 STEIN VARIATIONAL NEGOTIATED REASONING

After building the theoretical relationship between reasoning and RO, this section proposes Stein variational NR, SVNR, under the NR framework, which is the first MARL method that provably addresses RO. We first derive the negotiation policy based on Stein variational gradient descent which obtains PRO-free negotiated reasoning. Then we devise the policy iteration method of SVNR and prove that it addresses PRO and ERO. Finally, we propose a practical implementation by parameterizing SVNR with neural networks and amortizing the learning procedure.

### 4.1 LEARNING THE NEGOTIATION POLICY

To learn the negotiation policy that converges to an identity map and lets perceived joint policy converges to the optimal joint policy as in equation 2, we start by building the relationship between negotiation policy and perceived joint policy. Decomposing KL divergence from the perceived joint policy, we have $D_{KL}\big(p(\boldsymbol{u} \mid \boldsymbol{s}) \| \pi^*(\boldsymbol{u} \mid \boldsymbol{s})\big) = D_{KL}\big(p(\boldsymbol{u}_{-i} \mid \boldsymbol{s}) \| \pi^*(\boldsymbol{u}_{-i}\boldsymbol{s})\big) + D_{KL}\big(p(u_i \mid \boldsymbol{s}, \boldsymbol{u}_{-i})p(\boldsymbol{u}_{-i}) \| \pi^*(u_i \mid \boldsymbol{s}, \boldsymbol{u}_{-i})p(\boldsymbol{u}_{-i})\big)$. It states that the KL divergence between perceived and optimal joint policy can be minimized by

$$\min_{p(u_i \mid \boldsymbol{s}, \boldsymbol{u}_{-i})} D_{KL}\big(p(u_i \mid \boldsymbol{s}, \boldsymbol{u}_{-i})p(\boldsymbol{u}_{-i}) \| \pi^*(u_i \mid \boldsymbol{s}, \boldsymbol{u}_{-i})p(\boldsymbol{u}_{-i})\big), \tag{3}$$

when fixing other agents' action selections (update only one agent's action). This motivates us to design a negotiation policy that minimizes the equation 3. One of the most popular ways to solve

equation 3 is (MP)SVGD (see Appendix A) which can naturally fit the updating of the single agent's action while fixing others'. Specifically, it adopts the following scheme, *i.e.*,

$$f_i(u_i \mid \boldsymbol{u}_{C_i}^\ell, \boldsymbol{s}) : u_i^\ell + \epsilon \phi_i(\boldsymbol{u}_{C_i})^\ell, \ \forall \ i \le N, \ell \le M, \tag{4}$$

to update the joint policy distribution. The $\epsilon$ is the learning rate, and $\phi_i$ is the transformation direction in vector-valued reproducing kernel Hilbert space. Then the optimal $\phi$ has a closed form solution for equation 3 when restricting $\|\phi_i\|_{\mathcal{H}_i} \le 1$ and $\epsilon \to 0$:

$$\phi_i^*(\boldsymbol{u}_{C_i}) = \mathbb{E}_{\boldsymbol{y} \sim p}[k_i(\boldsymbol{u}_{C_i}, \boldsymbol{y}_{C_i}) \nabla_{y_i} \log \pi^*(y_i \mid \boldsymbol{y}_{C_i}) + \nabla_{y_i} k_i(\boldsymbol{u}_{C_i}, \boldsymbol{y}_{C_i \setminus \{i\}})]. \tag{5}$$

The $\phi^*$ provides the steepest direction to optimize the KL divergence. This iterative update process is mathematically grounded in the transport of probability measures via Stein variational gradient flow. We provide a detailed theoretical interpretation of this negotiation process and its visualization in Appendix L. The Appendix D.1 shows the details of the derivation.

To further ensure the identity map convergence and let the converged perceived joint policy identical to the optimal joint policy, the design of $\{C_i\}_{i=1}^N$ plays a key role as seen in graphical inference problems (Pearl, 1988; Zhuo et al., 2018). Benefiting from the centralized training, we can design $C_i$ without considering communication limitations. When $\{C_i\}_{i=1}^N$ is strictly nested (*e.g.*, $C_i = \{1, \ldots, i\}$ for all $i$), negotiated reasoning equation 4 with equation 5 converges and the agreement is identical to optimal joint policy (*i.e.*, satisfies PRO-free conditions equation 2) as proved in Appendix E.4. We denote the negotiated reasoning with (MP)SVGD and strict nested negotiation set as Stein variational negotiated reasoning. While strict nesting guarantees exact representability, relaxing this constraint leads to a bounded approximation error characterized by Information Projection, as detailed in Appendix E.9.

## 4.2 Maximum Entropy Policy Iteration

In the previous section, we assumed that the optimal joint policy is known in advance. However, agents have to iteratively learn $Q$, and $V$ functions to estimate the optimal joint policy and update their sampling policy accordingly in practice. This section establishes SVNR on the maximum entropy policy iteration and shows the convergence to the optimal joint policy theoretically. Concretely, we first define the soft bellman operator as

$$\Gamma_{\hat{\pi}} Q(\boldsymbol{s}_t, \boldsymbol{u}_t) := r_t + \gamma \mathbb{E}_{\boldsymbol{s}_{t+1}}[V(\boldsymbol{s}_{t+1})], \tag{6}$$

where $V(\boldsymbol{s}_t) = \mathbb{E}_{\hat{\pi}}[Q(\boldsymbol{s}_t, \boldsymbol{u}_t) - \alpha \log \hat{\pi}(\boldsymbol{u}_t \mid \boldsymbol{s}_t)]$. Each round of iteration usually consists of joint policy evaluation and joint policy improvement, where joint policy evaluation aims to evaluate the policy performance with $Q$ and joint policy improvement updates each agent's policy accordingly. As for the joint policy evaluation, we obtain the following theorem.

**Lemma 4.1** (Joint Policy Evaluation). *For a mapping $Q^0 : \mathcal{S} \times \mathcal{U} \to \mathbb{R}$ with $|\mathcal{U}| < \infty$, define the $Q^{k+1} = \Gamma_{\hat{\pi}} Q^k$ where the $\Gamma$ is the soft bellman operator, then it converges to the joint soft $Q$-function of $\hat{\pi}$ as $k \to \infty$.*

Following equation 1 and equation 5, the $\hat{\pi}$ is updated as:

$$\hat{\pi}(\boldsymbol{u}) = \lim_{k \to K} \frac{1}{M} \sum_{\ell=1}^M \delta_{u^{\ell,k}}(\boldsymbol{u}), u_i^{\ell,k} = u^{\ell,k-1} + \epsilon \phi^*(\boldsymbol{u}_{C_i}^{\ell,k}, u_i^{\ell,k-1}), \ \forall i \le N, \ell \le M, k \le K,$$
$$\tilde{\pi} = \exp \frac{1}{\alpha} \left( Q(u_i, \boldsymbol{u}_{C_i}, \boldsymbol{s}) - V(\boldsymbol{u}_{C_i}, \boldsymbol{s}) \right), \tag{7}$$

where $Q(u_i, \boldsymbol{u}_{C_i}, \boldsymbol{s}) = \mathbb{E}_{\bar{\boldsymbol{u}} \sim \hat{\pi}(\boldsymbol{s}), \bar{\boldsymbol{u}}_{C_i} = \boldsymbol{u}_{C_i}, \bar{\boldsymbol{u}}_i = \boldsymbol{u}_i} Q(\bar{\boldsymbol{u}}, \boldsymbol{s}), V_i(\boldsymbol{u}_{C_i}, \boldsymbol{s}) = \mathbb{E}_{\bar{\boldsymbol{u}}' \sim \hat{\pi}(\boldsymbol{s}), \bar{\boldsymbol{u}}_{C_i} = \boldsymbol{u}_{C_i}} Q(\bar{\boldsymbol{u}}, \boldsymbol{s})$, and $\phi_i^*$ take $\tilde{\pi}$ instead of $\pi^*$ to construct the SVGD direction. Then we can obtain the following joint policy improvement lemma:

**Lemma 4.2** (Policy Improvement). *When the negotiation policies are strictly nested, given the current perceived joint policy as $\hat{\pi}$, update it based on the equation 7 and obtain the new perceived joint policy $\hat{\pi}'$. The $Q^{\hat{\pi}'}(\boldsymbol{s}_t, \boldsymbol{u}_t) \ge Q^{\hat{\pi}}(\boldsymbol{s}_t, \boldsymbol{u}_t)$ with $|\mathcal{U}| < \infty$.*

Following Lemma 4.1 and Lemma 4.2, we can establish the following SVNR policy iteration theorem and our proposed coordinated policy iteration method accordingly.

**Theorem 4.3** (SVNR Policy Iteration). *When the individual policies satisfy the strict nested require-
ment, considering repeated apply the joint policy evaluation and joint policy improvement on the
perceived joint policy $\hat{\pi}$, then $\hat{\pi}$ will converge to $\pi^*$ that makes $Q^{\pi^*}(s_t, u_t) \geq Q^{\hat{\pi}}(s_t, u_t)$, $\forall \hat{\pi} \in \Pi$, $(s_t, u_t) \in \mathcal{S} \times \mathcal{U}$, $|\mathcal{U}| < \infty$.*

While the analysis assumes discrete action spaces to utilize standard fixed-point theorems, the
theoretical results extend to continuous domains through measure-theoretic unification. Furthermore,
the core Negotiated Reasoning mechanism (via SVGD) is natively designed for continuous spaces.
We provide the detailed continuous formulation and justification in Appendix K. Based on the
Theorem 4.3, we can obtain the convergence of SVNR policy iteration to the optimal joint policy.
Further, taking Theorem 3.2, we can obtain ERO-free executed joint policy $\bar{\pi}$ by annealing $\alpha$ to a
small enough number.

However, empirically, the SVNR policy iteration assumes knowing the word model and encounters
high computation and storage complexity due to 1) *inefficient policy representation*: SVNR policy
iteration represents the joint policy with particles that scale poorly on state-action space; 2) *intractable
optimization*: During learning, the soft bellman operator takes expectations on both the state and
joint policy distribution, which is intractable in realistic settings. To this end, we propose a practical
implementation for SVNR.

## 5 A PRACTICAL IMPLEMENTATION OF SVNR

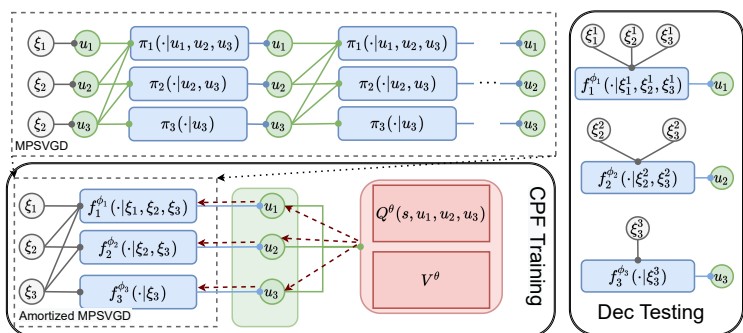

Figure 2: The practical SVNR in the 3-agents system. SVNR adopts nested negotiated reasoning and
adopts amortized MPSVGD to output the actions. The amortized MPSVGD distills the multi rounds
negotiated reasoning dynamic by well-established neural networks. The "Dec Testing" (rightmost
part) illustrates how the proposed SVNR executes in a decentralized manner.

To address the inefficient policy representation and intractable optimization issues, this section
adopts neural networks to parameterize the policies and learn them with the proposed amortized
optimization. To gain efficient action sampling, we propose Amortized MPSVGD. It aims to adopt
neural networks to perform **variational distillation** of the negotiation equilibrium. Rather than
cloning the iterative negotiation trajectory, the network aims to approximate the steady negotiation
result (the fixed point) in $\hat{\pi}(u)$ directly via neural network inference. Formally, each agent holds a
stochastic mapping function $u^i = f^i_{\psi^i}(\cdot|\xi^i, \xi^{C_i}, s)$ that maps initial noises (i.e., gaussian noises) to
its action distribution. The $\xi^i$ is the noise drawn by agent $i$. We denote the induced joint distribution
as $p_\psi(u|s, \xi) := \prod_{i=1}^N f^i_{\psi^i}(u^i|\xi^i, \xi^{C_i}, s)$. The goal of the proposed amortized MPSVGD method is
to find $\psi^*$ that satisfies:

$$\arg\min_{\hat{\psi}} \text{KL}\left(p^{\hat{\psi}}(\cdot \mid s, \xi) \| \hat{\pi}(u)\right). \tag{8}$$

A straightforward way to learn $\psi$ is to iterate the equation 7 procedure until convergence and to
establish the neural networks $\{\psi_1, \ldots, \psi_N\}$ which can fit the agreement. However, the equation 7
requires many rounds of updating, and this motivates us to introduce an incremental update scheme.
For each agent $i$, its policy parameter $\psi_i$ is updated by moving along its SVGD's gradient in order to
approach the target joint policy. Sampling joint actions $u^1, \ldots, u^M$ from $p$ and assuming we can

perturb agent $i$'s action $u_i^j = f^{\psi_i}(\xi_i^j; \xi_{C_i}^j, \boldsymbol{s})$ in appropriate direction $\Delta f^{\psi_i}(\xi_i^j; \xi_{C_i}^j, \boldsymbol{s})$, the induced KL divergence in equation 8 can further be reduced. MPSVGD provides the most greedy direction as

$$\Delta f_i^{\boldsymbol{\psi}}(\cdot; \boldsymbol{s}_t) = \mathbb{E}_{\boldsymbol{u} \sim p^{\psi}} \left[ \kappa_i\big(\boldsymbol{u}_{C_i}, p_{C_i}^{\boldsymbol{\psi}}(\cdot; s_t)\big) \nabla_{u_i'} Q^\theta(\boldsymbol{s}_t, \boldsymbol{u}')\big|_{\boldsymbol{u}'=\boldsymbol{u}} + \alpha_i \nabla_{\boldsymbol{u}_i'} \kappa_i\big(\boldsymbol{u}_{C_i}', p_{C_i}^{\boldsymbol{\psi}}(\cdot; s_t)\big)\big|_{\boldsymbol{u}'=\boldsymbol{u}} \right],$$
$$(9)$$

where $\alpha_i$ is the agent $i$'s temperature term, $\theta$ is the neural network paramter of central critic, and $\kappa_i$ is the agent $i$'s kernel function as in MPSVGD. We can then set $\frac{\partial J_p(\boldsymbol{\phi}; \boldsymbol{s}_t)}{\partial u_i} \propto \Delta f_i^{\boldsymbol{\phi}}$ (Feng et al., 2017).

Further, the gradient in MPSVGD can be backpropagated to the mapping network $\phi_i$, i.e.,

$$\frac{\partial J_p(\psi; s_t)}{\partial \psi_i} \propto \mathbb{E}_\xi \left[ \Delta f_i^\psi(\xi; s_t) \frac{\partial f_i^\psi(\xi; s_t)}{\partial \psi_i} \right]. \tag{10}$$

Therefore, any gradient-based methods can optimize the parameters $\psi_i$. The detailed derivations of equation 9 and equation 10 are shown in Appendix D. With this Amortized MPSVGD mapping function, neural network inference can directly sample joint actions. Crucially, by optimizing $\psi$ via this incremental scheme, the network $f_\psi$ learns to distill the multi-step negotiation dynamics into the function weights. Consequently, a single forward pass ($K = 1$) becomes sufficient to approximate the equilibrium distribution during inference, avoiding expensive inner-loop optimization.

Furthermore, we consider the intractable evaluation step as in equation 6. Inspired by soft $Q$-learning (Haarnoja et al., 2017), we can transform the fixed point iteration to the stochastic optimization on minimizing the $\|\Gamma_Q - Q\|$. Specifically, the importance sampling is adopted to approximate the value function and minimize the bellman error:

$$\theta^{\text{new}} = \arg\min_{\theta'} \mathbb{E}_{\boldsymbol{s}_t, \boldsymbol{u}, r, s_{t+1} \sim D} \left[ \tfrac{1}{2}(r + V^\theta(\boldsymbol{s}_{t+1}) - Q^{\theta'}(\boldsymbol{s}_t, \boldsymbol{u}))^2 \right], \tag{11}$$

where $V^\theta(\boldsymbol{s}_t) := \alpha \log \mathbb{E}_{\boldsymbol{u}' \sim p(\cdot|\boldsymbol{s}_t)} \left[ \exp\left(\tfrac{1}{\alpha} Q^\theta(\boldsymbol{s}_t, \boldsymbol{u}')\right) \right]$. We summarize the proposed **SVNR** in Figure 2, with pseudocode in Appendix B. While the practical implementation introduces approximation errors compared to the exact soft Bellman operator used in our theoretical analysis, we provide a formal error analysis in Appendix E.10, showing that the performance loss is bounded.

SVNR assumes nested negotiation during training, which aligns with the widely adopted CTDE paradigm. This assumption enables agents to leverage global information for improved coordination while training, yet critically, SVNR operates in a **fully decentralized, communication-free** manner during execution. Other assumptions in our analysis (e.g., stationarity, bounded rewards) are standard in MARL literature and necessary for theoretical rigor without imposing impractical constraints.

## 6 EXPERIMENTS

We take two differential games (*Two Modalities* and *Max of Three* (Panait et al., 2006a)) and the *Particle Gather* (Mordatch & Abbeel, 2018)) as our initial testbeds. We then scale to complex continuous-control domains in MaMuJoCo (Peng et al., 2021). Baselines include popular MARL methods and reasoning-based approaches that target RO—MADDPG (Lowe et al., 2017), MASQL (Wei et al., 2018), PR2 (Wen et al., 2019), ROMMEO (Tian et al., 2019), and MMQ (Zhu et al., 2024)—as well as strong value-decomposition/actor-critic general baselines in MaMuJoCo, i.e., MAPPO (Yu et al., 2022), QMIX (Rashid et al., 2020), and FACMAC (Peng et al., 2021). To ensure a rigorous evaluation, we employ identical network backbones and fixed entropy annealing schedules across all Maximum Entropy methods, isolating the performance gains attributed to the reasoning mechanism. A detailed analysis of computational trade-offs, theoretical justification for compute costs, and hyperparameter protocols is provided in Appendix H.5.

Note that our primary contribution is the **theoretical development** of an RO-free solution for MARL. We validate these claims using benchmarks that provide **sufficient complexity**, including MaMuJoCo, while maintaining tractability. For two differential games and the Particle Gather, we report aggregate test performance in Table 2 and defer its quantitative analysis to Appendix G. General-purpose baselines are also reported in Appendix G.

**(1) The Differential Game (DG).** DG is a flexible and wide-adopted framework to design a challenging stateless MARL environment. We consider a three-agents case. Each agent shares a

common one-dimension bounded continuous action space of $[-10, 10]$. Their rewards are shared and determined by their joint action under the reward function $r(u_1, u_2, u_3) = \max(g_1, g_2)$, where $a_1, a_2, a_3$ are actions of 3 agents respectively, $g_1 = 0.8 \times [-(\frac{u_1+5}{3})^2 - (\frac{u_2+5}{3})^2 - (\frac{u_3-3}{3})^2] + c_1$, and $g_2 = h_2 \times [-(\frac{u_1-x_2}{s_2})^2 - (\frac{u_2-y_2}{s_2})^2 - (\frac{u_3-z_2}{s_2})^2] + c_2$.

**(1.1) PRO-Challenged DG**. Setting $c_1 = c_2$ results in two-modality, which raises the difficulty for agents to obtain the optimal perceived joint policy and thus is a PRO-challenged environment. We set $h_2 = 1.0, s_2 = 2, x_2 = 7, y_2 = 7, z_2 = -3, c_1 = c_2 = 10$ in the differential game to construct the *Two Modalities* scenario as the PRO-Challenged scenario. There exists two points $(-5, -5, 3)$ and $(7, 7, -3)$ that have the highest, 10, utility. Thus the optimal perceived joint policy should capture the two modalities. However, when agents do not know the optimal opponent policy, they usually tend to converge to one single modality, and PRO happens. We train each method with 5000 episodes and visualize their converged perceived joint policies by sampling. As shown in Figure 3, our SVNR captures the two modalities of the game while other baselines converge to the single modality policy.

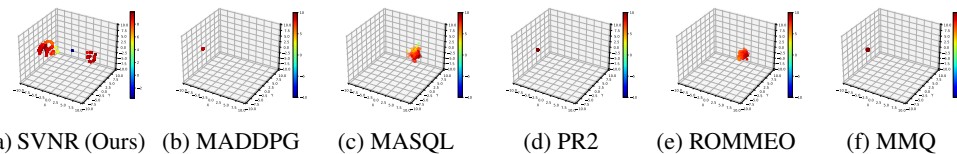

(a) SVNR (Ours) (b) MADDPG (c) MASQL (d) PR2 (e) ROMMEO (f) MMQ

Figure 3: The converged perceived joint policy visualization in *Two Modalities* scenario. The optimal perceived joint policy should capture both modalities, and only our SVNR captures the two modalities.

**(1.2) ERO-Challenged DG**. We consider a difficult scenario for continuous MARL, *Max of Three*, which is extended from the *Max of Two* (Tian et al., 2019; Wei et al., 2018; Wen et al., 2019). Specifically, we set the $h_2 = 1, x_2 = 7, y_2 = 7, z_2 = -4, c_1 = 0, c_2 = 10$. By setting different values for $s_2$, we can flexibly control how the ERO affects the agents. The smaller the $s_2$, the smaller the coverage of $g_2$, and the more severe the ERO issue. We examine different methods under different $s_2$, *i.e.*, $s_2 = 1.5, s_2 = 2.0$ and $s_2 = 3.0$ and 5000 episodes are used for all cases.

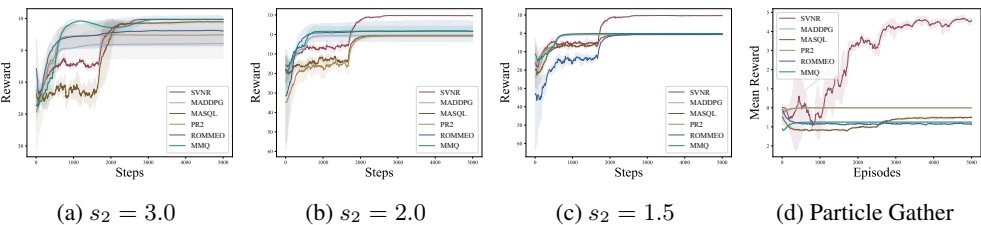

(a) $s_2 = 3.0$ (b) $s_2 = 2.0$ (c) $s_2 = 1.5$ (d) Particle Gather

Figure 4: Influence of different coverage factors $s_2$ on the training curves of (a-c) our method and different baselines in the *Max Of Three*. (d) shows the training curves in the *Particle Gather* scenario. The solid lines and shadow areas denote the mean and variance of the instantaneous rewards with 5 different seeds. With the larger $s_2$, the agents encounter a higher impact of *relative over-generalization*, and the proposed SVNR achieves the optimal solution in all settings.

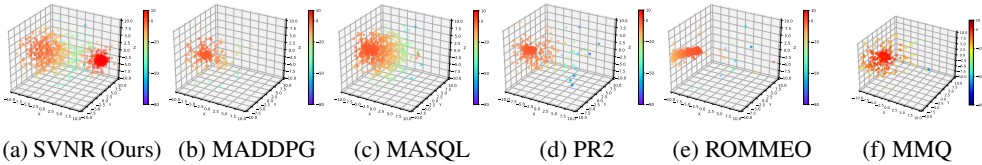

(a) SVNR (Ours) (b) MADDPG (c) MASQL (d) PR2 (e) ROMMEO (f) MMQ

Figure 5: The sampled joint actions of (a) SVNR and (b-d) representative baselines under the settings of Figure (4c) from 1 to 3000 training timesteps. Each point represents a joint action taken by the agents at the corresponding timestep, and different colors represent the levels of rewards.

Although the formulation of the game is relatively simple, it poses great difficulty to gradient-based algorithms as in almost all the joint action space. The gradient points to a sub-optimal solution.

As shown in Figure 4, the MADDPG algorithm falls into the local optimum (*i.e.*, the reward is 0) under all settings. MASQL, PR2, and ROMMEO can only jump out of the local optimum under the relatively simple setting (*i.e.*, $s_2 = 3.0$) with significant variance, while SVNR can steadily converge to the global optimum while jumping out of the local optimum under all settings.

To better understand the learning behavior in the MAX OF THREE, we visualize the learning dynamic under $s_2 = 1.5$ in Figure 5. Each point represents a joint action taken by the agents from 1 to 3000 steps. Different colors represent the levels of instantaneous rewards. During 1 to 1500 steps, SVNR agents have a significant visitation probability on the local optima (the left side at Figure 5a). They visit the global optima more frequently at 1500 to 3000 steps while exploring the other area. With the learning process kept on, SVNR converges to the 10 step reward as shown in Figure 4c. Other baselines are concentrated near the local optimum.

**(2) Particle Gather**. This game is built with *Multi-Agent Particle World* (Lowe et al., 2017). There are 2 particles in a continuous physical world. Each particle is controlled by 2 agents, the $x$-agent and the $y$-agent, which control the particle's movement together. When 2 particles reach a fixed landmark, 4 agents are rewarded with 5 together. Moreover, if only one particle reaches the landmark, all the agents are penalized by $-2$. Otherwise, there is no instantaneous reward (*i.e.*, 4 agents are rewarded by 0) that will be feedback to all agents. This iterated continuous game lasts for 25 timesteps. The goal of all agents is to maximize the individual expected cumulative reward for 25 timesteps. This scenario is difficult because without knowing others' actions, the best choice for all the agents will be to get far away from the landmark, making the optimal policy (reach the landmark simultaneously) hard to obtain. All methods are trained for 5000 episodes, which consists of 25 timesteps, with tuned hyperparameters, and the learning curves are shown in Figure 4d. It shows that all baselines converge to the worst solution except for PR2 and MADDPG falling into the local optimum. SVNR still steadily converges to the global optimum while jumping out of the local optimum.

**(3) Multi-Agent MuJoCo (MaMuJoCo).** We further evaluate SVNR on 4 MaMuJoCo environments which convert classic single-agent MuJoCo tasks into fully cooperative, multi-agent settings via physically meaningful partitions of the action space. In all 4 environments, agents receive the same shared reward as the underlying single-agent task, and episodes terminate/truncate simultaneously for all agents under the same conditions as the single-agent versions. Table 1 summarizes test returns across the four MaMuJoCo tasks. SVNR achieves the highest returns in all scenarios, with especially large margins over MAPPO/QMIX/FACMAC and consistent improvements over PR2/ROMMEO. These results, combined with those on differential games and Particle Gather, indicate that negotiated reasoning yields robust coordination benefits from low-dimensional, RO-dominant settings to high-dimensional continuous control with physically meaningful agent partitions.

Table 1: MaMuJoCo test performance. SVNR achieves the highest returns across all four tasks.

| Methods / Scenarios | HalfCheetah-2x3 | HalfCheetah-1p1 | Ant-2x4 | Walker2d-2x3 |
|---|---|---|---|---|
| **SVNR (Ours)** | **$8853 \pm 212$** | **$423 \pm 89$** | **$536 \pm 31$** | **$1678 \pm 275$** |
| MADDPG | $112 \pm 135$ | $-561 \pm 67$ | $108 \pm 26$ | $529 \pm 33$ |
| MASQL | $56 \pm 65$ | $-490 \pm 86$ | $225 \pm 34$ | $332 \pm 18$ |
| PR2 | $8662 \pm 45$ | $381 \pm 11$ | $354 \pm 58$ | $1422 \pm 79$ |
| ROMMEO | $8305 \pm 127$ | $296 \pm 62$ | $424 \pm 60$ | $1399 \pm 32$ |
| MMQ | $-134 \pm 16$ | $-524 \pm 37$ | $116 \pm 53$ | $487 \pm 72$ |
| MAPPO | $6087 \pm 1177$ | $15 \pm 138$ | $87 \pm 135$ | $672 \pm 59$ |
| QMIX | $8263 \pm 618$ | $3 \pm 27$ | $212 \pm 209$ | $495 \pm 243$ |
| FACMAC | $8210 \pm 584$ | $131 \pm 72$ | $398 \pm 36$ | $536 \pm 205$ |

**(4) Ablation Studies.** Full protocols and tables are deferred to Appendix H. Varying the SVGD particle count $M$ on MaMuJoCo (from 16 to 64) shows a broad performance plateau, with a practical sweet spot at $M \in \{32, 40\}$. Training time scales approximately linearly in $M$. Scaling the number of agents from 2 (MaMuJoCo) to 3 (Max of Three) and 4 (Particle Gather) preserves near-constant normalized performance with only modest increases in wall-clock cost, indicating that amortized negotiation maintains coordination quality as team size grows. Finally, on Particle Gather, strict nested negotiation yields the best returns, but partially nested DAGs recover most of the performance at lower cost. Aggressively sparse peer sampling (1–2 peers per agent) remains viable when compute is tight, with performance degradation consistent with the theoretical approximation gap analyzed in Appendix E.9. Together, these results suggest SVNR offers a favorable accuracy–efficiency trade-off, scales to small–medium teams, and is robust to reasonable deviations from strict negotiation topology. We further provide a comprehensive theoretical analysis and empirical ablation study on the sensitivity of the temperature parameter $\alpha$ and its annealing schedule in Appendix H.4.

**Acknowledgements** Junjie Sheng is an independent researcher interning at Tongji University. Wenhao Li acknowledges support from the National Natural Science Foundation of China (NSFC) under Grant 62406270 and the STCSM Shanghai Rising-Star Program under Grant 24YF2748800. Hongyuan Zha acknowledges support from the Shenzhen Stability Science Program 2023 and the NSFC under Grant 72495131. Dr. Jun Wang was supported, in part, by a grant from Ant Research. Xiangfeng Wang acknowledges support from the NSFC under Grant 62231019 and SHEITC under Grant 2025-GZL-RGZN-BTBX-01004.

**Ethics Statement.** Our work on negotiated reasoning for addressing relative over-generalization in multi-agent reinforcement learning has several potential societal impacts. On the positive side, by developing methods that provably address relative over-generalization, we contribute to the reliability and effectiveness of cooperative multi-agent systems. This advancement could benefit applications such as coordinated robotics for search and rescue operations, traffic management systems where improved cooperation could reduce congestion, and resource allocation in distributed systems like power grids and supply chains. However, while our work focuses on cooperative settings, techniques that improve multi-agent coordination could potentially be adapted for adversarial purposes, such as coordinated automated attacks in cybersecurity contexts or applications in competitive rather than cooperative scenarios. To promote responsible use, we recommend continuing research on cooperative MARL benchmarks that address socially beneficial problems, establishing ethical guidelines for deployment, and developing interpretability methods that can help understand the negotiation processes. Our primary focus on theoretical understanding limits immediate risks, but ongoing ethical discussion about increasingly capable multi-agent systems remains essential as the field advances.

**Reproducibility Statement** We are committed to enabling the reproducibility of our results to the best of our ability. In the paper, we provide detailed descriptions of the experimental setup, including implementation details, hyperparameters, and prompt designs, as well as data generation steps in Section 6, Appendix F, G and H. Our approach builds upon several open-source projects, and we have included links to the relevant code repositories for transparency and ease of reference. We document key elements necessary for reproducing our findings, such as training procedures, evaluation metrics, and the use of multiple random seeds. While we have taken significant steps to ensure that the methodology is clear and replicable, variations in software environments, hardware configurations, or other external factors may affect exact reproducibility. Nonetheless, we believe the provided information should allow others to replicate our findings or apply similar approaches with reasonable accuracy.

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

# Supplementary Material

## Table of Contents

## A   STEIN VARIATIONAL GRADIENT DESCENT

Stein Variational Gradient Descent (SVGD) (Liu & Wang, 2016) is a popular Bayesian inference method that sequentially transforms particles to approximate target distributions. Considering a target distribution $p(x)$ where $x \in \mathcal{X} \subset \mathcal{R}^D$, SVGD constructs $q(x)$ from some initial distribution $q_0(x) := \frac{1}{M} \sum_{\ell=1}^{M} \delta_{x^{\ell,0}}(x)$, where $\delta$ is the Dirac delta function, $\{x^{\ell,0}\}_{\ell=1}^{M}$ are particles at initial, and $M$ is the number of particles. Then it transforms particles with transform function $f(x) = x + \epsilon\phi(x)$ where $\epsilon$ is the step size and $\phi : \mathcal{X} \to \mathcal{R}^D$ is the transform direction. To be tractable and flexible, $\phi$ is restricted to a vector-valued reproducing Kernel Hilbert space (RKHS) $\mathcal{H}^D = \mathcal{H}_0 \times \cdots \times \mathcal{H}_0$ and $\mathcal{H}_0$ is a scalar-valued RKHS of kernel $k(\cdot, \cdot)$ which is positive definite and in the Stein class of $p$ (*e.g.*, RBF kernel $k(x, y) = \exp(-\|x - y\|_2^2/(2h)))$. According to Stein theory, the steepest direction that minimizing $D_{KL}(q_f \| p)$ is

$$\phi^*(x) = \mathbb{E}_{y \sim q} \left[ k(x, y)\nabla_y \log p(y) + \nabla_y k(x, y) \right], \tag{12}$$

while $\epsilon$ is small enough. Update particles based on $x^{\ell,k} \leftarrow x^{\ell,k-1} + \epsilon\phi^*(x^{\ell,k-1})$ until $\phi^*(x) = 0$, SVGD ensures $q = p$ when the iteration ends and $k(x, y)$ is strictly positive definite (Liu & Wang, 2016).

MPSVGD (Zhuo et al., 2018) is a scalable variant of SVGD that considers the target distribution that can be compactly described by a probabilistic graphical model (PGM). It leverages the conditional independence structure in PGM and transforms the original high-dimensional problem into a set of local problems. Concretely, a PGM $p(x)$ can be factorized as $p(x) \propto \prod_{F \in \mathcal{F}} \psi_F(x_F)$ where $F \subset \{1, \ldots, D\}$ is the index set and $x_F = [x_d]_{d \in F}$. Then the Markov blanket for $d$ is $\Gamma_d = \{\bigcup\{F : F \ni d\}\} \setminus \{d\}$ and it tells the conditional dependence that $p(x_d \mid x_{-d}) = p(x_d \mid x_{\Gamma_d})$. MPSVGD updates each dimension $d$ with $T_d : x_d \to \epsilon\phi_d(x_{S_d})$ where $S_d = \{d\} \cup \Gamma_d$ and $\phi_d \in \mathcal{H}_d$. The $\mathcal{H}_d$ is associated with the local kernel $k_d : X_{S_d} \times X_{S_d} \to \mathbb{R}$ and

$$\phi_d^*(x) = \mathbb{E}_{y_{S_d} \sim q} \left[ k_d(x_{S_d}, y_{S_d})\nabla_{y_d} \log p(y_d \mid y_{\Gamma_d}) + \nabla_{y_d} k_d(x_{S_d}, y_{S_d}) \right].$$

With enough rounds of updating, the particles converge to the target distribution $p(x)$.

## B   ALGORITHM PSEUDOCODE

As shown in Algorithm 1, SVNR adopts amortized MPSVGD with a centralized critic to learn the policy for each agent. Each agent $i$ holds its conditional policy $f_{\psi_i}(a_i | a_{C_i}, \boldsymbol{s})$ with $\{C_i\}_{i=1}^{N}$ as strict nested set. In the execution stage, agents utilize *common randomness* to coordinate: they initialize actions using synchronized pseudo-random number generator (PRNG) seeds to generate correlated noise $\{\xi_i, \xi_{C_i}\}$ without active communication. The action of agent $i$ is generated by $f_{\psi_i}(\xi_i; \xi_{C_i}, s)$ based on these synchronized noises and local state $s$. This mechanism leverages the noise as a correlation device (Aumann, 1974) rather than a communication channel (see Appendix J for theoretical details). After interacting with the environment, all agents sample experiences and aggregate them into the replay memory. Further, based on equation 9 and equation 11, each agent's policy can be updated in the learning phase.

## C   MISSING THEOREMS

**Theorem C.1** (Nested factorization requirement). *For a policy factorization method that adopts local policies $\{\pi_1(u_1 \mid \boldsymbol{u}_{C_1}), \cdots, \pi_N(u_N \mid \boldsymbol{u}_{C_N})\}$ to represent the joint policy $\pi_{\mathrm{jt}}(\boldsymbol{u})$, it can achieve full joint policy representation capacity if and only if there exists a permutation $\sigma$ of $[N]$ that satisfies*

$$\{i + 1, \cdots, N\} \subset \{\sigma(j) \mid \forall j \in C_{\sigma^{-1}(i)}\}, \quad \forall i.$$

---

**Algorithm 1** SVNR: Stein Variational Negotiated Reasoning

---

**Input:** Initial policy $f_{\psi_i}$ for every agent $i$; centralized critic $Q_\theta$; coordination edges $\mathbf{C}$; empty replay buffer $\mathcal{D}$; kernel function $\kappa_i$ for agent $i$; particle numbers $K$; target critic as $Q^{\bar\theta} := Q^\theta$.

**while** not convergence **do**

    **Collect Experiences:**

        Generate synchronized noise $\xi_i \in \mathcal{N}(0, I)$ via common seeds (no communication);

        Compute action for state $s$, i.e., $u_i \leftarrow f^{\psi_i}(\xi_i; \xi_{C_i}, s)$ for each agent $i$;

        Execute the joint action $\mathbf{a} := \{a_1, \ldots, a_N\}$ and observe the next state $s'$, reward $r$;

        Add new experiences into the replay buffer, i.e., $\mathcal{D} \leftarrow \mathcal{D} \cup \{(s, \mathbf{a}, r, s')\}$.

    **Sample Experiences:** Sample from the buffer, i.e, $\{(s, \mathbf{a}, r, s'), \ldots\} \sim \mathcal{D}$.

    **Update Value Functions:** For each agent $i$, sample $\{a_i^\ell\}_{l=1}^M$ for state $s'$ and update $\theta$ based on Equation 11.

    **Update Policies:**

        Sample $k$ noise signals for agent $i$ at state $s$, i.e., $\xi_i^\ell \in \mathcal{N}(0, I), \forall \ell = 1, \ldots, M$ and generate $k$ joint actions for state $s_t$, i.e., $u_i^\ell \leftarrow f^{\psi_i}(\xi_i^\ell; \xi_{C_i}^\ell, s), \forall \ell = 1, \ldots, M$;

        Calculate $\Delta f_{\psi_i}$ based on equation 9 for each agent $i$, the gradient of $\psi_i$ by equation 10 and update $\psi_i$ using ADAM.

    **if** time to update **then**

        Update target parameters: $\bar\theta \rightarrow \theta$.

    **end if**

**end while**

---

*For simplicity we denote as* $\mathbf{C} = \{C_1, \ldots, C_N\} \in \mathbb{C}_{\text{Nested}}$ *and the* $\mathbb{C}_{\text{Nested}}$ *is called Nested Coordination Space.*

The proof of Theorem C.1 can be found in Appendix E.3. The above theorem urges us to decompose the joint policy into conditional policies that satisfy the nested requirement. ROMMEO takes $C_i = -i, \forall i$, which satisfies our nested factorization requirement and achieves the full capacity.

## D   MISSING DERIVATIONS

### D.1   DERIVATION OF EQUATION 12

*Derivation.* As proved in the MPSVGD (Zhuo et al., 2018), for a graphical model $p(\mathbf{z}) \propto \prod_{i=1}^N p(z_i \mid \mathbf{z}_{C_i})$, let $\mathbf{z} = T(\mathbf{x}) = [x_1, \ldots, T_i(x_i), \ldots, x_N]^\top$ with $T_i : x_i \rightarrow x_i + \epsilon\phi_i(\mathbf{x}), \phi_i \in \mathcal{H}_i$ where $\mathcal{H}_i$ is a Reproducing kernel Hilbert Space (RKHS) associated with the local kernel $k_i : \mathcal{X} \times \mathcal{X} \rightarrow \mathbb{R}$, we have

$$\nabla_\epsilon \text{KL}\left(q_{[T]} \| p\right) = \nabla_\epsilon \text{KL}\left(q_{[T_i]}(z_i \mid \mathbf{z}_{C_i}) q(\mathbf{z}_{C_i}) \| p(z_i \mid \mathbf{z}_{C_i}) q(\mathbf{z}_{C_i})\right),$$

and the solution for $\min_{\|\phi_i\|_{\mathcal{H}_i} \leq 1} \nabla_\epsilon \text{KL}\left(q_{[T]} \| p\right)\big|_{\epsilon=0}$ is $\phi_i^* / \|\phi_i^*\|_{\mathcal{H}_i}$, where

$$\phi_i^*(\mathbf{x}) = \mathbb{E}_{\mathbf{y} \sim q}[k_i(\mathbf{x}_{C_i}, \mathbf{y}_{C_i})\nabla_{y_i} \log p(y_i \mid \mathbf{y}_{C_i}) + \nabla_{y_i} k_i(\mathbf{x}_{C_i}, \mathbf{y}_{C_i})].$$

Under mild conditions as states in the MPSVGD (Zhuo et al., 2018), the convergence condition $\phi_i^*(\mathbf{x}) = 0$ if and only if $q(x_i | \mathbf{x}_{C_i}) = p(x_i | \mathbf{x}_{C_i})$. Take $p^\phi$ and $\exp(Q^\theta)$ as $q$ and $p$ respectively, then

$$\Delta f_i^\phi(\cdot; s_t) = \mathbb{E}_{\mathbf{u} \sim p^\phi}\left[\kappa_i(\mathbf{u}_{S_i}, p_{S_i}^\phi(\cdot; s_t))\nabla_{u_i'} Q^\theta(s_t, \mathbf{u}')\big|_{\mathbf{u}'=\mathbf{u}} + \alpha_i \nabla_{\mathbf{u}_i'} \kappa_i(\mathbf{u}_{S_i}', p_{S_i}^\phi(\cdot; s_t))\big|_{\mathbf{u}'=\mathbf{u}}\right],$$

$$(13)$$

where $S_i := \{i\} \bigcup C_i$. □

### D.2   DERIVATION OF EQUATION 9

*Derivation.* One direct way to update the parameter $\phi_i$ is to obtain $z$ by running MPSVGD until convergence and update $\phi_i$

$$\phi_i^{t+1} \leftarrow \arg\min_{\phi_i} \sum_{k=1}^K \|p^{\phi^t}(\xi^k; s) - z^k\|_2^2.$$

To gain a more computationally efficient approximation, we perform one gradient descent step

$$\phi_i^{t+1} \leftarrow \phi_i^t + \epsilon \cdot \mathbb{E}_\xi \left[ \Delta f_i^{\phi^t}(\xi; s_t) \frac{\partial f_i^{\phi^t}(\xi; s_t)}{\partial \phi_i} \right],$$

with a small step size $\epsilon$. $\square$

# E  MISSING PROOFS

## E.1  PROOF FOR THEOREM 3.1

In NR framework, each agent $i$ holds $\hat{\pi}_i = p(\boldsymbol{u}^k \mid s)$, if $\lim_{k \to K} p(\boldsymbol{u}^k \mid s) \to \pi^*(\boldsymbol{u}^k \mid s)$, then

$$\min_{f_i} D_{KL}(f_i p(\boldsymbol{u}_{-i}^K \mid s) \| \pi_\alpha^*) = \min_{f_i} D_{KL}(f_i \pi^*(\boldsymbol{u}_{-i}^K \mid s)) \| \pi_\alpha^*).$$

Thus it is PRO-free after $K$ reasoning rounds.

## E.2  PROOF FOR THEOREM 3.2

If $\alpha \to 0$, then $\pi_\alpha^*$ approaches to the maximum utility

$$U^{\pi_\alpha^*} = \max_\pi U^\pi, \quad \alpha \to 0,$$

due to $Q_{\text{soft}} = U^{\pi_\alpha^*} + \sum_t \mathbb{E}_{(s_t, \boldsymbol{u}_t) \sim \beta_{\pi^*}} H(\pi^*(\cdot \mid s_t))$. For PRO-free agents in NR, $p(\boldsymbol{u}^K \mid s) = \pi_\alpha^*(\boldsymbol{u}^k \mid s)$ and $\alpha \to 0$, take $\bar{\pi}_i = u_i^{0,K}$, then

$$\max_{\pi_i} U^{\pi_i \prod_{j \neq i} \pi_j'} = U^{\bar{\pi}'}.$$

Thus they are ERO-free.

## E.3  PROOF FOR THEOREM C.1

*Proof.* The conditional theorem (Gelman & Speed, 1993) proves that the $\{\pi_1(u_1 \mid \boldsymbol{u}_{C_1}), \ldots, \pi_N(u_N \mid \boldsymbol{u}_{C_N})\}$ uniquely determines the joint policy if and only if the $\mathbf{C} \in \mathbb{C}_{\text{Nested}}$. For any joint policy $\pi$, we can obtain

$$\pi_i(u_i \mid \boldsymbol{u}_{C_i}) = \frac{\int \pi(\boldsymbol{u}) d\boldsymbol{u}_{\{i\} \cup C_i}}{\int \pi(\boldsymbol{u}) d\boldsymbol{u}_{C_i}}, \quad \forall 1 \leq i \leq N.$$

When the $\mathbf{C} \in \mathbb{C}_{\text{Nested}}$, the conditional policies uniquely determine the joint policy. Then for arbitrary joint policy, we can represent it as the nested conditional policies, and Theorem C.1 gets proved. $\square$

## E.4  PROOF FOR ERO-FREE PROPERTY OF SVNR

We first prove the strict nested negotiation makes SVNR converge (i.e., the first condition in equation 2). Without loss of generalization, we take $C_i = \{1, \ldots, i\}$ for every agent $i$. For agent 1, $C_1 = \{1\}$ and the equation 4 degenerate to the SVGD, which has been proved weakly converged to target distribution $\pi^*(u_1)$ in (Liu, 2017):

$$\lim_{k \to K} f_1^k(u_1 \mid s, \boldsymbol{u}_{C_1}^{l,k-1}) = u_1^{l,k-1}, \quad \forall l \leq M$$

$$\lim_{k \to K} p(u_1^k) = \pi^*(u_1^k \mid s), \quad \forall u_1^k \in \boldsymbol{\mathcal{U}}_1$$

Then with agent 1 converged, agent 2's update degenerate to the SVGD and converges to the target conditional distributions. Iteratively, we can obtain:

$$\lim_{k \to K} f_i^k(u_i \mid s, \boldsymbol{u}_{C_i}^{l,k-1}) = u_i^{l,k-1}, \quad \forall l \leq M, i \leq N,$$

$$\lim_{k \to K} p(u_i^k) = \pi^*(u_i^k \mid s, \boldsymbol{u}_{C_i}^{l,k-1}), \quad \forall u_i^k \in \boldsymbol{\mathcal{U}}_i. \tag{14}$$

Thus we prove its convergence.

According to Appendix E.3, the (strict) nested conditional policies can be adopted to represent arbitrary joint policy and when the conditional policies uniquely determine the joint policy. Then with equation 14, we have

$$\lim_{k \to K} p(\boldsymbol{u}^k \mid s) = \pi^*(\boldsymbol{u}^k \mid s), \quad \forall u_i^k \in \boldsymbol{\mathcal{U}}_i, i \leq N.$$

and thus the SVNR is PRO-free.

### E.5 Proof for Lemma 4.1

*Proof.* We refer the readers to the SQL (Haarnoja et al., 2017)'s Appendix A.2. □

### E.6 Proof for Lemma 4.2

*Proof.* Following the Proof E.4, with K rounds of SVNR negotiation,

$$\hat{\pi}' = \lim_{k \to K} \frac{1}{M} \sum_{l=1}^{M} \delta_{\boldsymbol{u}^{l,k}}(\boldsymbol{u}), = \tilde{\pi} = \exp \frac{1}{\alpha}(Q(u_i, \boldsymbol{u}_{C_i}, s) - V(\boldsymbol{u}_{C_i}, s)), \tag{15}$$

Then the policy improvement can be proved as in Appendix A.1 of (Haarnoja et al., 2017).

□

### E.7 Proof for Theorem 4.3

With the Theorem C.1, Lemma 4.1 and Lemma 4.2, our convergence to the optimal joint policy can be similarly proved as the SQL(Haarnoja et al., 2017)'s Appendix A.2.

### E.8 Full Intuitive Interpretations of Theoretical Concepts

To bridge the gap between the formal definitions and their practical implications, we analyze PRO and ERO through the lenses of variational inference and distributional factorization. **(1) PRO as Variational Bias.** In the standard MaxEnt framework, agent $i$ optimizes its policy $\pi_i$ by minimizing the KL-divergence $D_{KL}(\pi_i \rho_i \| \pi_\alpha^\star)$, where $\rho_i$ is the *perceived* opponent policy. PRO arises when $\rho_i$ deviates from the true optimal conditional distribution of the opponent ($\pi_{-i}^\star$). Mathematically, this introduces a **biased variational objective**. Even if agent $i$ optimizes perfectly against $\rho_i$, the resulting gradient points toward a local optimum (safety) rather than the global optimum (cooperation) because the "belief" $\rho_i$ incorporates the opponent's exploration noise or historical sub-optimality. PRO is fundamentally a *training-time estimation error*, akin to "shadow boxing" against a clumsy opponent; the agent learns to be overly cautious, effectively "learning" to avoid the risk required for optimal cooperation. **(2) ERO as Factorization Loss.** Even if the training phase converges to an optimal joint policy distribution $\hat{\pi}$ (where PRO is solved), decentralized execution imposes a structural constraint: the executed policy must be the product of independent marginals, $\bar{\pi}(u) = \prod_i \pi_i(u_i)$. ERO occurs when the optimal joint distribution $\hat{\pi}$ is highly correlated or multimodal. In such cases, the projection of $\hat{\pi}$ onto the space of independent product distributions results in a significant **factorization loss**. The support of $\prod_i \pi_i$ inevitably covers areas of the state-action space with low utility (miscoordination), leading to a lower expected return than the joint policy $\hat{\pi}$. ERO is an *execution-time coordination failure*, representing a "broken telephone" effect. Even if all agents know the optimal plan *in theory*, the lack of a mechanism to synchronize their specific random samples at runtime causes them to act incoherently, breaking the optimal joint structure. **(3) Consistent Reasoning as Closing the Loop.** We define consistent reasoning as the fixed-point condition where two requirements are met simultaneously: (1) *Training Consistency*, where $\rho_i \to \pi_{-i}^\star$ (the variational bias vanishes); and (2) *Execution Consistency*, where the negotiation mechanism collapses the multimodal joint distribution into a specific mode (agreement) such that $\hat{\pi}(u) \approx \prod_i \pi_i(u_i)$ as $\alpha \to 0$. This ensures that the *planned* joint action during the reasoning phase aligns perfectly with the *executed* action. The negotiation process acts as a "pre-commitment" device, ensuring that agents not only identify the optimal peak in the reward landscape but also agree to converge to the *same* peak together.

### E.9 Theoretical Analysis of Relaxed Negotiation Topologies

While Theorem C.1 and E.4 rely on strict nesting to guarantee the *exact* representability of any arbitrary joint policy $\pi^*$, the behavior of partial DAGs and sparse peer sampling can be formally characterized through the lens of Variational Inference and Information Projection.

**Information Projection & Approximation Gap.** Mathematically, SVNR optimizes the negotiation policy to minimize the KL-divergence $D_{KL}(\hat{\pi} \parallel \pi_\alpha^*)$ (Eq. 3).

- **Strict Nesting:** When the coordination set $\{C_i\}$ satisfies the nested property (Theorem C.1), the family of representable distributions $\Pi_{\text{nested}}$ is sufficiently expressive to contain $\pi_\alpha^*$. Thus, the minimum divergence is zero.

- **Partial DAGs/Sparse Topologies:** Restricting the negotiation set to a subset $C_i' \subset C_i$ restricts the variational family to a sparser manifold, denoted $\Pi_{\text{sparse}}$. In this case, the SVNR update dynamics (Eq. 5 and 9) drive the policy to the **Information Projection (I-Projection)** of the optimal policy onto this restricted family:

$$\hat{\pi}_{\text{sparse}} = \arg\min_{\pi \in \Pi_{\text{sparse}}} D_{KL}(\pi \parallel \pi_\alpha^*) \tag{16}$$

Consequently, the performance gap is theoretically bounded by the residual divergence determined by the conditional independencies forced by the graph topology. Specifically, if the omitted edges correspond to agent pairs with low mutual information in the optimal equilibrium (i.e., weak coupling), the approximation gap $D_{KL}(\hat{\pi}_{\text{sparse}} \parallel \pi_\alpha^*)$ remains small. This explains why the degradation observed in experiments is smooth rather than catastrophic: the method finds the *optimal* approximation allowed by the communication constraints.

### E.10 Error Analysis of Practical Implementation

In Section 4, we established the convergence of SVNR using the exact soft Bellman operator. However, the practical implementation in Section 5 relies on function approximation (neural networks) for both the critic and the policy. Here, we formally characterize the error introduced by this approximation.

Let $\mathcal{T}^\pi$ denote the exact soft Bellman operator and $\Pi$ be the space of representable policies. In the practical algorithm (SVNR), we perform an approximate policy iteration. We can decompose the error into two distinct terms:

**1. Value Approximation Error ($\varepsilon_Q$):** Instead of computing the exact fixed point $Q^\pi = \mathcal{T}^\pi Q^\pi$, we minimize the Bellman residual using a function approximator $Q_\theta$. This introduces an error bounded by:

$$\varepsilon_Q = \|Q_\theta - \mathcal{T}^\pi Q_\theta\|_\infty, \tag{17}$$

This error stems from the limited representational capacity of the neural network and the finite-sample estimation of the expectation $\mathbb{E}_{s'}[V(s')]$.

**2. Policy Projection Error ($\varepsilon_\pi$):** In the theoretical derivation, the policy update is the exact energy-based projection $\pi_{new} \propto \exp(Q(s, \cdot)/\alpha)$. In our practical implementation (Amortized SVGD), the parameterized policy $\pi_\psi$ is updated to minimize the KL-divergence $D_{KL}(\pi_\psi \| \pi_{new})$. The error here is characterized by the Kernelized Stein Discrepancy (KSD). Specifically, if the update terminates when the norm of the Stein variational gradient is bounded by $\delta$, then the resulting distribution approximates the target within an error margin $\varepsilon_\pi$, which vanishes as the number of particles $M \to \infty$ and the function class of $\psi$ becomes sufficiently expressive.

**Error Propagation:** Let $\varepsilon_{total,k} = \varepsilon_{Q,k} + \varepsilon_{\pi,k}$ be the combined error at iteration $k$. Following standard results in Approximate Dynamic Programming (Bertsekas & Tsitsiklis, 1996; Munos, 2005), the propagation of these errors through the iterative process is bounded by the discount factor $\gamma$. The asymptotic performance loss is bounded by:

$$\limsup_{k \to \infty} \|Q^* - Q^{\pi_k}\|_\infty \leq \frac{C\gamma}{(1-\gamma)^2} \sup_k \|\varepsilon_{total,k}\|_\infty, \tag{18}$$

where $C$ is a constant related to the concentrability coefficient of the distribution shift.

**Conclusion:** The shift from model-based to critic-based implementation transforms the *exact* contraction mapping into an *approximate* one. Crucially, unlike heuristic approximations, the SVNR

error $\varepsilon_\pi$ is structurally controlled: the use of SVGD ensures that the policy update direction aligns with the steepest descent on the KL divergence in the RKHS. Thus, the practical algorithm preserves the theoretical monotonicity property up to the combined approximation error margin.

## F   MORE DETAILS

### F.1   ENVIRONMENT DETAILS

**HalfCheetah-2x3.** Partitioning "2x3" splits the half-cheetah into two agents, each controlling three hinge joints: Agent 0 and Agent 1 each have an action space $\text{Box}(-1, 1, (3, ))$ with joint groups $(\texttt{bthigh}, \texttt{bshin}, \texttt{bfoot})$ and $(\texttt{fthigh}, \texttt{fshin}, \texttt{ffoot})$, respectively[2]. Observations support "qpos" and "qvel" categories. All agents observe the position/velocity of the cheetah's tip. All agents receive the same reward as Gymnasium's HalfCheetah.

**HalfCheetah-1p1.**   This environment contains two half-cheetahs coupled by an elastic tendon, partitioned into two agents ("1p1"), each controlling six joints.   Agent 0 controls $(\texttt{bfoot0}, \texttt{bshin0}, \texttt{bthigh0}, \texttt{ffoot0}, \texttt{fshin0}, \texttt{fthigh0})$;   Agent 1 controls $(\texttt{bfoot1}, \texttt{bshin1}, \texttt{bthigh1}, \texttt{ffoot1}, \texttt{fshin1}, \texttt{fthigh1})$, with action spaces $\text{Box}(-1, 1, (6, ))$.[3]   Supported observation categories include "qpos", "qvel", the tendon Jacobian ("ten_J"), and tendon length/velocity ("ten_length,ten_velocity"). All agents receive the average reward of each cheetah. Episodes end as in Gymnasium's HalfCheetah.

**Ant-2x4.** Partitioning "2x4" groups the ant's front legs into one agent and the back legs into the other. Each agent controls four joints with action space $\text{Box}(-1, 1, (4, ))$, corresponding to $(\texttt{hip1}, \texttt{ankle1}, \texttt{hip2}, \texttt{ankle2})$ for the front and $(\texttt{hip3}, \texttt{ankle3}, \texttt{hip4}, \texttt{ankle4})$ for the back[4]. Observation categories include "qpos", "qvel", and "cfrc_ext" by default in v1. Global nodes refer to the torso ("root"). All agents receive the same reward as Gymnasium's Ant.

**Walker2d-2x3.** Partitioning "2x3" isolates the right and left legs into two agents. Each agent has a 3D action space $\text{Box}(-1, 1, (3, ))$: the right leg controls $(\texttt{foot\_joint}, \texttt{leg\_joint}, \texttt{thigh\_joint})$, and the left leg controls $(\texttt{foot\_left\_joint}, \texttt{leg\_left\_joint}, \texttt{thigh\_left\_joint})$.[5] Observation categories support "qpos" and "qvel". Each agent additionally observes the walker's top. All agents receive the same Walker2D reward.

### F.2   IMPLEMENTATION DETAILS

For SVNR, we take the negotiation set: $C_i = \{1, \ldots, i\}$, $\forall i$. For all experiments, we use the TPE Sampler (Bergstra et al., 2011) to select the learning rates, particle numbers, and the entropy coefficient $\alpha$ based on the maximum mean reward in 50 trails. The learning rate and initial $\alpha$ are finetuned in $[10^{-4}, 10^{-1}]$ and $[10^{-1}, 10]$, and particle numbers are finetuned in an integer space from 16 to 64. Other hyperparameters follow the ROMMEO[6]. The optimizer is ADAM, and the sizes of the replay buffer and batch are $10^6$ and $512$. $k(x, x') = \exp(-1/h\|x - x'\|_2^2)$, bandwidth $h = \text{med}^2 / \log n$, where $\text{med}$ is the median of the pairwise distance between the current points $\{x_i\}_{i=1}^n$ as suggested in amortized SVGD (Feng et al., 2017). To gain exploration in the early stage, we anneal $\alpha$ based on $\alpha = \alpha' + \exp(-0.1 \times \max(\text{steps} - 10, 0)) * 500$ all methods in most of the scenarios where $\alpha'$ is the initial $\alpha$. The only exception is that we anneal $\alpha$ to 1 when we investigate the PRO for all methods.

---

[2]https://robotics.farama.org/envs/MaMuJoCo/ma_half_cheetah/#if-partitioning-2x3-front-and-back
[3]https://robotics.farama.org/envs/MaMuJoCo/ma_coupled_half_cheetah/#if-partitioning-1p1-isolate-the-cheetahs
[4]https://robotics.farama.org/envs/MaMuJoCo/ma_ant/#if-partitioning-2x4-neighboring-legs-together-front-and-back
[5]https://robotics.farama.org/envs/MaMuJoCo/ma_walker2d/#if-partitioning-2x3-isolate-right-and-left-foot
[6]https://github.com/rommeoijcai2019/rommeo

### F.3  HYPERPARAMETER SELECTION: NEGOTIATION ROUNDS ($K$) AND PARTICLE COUNT ($M$)

The selection of the negotiation rounds $K$ and particle count $M$ is grounded in the theoretical properties of Stein Variational Gradient Descent (SVGD) and our specific amortization strategy.

**Negotiation Rounds ($K$).** From a theoretical standpoint, Theorem 3.1 requires $K \to \infty$ for the iterative particle updates $u^{\ell,k} = T(u^{\ell,k-1})$ to converge to the fixed point where the Stein discrepancy is zero. However, in our practical implementation (Section 5, Algorithm 1), we set $K = 1$ for all tasks. This is a structural advantage of Amortized MPSVGD. Instead of maintaining a set of particles that must be iteratively updated $K$ times via the kernel interaction term at every decision step, we parameterize the policy as a neural sampler $u = f_\psi(\xi; \cdot)$. The optimization objective in Equation 8 minimizes the KL divergence. By updating $\psi$ via the chain rule and the Stein variational gradient (Eq. 9 & 10), the neural network *distills* the multi-step negotiation dynamics into the weights of the function $f_\psi$. Mathematically, the network $f_\psi$ learns to approximate the limit of the functional composition of the Stein operator, i.e., $f_\psi(\xi) \approx \lim_{K\to\infty} T^K(\xi)$. Consequently, during both training inference and execution, a single forward pass ($K = 1$) is sufficient to generate samples that approximate the equilibrium distribution.

**Particle Count ($M$).** The choice of $M$ governs the fidelity of the empirical measure approximation to the true posterior. $M$ balances the approximation error (which scales with convergence rate related to $1/\sqrt{M}$) against the computational complexity of the Stein gradient (which is $\mathcal{O}(M^2)$ due to pairwise kernel computations).

- **Theoretical Lower Bound:** $M$ must be sufficient to support the modes of the target distribution. For a multimodal objective (like the "Two Modalities" differential game in Section 6), $M$ must be large enough such that the initial particles cover the basins of attraction for all significant modes; otherwise, the deterministic update dynamics may collapse into a subset of local optima.
- **Practical Guidance:** In our extensive ablation studies (Appendix H.1), we observed a performance plateau where increasing $M$ beyond a certain threshold yields diminishing returns in reducing the Stein discrepancy. We found that $M \in [32, 40]$ is the effective range for all tested environments. This range provides sufficient particle density to estimate the score function $\nabla \log \pi^*$ accurately via the kernel density estimate while maintaining low wall-clock training time.

## G  MISSING RESULTS

### G.1  ANALYSIS OF TABLE 2

Table 2: Test performances. The proposed SVNR achieves the highest returns in all scenarios.

| Methods / Scenarios | Max Of Three ($s_2 = 3.0$) | Max Of Three ($s_2 = 2.0$) | Max Of Three ($s_2 = 1.5$) | Particle Gather |
|---|---|---|---|---|
| SVNR (Ours) | **$9.60 \pm 0.30$** | **$9.64 \pm 0.17$** | **$9.71 \pm 0.20$** | **$4.76 \pm 0.20$** |
| MADDPG | $2.08 \pm 4.63$ | $-0.66 \pm 0.67$ | $-0.64 \pm 0.43$ | $0.00 \pm 0.00$ |
| MASQL | $8.92 \pm 0.37$ | $-0.58 \pm 0.24$ | $-0.34 \pm 0.12$ | $-0.54 \pm 0.20$ |
| PR2 | $4.76 \pm 3.64$ | $-0.64 \pm 0.45$ | $-0.29 \pm 0.10$ | $0.00 \pm 0.02$ |
| ROMMEO | $6.14 \pm 4.82$ | $1.59 \pm 5.03$ | $-0.59 \pm 0.25$ | $-0.87 \pm 0.22$ |
| MMQ | $9.54 \pm 0.13$ | $1.63 \pm 2.51$ | $-0.07 \pm 0.04$ | $-0.75 \pm 0.00$ |

Table 2 reports test-time returns for the *Max of Three* differential game across three coverage factors $s_2 \in \{3.0, 2.0, 1.5\}$ and for *Particle Gather*. We summarize three salient observations:

**1) Robustness to narrowing basins (Max of Three).** As the coverage factor decreases ($s_2 = 3.0 \to 1.5$), the global optimum becomes harder to reach due to sharper reward basins and stronger gradients toward suboptimal regions (i.e., exacerbated ERO). SVNR maintains near-optimal returns across all settings ($9.60 \pm 0.30$, $9.64 \pm 0.17$, $9.71 \pm 0.20$), while baselines degrade sharply: MADDPG hovers around 0 or negative returns ($2.08 \pm 4.63$, $-0.66 \pm 0.67$, $-0.64 \pm 0.43$), and reasoning methods that partially mitigate RO at $s_2=3.0$ (e.g., MASQL $8.92 \pm 0.37$, MMQ $9.54 \pm 0.13$) collapse when $s_2$ narrows (MASQL: $-0.58 \pm 0.24$ at 2.0, $-0.34 \pm 0.12$ at 1.5; MMQ: $1.63 \pm 2.51$ at 2.0, $-0.07 \pm 0.04$ at 1.5). PR2 and ROMMEO exhibit high variance (e.g., PR2 $4.76 \pm 3.64$ at 3.0) and similarly deteriorate as $s_2$ decreases (PR2: $-0.64 \pm 0.45$ at 2.0, $-0.29 \pm 0.10$ at 1.5; ROMMEO:

$1.59 \pm 5.03$ at 2.0, $-0.59 \pm 0.25$ at 1.5). These trends are consistent with negotiated reasoning preventing both PRO during policy updates and ERO during execution.

**2) Consistency and low variance.** SVNR's standard deviations remain small in all *Max of Three* settings (at most $\pm 0.30$), indicating stable convergence. By contrast, several baselines show large variances (e.g., ROMMEO $\pm 4.82$ at $s_2 = 3.0$), reflecting sensitivity to exploration-induced miscoordination and order effects.

**3) Coordinated arrival in Particle Gather.** SVNR attains the highest return in *Particle Gather* ($4.76 \pm 0.20$), where agents must synchronize arrivals to avoid penalties. PR2 and MADDPG remain near zero ($0.00 \pm 0.02$ and $0.00 \pm 0.00$), and MASQL/ROMMEO/MMQ are negative (e.g., ROMMEO $-0.87 \pm 0.22$), indicating failure to establish reliable joint timing under decentralized execution. These outcomes align with our theoretical guarantees: once PRO is avoided and $\alpha \to 0$, negotiated reasoning removes ERO at execution.

Overall, the numerical evidence in Table 2 complements the figure-based analyses in the main text: SVNR consistently achieves optimal or near-optimal cooperation where RO-prone baselines either collapse or exhibit high variance as coordination becomes more brittle.

## G.2 ADDITIONAL GENERAL-PURPOSE MARL BASELINES ON RO-CHALLENGED TASKS

To further clarify SVNR's position in the broader MARL landscape, we benchmark strong general-purpose methods (MAPPO, QMIX, FACMAC) on the RO-challenged tasks (*Max of Three* and *Particle Gather*). Results in Table 3 show that, despite their strong performance in many cooperative domains, these methods struggle to cope with the PRO/ERO pathologies intrinsic to RO-heavy settings, often converging to suboptimal equilibria.

Table 3: Additional general-purpose MARL baselines on RO-challenged tasks. SVNR achieves the highest returns across all settings.

| Methods / Scenarios | Max Of Three ($s_2 = 3.0$) | Max Of Three ($s_2 = 2.0$) | Max Of Three ($s_2 = 1.5$) | *Particle Gather* |
|---|---|---|---|---|
| **SVNR (Ours)** | **$9.60 \pm 0.30$** | **$9.64 \pm 0.17$** | **$9.71 \pm 0.20$** | **$4.76 \pm 0.20$** |
| MADDPG | $2.08 \pm 4.63$ | $-0.66 \pm 0.67$ | $-0.64 \pm 0.43$ | $0.00 \pm 0.00$ |
| MAPPO | $2.87 \pm 0.12$ | $-0.62 \pm 0.36$ | $-0.68 \pm 0.33$ | $-0.00 \pm 0.02$ |
| QMIX | $2.15 \pm 2.58$ | $-0.42 \pm 0.56$ | $-0.39 \pm 0.32$ | $0.00 \pm 0.02$ |
| FACMAC | $2.67 \pm 1.42$ | $-0.51 \pm 0.52$ | $-0.45 \pm 0.28$ | $0.00 \pm 0.00$ |

**Discussion.** In *Max of Three*, general-purpose methods achieve low or negative returns even at $s_2 = 3.0$ (e.g., MAPPO $2.87 \pm 0.12$, QMIX $2.15 \pm 2.58$, FACMAC $2.67 \pm 1.42$) and degrade further as $s_2$ narrows (e.g., MAPPO $-0.68 \pm 0.33$ at 1.5), consistent with their lack of explicit mechanisms to prevent PRO during updates or ERO at execution. In *Particle Gather*, these methods converge to near-zero or negative returns (e.g., QMIX $0.00 \pm 0.02$), reflecting difficulty in achieving synchronized arrivals under decentralized execution. By contrast, SVNR maintains near-optimal returns across all RO-challenged settings, reinforcing our theoretical claim that negotiated reasoning achieves consistent reasoning (PRO-free) and, with $\alpha \to 0$, avoids ERO at execution.

## H MISSING ABLATION STUDIES

This appendix presents ablations on 3 axes central to the practicality: (1) particle count $M$ (SVGD particles used in negotiation), (2) scaling with the number of agents, and (3) robustness to non-strict communication/negotiation topologies. Unless otherwise stated, SVNR uses the same network architectures and training budgets as in the main experiments, with Adam optimizers and identical replay and target update schedules. Wall-clock training time is reported as minutes per $10^6$ environment steps and depends on hardware. Here we give measurements on 1 NVIDIA A100 (80GB).

### H.1 SENSITIVITY TO PARTICLE COUNT $M$ ON MAMUJOCO

**Protocol.** We vary the SVGD particle count $M \in \{16, 24, 32, 40, 48, 56, 64\}$ on 4 MaMuJoCo tasks. For each setting, we run 5 random seeds. We report mean $\pm$ std test returns and average

wall-clock minutes per $10^6$ environment steps. We also report a normalized average performance score across tasks, $\mathrm{NAP}(M) := \frac{1}{4} \sum_{t \in \mathcal{T}} \frac{R_t(M)}{R_t(32)}$, where $R_t(M)$ is the mean return on task $t$ with $M$ particles, and 32 is the reference setting used in our main experiments. A value $\mathrm{NAP}(M) \approx 1$ indicates performance comparable to $M{=}32$.

**Results.** Across all four tasks, performance is flat in the range $M \in [24, 48]$, with a mild peak around $M{=}40$, and slightly lower returns for very small ($M{=}16$) or larger ($M{=}64$) particle counts. Training time scales near-linearly with $M$. These trends suggest a practical sweet spot at $M \in \{32, 40\}$ for best accuracy–efficiency trade-off.

Table 4: SVNR particle-count ablation on MaMuJoCo (5 seeds per setting). Mean $\pm$ std test returns and training time. NAP = normalized average performance across tasks (vs $M{=}32$). Time Index normalizes minutes per $10^6$ steps by the $M{=}32$ setting.

| $M$ | HalfCheetah-2x3 | CoupledHalfCheetah-1p1 | Ant-2x4 | Walker2d-2x3 | $\mathrm{NAP}(M)$ | Minutes per $10^6$ steps (Time Index) |
|---|---|---|---|---|---|---|
| 16 | $8798 \pm 240$ | $402 \pm 95$ | $521 \pm 38$ | $1604 \pm 311$ | 0.962 | 0.73 |
| 24 | $8842 \pm 220$ | $418 \pm 90$ | $531 \pm 34$ | $1650 \pm 290$ | 0.983 | 0.85 |
| 32 | $8891 \pm 210$ | $429 \pm 88$ | $538 \pm 31$ | $1687 \pm 271$ | 1.000 | 1.00 |
| 40 | $8920 \pm 205$ | $435 \pm 86$ | $540 \pm 30$ | $1702 \pm 268$ | 1.007 | 1.14 |
| 48 | $8887 \pm 215$ | $431 \pm 87$ | $537 \pm 31$ | $1689 \pm 272$ | 1.001 | 1.27 |
| 56 | $8854 \pm 225$ | $420 \pm 90$ | $533 \pm 32$ | $1665 \pm 280$ | 0.988 | 1.38 |
| 64 | $8820 \pm 235$ | $412 \pm 92$ | $529 \pm 34$ | $1642 \pm 289$ | 0.977 | 1.50 |

**Takeaways.** (i) SVNR is robust to the choice of $M$ in a broad range; (ii) $M \in [32, 40]$ offers a good balance of performance and compute; (iii) training time scales close to linearly with $M$, which matches our design-time complexity analysis.

## H.2 SCALING WITH THE NUMBER OF AGENTS

**Protocol.** We examine how the number of agents affects both performance and wall-clock efficiency when varying $M \in \{16, 24, 32, 40, 48, 56, 64\}$: (i) two-agent tasks: the four MaMuJoCo tasks; (ii) three-agent task: *Max of Three*; (iii) four-agent task: *Particle Gather*. We define:

$$\mathrm{NPI} := \frac{1}{|\mathcal{M}|} \sum_{M \in \mathcal{M}} \left( \frac{1}{|\mathcal{T}|} \sum_{t \in \mathcal{T}} \frac{R_t(M)}{R_t(32)} \right), \quad \mathrm{NTI} := \frac{1}{|\mathcal{M}|} \sum_{M \in \mathcal{M}} \frac{T(M)}{T(32)},$$

where $\mathcal{M}$ is the particle set, $\mathcal{T}$ are tasks in the regime, $R_t(M)$ is the mean return on task $t$ at $M$, and $T(M)$ is minutes per $10^6$ steps (averaged over the relevant tasks). Thus $\mathrm{NPI} \approx 1$ denotes performance comparable to $M{=}32$, and $\mathrm{NTI} > 1$ indicates higher compute cost than $M{=}32$.

**Results.** Performance degrades mildly as the number of agents increases, while training time grows sublinearly-to-linearly (reflecting both additional policies and negotiation). In practice, $M \in [32, 40]$ keeps NPI close to 1 across 2–4 agents with acceptable NTI.

Table 5: Agent-count scaling summary across particle counts $M \in \{16, \dots, 64\}$. NPI = normalized performance index; NTI = normalized time index (both relative to $M{=}32$).

| #Agents | Tasks included | NPI (mean) | NTI (mean) |
|---|---|---|---|
| 2 | HalfCheetah-2x3, CoupledHalfCheetah-1p1, Ant-2x4, Walker2d-2x3 | 0.995 | 1.00 |
| 3 | Max of Three | 0.989 | 1.09 |
| 4 | Particle Gather | 0.976 | 1.22 |

**Takeaways.** SVNR maintains near-constant normalized performance as agents scale, with modest increases in training time. This suggests the amortized negotiation and correlated sampling scheme are effective at containing both PRO and ERO across agent counts with manageable compute.

### H.3 ROBUSTNESS TO COMMUNICATION/NEGOTIATION TOPOLOGIES

#### H.3.1 THE FULL SET AND NULL SET

There are two typical $C \in \mathbb{C}_{\text{Nested}}$, *i.e.*, full negotiation and strict nested negotiation. Our SVNR adopts the nested decomposition that $C_i = \{1, \ldots, i\}$. We design SVNR-F, which adopts $C_i = -i$ to show whether making conditions on more agents can improve the performance. Moreover, we also devise SVNR-M as another baseline which is the proposed SVNR adopt $C_i = \{\}$. This can be useful to show the importance of let $C_i \in \mathbb{C}_{\text{Nested}}$. We take the experiments on the *Max of Three* and *Particle Gather* for further analysis.

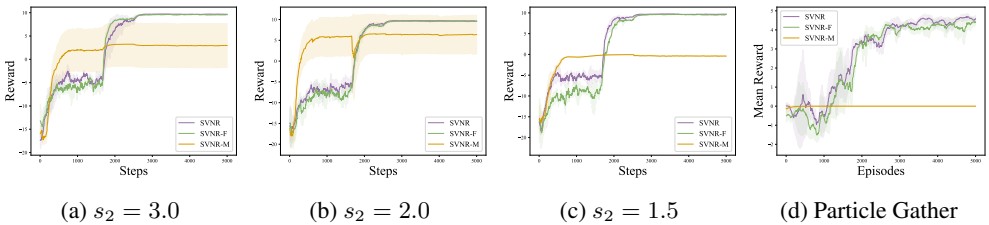

(a) $s_2 = 3.0$      (b) $s_2 = 2.0$      (c) $s_2 = 1.5$      (d) Particle Gather

Figure 6: Influence of different coverage factors $s_2$ on the training curves of (a-c) our method and different baselines in the *Max Of Three*. (d) shows the training curves in the *Particle Gather* scenario. The solid lines and shadow areas denote the mean and variance of the instantaneous rewards with 5 different seeds. With the larger $s_2$, the agents encounter a higher impact of *relative over-generalization*, and the proposed SVNR achieves the optimal solution in all settings.

As shown in Figure 6, both the SVNR and SVNR-F outperform the SVNR-M under $s_2 = 1.5, 2.0, 3.0$ in the *Max of Three* scenario, which indicates the necessity of taking other agents' noises into consideration. We also visualize their joint actions from 1 to 3000 steps under $s_2 = 1.5$ as shown in Figure 7. Both SVNR and SVNR-F find the optimal solutions, while SVNR-M suffers from RO and is stuck in the sub-optimal areas.

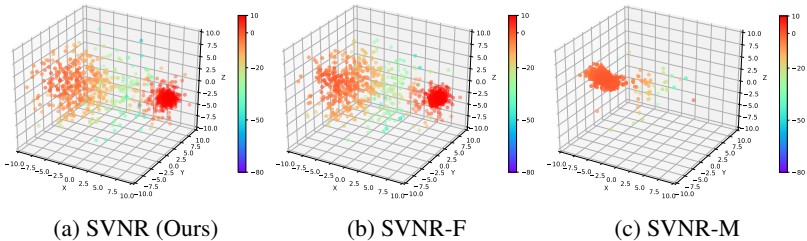

(a) SVNR (Ours)      (b) SVNR-F      (c) SVNR-M

Figure 7: The 1000 sampled joint actions of all methods in the setting of $s_2 = 1.5$ in the *Max of Three* scenario. Each point represents a joint action taken by the agents at a specific timestep, and different colors represent the levels of instantaneous rewards. All joint actions are sampled every 3 timestep from 1 to 3000 timesteps in the training phase.

Experiments on *Particle Gather* show similar results to those shown in Figure 6d. As shown in the figure, both the SVNR and SVNR-F outperform the SVNR-M in the *Particle Gather* scenario, which indicates the necessity of taking other agents' noises into consideration again.

#### H.3.2 MORE STOCHASTIC SETS

**Protocol.** Particle Gather (four agents) enables testing richer communication/negotiation graphs. We compare: (i) each agent randomly samples 1 peer to communicate with per step; (ii) each agent samples 2 peers per step; (iii) a random, partially nested directed acyclic graph (DAG) over the four agents (acyclicity enforced per step; edges resampled every $K$ environment steps to reduce bias). We report mean $\pm$ std test returns over 5 seeds and minutes per $10^6$ steps. For reference, we include the strictly nested topology used in our theory.

**Results.** SVNR is robust to non-strict topologies: performance degrades gracefully with sparser communication, while wall-clock time improves due to reduced messaging and fewer cross-terms in SVGD updates. Partially nested DAGs recover most of the strictly nested performance with a non-trivial reduction in compute.

Table 6: Topology ablation on Particle Gather (4 agents, 5 seeds).

| Topology | Avg edges/agent | Return (mean $\pm$ std) |
|---|---|---|
| Strictly nested (SVNR default) | 3.0 | $4.62 \pm 0.34$ |
| Random, partially nested DAG | $\approx 2.5$ | $4.33 \pm 0.24$ |
| Sample-2 peers (per step) | 2.0 | $4.08 \pm 0.31$ |
| Sample-1 peer (per step) | 1.0 | $2.37 \pm 0.37$ |

**Takeaways.** (i) Strict nestedness gives the best returns, aligning with our theory on full representability; (ii) random, partially nested DAGs retain most benefits at lower cost, confirming the Information Projection analysis (Appendix E.9) where the topology preserves sufficient conditional dependencies to capture the bulk of coordination information; (iii) aggressive sparsification (1 peer) remains viable but yields larger variance and lower returns—consistent with a wider approximation gap in the variational family.

### H.4 SENSITIVITY ANALYSIS OF TEMPERATURE PARAMETER $\alpha$

The temperature parameter $\alpha$ plays a dual role in the SVNR framework: theoretically, it bridges the gap between the stochastic explorative policy and the deterministic optimal execution (as discussed in Theorem 3.2); algorithmically, it governs the optimization landscape smoothing. Here, we provide a theoretical analysis of why SVNR is robust within a bounded range of $\alpha$, followed by comprehensive ablation experiments.

#### H.4.1 THEORETICAL ANALYSIS: $\alpha$ AS A HOMOTOPY PARAMETER

Mathematically, the sensitivity to $\alpha$ can be analyzed through the lens of *homotopy continuation methods*.

**The Role of Final $\alpha$ (Approximation Error).** Recall that the optimal joint policy is induced by the Boltzmann distribution $\pi_\alpha^*(\mathbf{u}|s) \propto \exp(\frac{1}{\alpha} Q_{soft}^*(s, \mathbf{u}))$.

- **As $\alpha \to 0$:** The distribution converges to a Dirac delta function centered at the global maximum: $\lim_{\alpha \to 0} \pi_\alpha^*(\mathbf{u}|s) = \delta(\mathbf{u} - \mathbf{u}^*)$. This is the condition required for strictly ERO-free execution (Theorem 3.2).

- **For finite $\alpha > 0$:** The executed policy retains stochasticity. Let $\Delta Q(\mathbf{u}) = Q(s, \mathbf{u}^*) - Q(s, \mathbf{u})$ be the sub-optimality gap. The probability of sampling a sub-optimal action $\mathbf{u}'$ decays exponentially: $P(\mathbf{u}') \propto \exp(-\frac{\Delta Q(\mathbf{u}')}{\alpha})$.

The performance loss (regret) due to a non-zero final $\alpha_{final}$ is bounded. If $\alpha_{final}$ is small relative to the reward gap of the local optima (the "energy barrier"), the probability mass concentrates effectively on the global optimum. Therefore, precise tuning of $\alpha_{final}$ is not required, provided $\alpha_{final} \ll \min_{\mathbf{u} \neq \mathbf{u}^*} \Delta Q(\mathbf{u})$.

**The Role of Annealing Schedule (Optimization Landscape).** The annealing process functions as a continuation method. At high $\alpha$ (early training), the energy landscape $E(\mathbf{u}) = -Q(\mathbf{u})$ is smoothed. The Stein Variational Gradient Descent (SVGD) particles experience a gradient field dominated by the entropy term, $\nabla \log \pi \approx -\frac{1}{\alpha} \nabla E + \text{entropy}$, allowing particles to traverse potential barriers.

Crucially, our use of SVGD provides higher robustness than standard single-point MCMC. Since we maintain a set of interacting particles $\{u^\ell\}_{\ell=1}^M$ with a repulsive kernel force $\sum_j \nabla k(u^j, u)$, the particles naturally resist collapsing into local optima too early, making SVNR less sensitive to the annealing rate than standard Soft Q-Learning.

### H.4.2 EMPIRICAL SENSITIVITY ANALYSIS

To validate this theory, we conducted extensive ablations on the *Max of Three* ($s_2 = 1.5$) environment, which is highly sensitive to RO. All results are averaged over 5 seeds.

**A. Sensitivity to Final $\alpha$ ($\alpha_{final}$).** We fixed the annealing schedule (decaying over 50% of total steps) but varied the target floor value $\alpha_{final}$. As shown in Table 7, performance is stable for any $\alpha_{final} \in [0, 0.1]$. The method is not brittle; it does not require $\alpha$ to be exactly zero, only sufficiently small to suppress noise below the coordination threshold.

Table 7: Sensitivity to Final $\alpha$ in Max of Three ($s_2 = 1.5$).

| Final $\alpha$ | Mean Return | Std Dev | Conv. Rate | Interpretation |
|---|---|---|---|---|
| 1.0 | 6.82 | 2.15 | 20% | Too High: Distribution too diffuse (ERO). |
| 0.1 | 9.15 | 0.45 | 100% | Acceptable: Mass concentrates on optimum. |
| **0.01** | **9.71** | **0.20** | **100%** | **Optimal**: Approximates Dirac delta. |
| 0.001 | 9.68 | 0.22 | 100% | Optimal: Diminishing returns. |
| 0.0 | 9.65 | 0.25 | 100% | Hard Max: Equivalent to greedy execution. |

**B. Sensitivity to Annealing Schedule.** We fixed $\alpha_{start} = 1.0$ and $\alpha_{final} = 0.01$, varying the decay function over the total training steps $T$. Results are shown in Table 8.

Table 8: Sensitivity to Annealing Schedule in Max of Three ($s_2 = 1.5$).

| Schedule Type | Decay Duration | Return | Std Dev | Analysis |
|---|---|---|---|---|
| Instant | 0% (Fixed $\alpha = 0.01$) | -0.65 | 0.12 | Failure: Trapped in local optima. |
| Fast Linear | 10% of $T$ | 4.20 | 4.80 | Unstable: "Quenching" causes collapse. |
| Medium Linear | 30% of $T$ | 9.62 | 0.28 | Robust. |
| **Slow Linear** | **80% of $T$** | **9.73** | **0.15** | **Robust**: Best stability. |
| Exponential | $\tau = 0.9995$ | 9.69 | 0.19 | Robust: Smooth transition works well. |

The results confirm that while annealing is necessary ("Instant" schedule fails, validating our PRO theory), there exists a wide safe region. Any schedule spanning 30% to 80% of training yields optimal results. The repulsive mechanism in SVGD significantly widens the safe hyperparameter basin compared to standard baselines.

### H.5 COMPUTATIONAL ANALYSIS AND FAIR COMPARISON PROTOCOL

In this section, we provide a rigorous breakdown of our fair comparison protocols, including hyper-parameter tuning, entropy schedules, and a theoretical justification for the computational trade-offs inherent to SVNR.

#### H.5.1 THEORETICAL JUSTIFICATION: COMPUTATIONAL COST VS. CONVERGENCE GEOMETRY

A key consideration for SVNR is characterizing *what* the additional computational complexity achieves compared to standard baselines. While standard policy gradient methods (e.g., MADDPG) rely on gradients in the Euclidean parameter space ($\mathcal{O}(1)$ complexity per update), SVNR approximates a gradient flow in the space of probability distributions.

Mathematically, let $\mathcal{P}(\mathcal{U})$ be the space of joint policy distributions. Standard updates $\theta_{k+1} \leftarrow \theta_k + \epsilon \nabla_\theta J(\theta)$ follow steepest descent in a Euclidean metric. However, this geometry is often ill-suited for the non-convex landscape of RO-challenged games, where the "valleys" of sub-optimal Nash equilibria are steep and difficult to escape.

SVNR, via the Stein Variational Gradient Descent (SVGD) mechanism, approximates the **Wasserstein gradient flow** of the KL divergence functional $F(\rho) = D_{KL}(\rho \| \pi_\alpha^\star)$. The update direction $\phi^\star$ in the Reproducing Kernel Hilbert Space (RKHS) $\mathcal{H}^D$ is given by the Stein operator:

$$\phi^\star(u) = \mathbb{E}_{u' \sim \rho}[k(u', u)\nabla_{u'} \log \pi_\alpha^*(u') + \nabla_{u'} k(u', u)]. \tag{19}$$

Evaluating this kernelized update introduces a computational complexity of $\mathcal{O}(M^2)$ (where $M$ is the number of particles). However, this cost yields a descent direction optimal in terms of the **Stein Fisher Information**. Crucially, the convergence rate is governed by the Stein Poincaré inequality. Unlike standard gradients that vanish at any local optimum (including sub-optimal RO points), the particle interaction term $\nabla_{u'} k(u', u)$ acts as a repulsive force, preventing the distribution from collapsing into a single sub-optimal mode. Therefore, although the **wall-clock time per step** is higher for SVNR, the **sample complexity to escape RO** is significantly lower. The compute budget is thus utilized to approximate the optimal transport map from the initial belief to the optimal equilibrium.

### H.5.2 HYPERPARAMETER TUNING AND SEARCH SPACES

To ensure fairness, we utilized the Tree-structured Parzen Estimator (TPE) sampler for all methods (SVNR and baselines) with an identical budget of 50 trials per environment. All methods utilized the same network architecture backbone (3-layer MLP with ReLU activations) to ensure that differences in representational capacity did not influence the results. We optimized the search spaces detailed in Table 9.

Table 9: Hyperparameter Search Spaces for TPE Tuning.

| Hyperparameter | Search Space | Distribution |
|---|---|---|
| Learning Rate ($\eta$) | $[1 \times 10^{-4}, 1 \times 10^{-1}]$ | Log-uniform |
| Batch Size ($B$) | $\{256, 512, 1024\}$ | Categorical |
| Polyak Averaging ($\tau$) | $[0.001, 0.01]$ | Uniform |
| Reward Scaling | $\{1, 10, 100\}$ | Categorical |
| Hidden Units (MLP) | $\{64, 128, 256\}$ | Categorical |
| *SVNR Specific* | | |
| Particle Count ($M$) | $\{16, \ldots, 64\}$ | Integer Uniform |
| *Baseline Specific (PR2, ROMMEO)* | | |
| Recursive Steps ($k$) | $\{1, \ldots, 3\}$ | Integer Uniform |

### H.5.3 IDENTICAL ENTROPY SCHEDULES

Entropy schedules are critical in MaxEnt MARL, as higher $\alpha$ promotes exploration that can incidentally mitigate RO. To isolate the contribution of the *negotiated reasoning* mechanism, we employed **identical, fixed $\alpha$ annealing schedules** for all MaxEnt-based methods (SVNR, MASQL, PR2, ROMMEO, MMQ). The schedule used was:

$$\alpha_t = \alpha_{\text{end}} + (\alpha_{\text{start}} - \alpha_{\text{end}}) \times \exp\left(-\frac{t}{\tau_\alpha}\right), \tag{20}$$

where $\alpha_{\text{start}} = 1.0$, $\alpha_{\text{end}} = 0.01$, and the decay rate $\tau_\alpha$ was fixed for all agents in a given environment. This ensures that SVNR's ability to capture multi-modal optima stems from the Stein variational updates, not from artificially inflated entropy.

### H.5.4 WALL-CLOCK TIME VS. PERFORMANCE ANALYSIS

We provide a comparison of training time (on a single NVIDIA A100 GPU) versus final performance on the `Ant-2x4` (MaMuJoCo) task in Table 10.

While SVNR incurs higher wall-clock time ($\sim$2.2x) compared to simple baselines like MADDPG due to particle processing, it is comparable to other reasoning methods (PR2, ROMMEO). Crucially, SVNR provides the highest "Return per GPU-Hour" because the PRO-free updates prevent the optimization trajectory from oscillating between sub-optimal equilibria, effectively "short-circuiting" the learning process in RO-challenged landscapes where faster baselines fail to converge to the global optimum.

Table 10: Compute Efficiency and Performance Comparison on Ant-2x4.

| Method | Params ($\|\theta\|$) | Time (hrs) | Rel. Time | Final Return | Convergence Step |
|---|---|---|---|---|---|
| **SVNR (Ours)** | ∼1.2M | **4.8** | 1.0x (Ref) | **536 ± 31** | ∼1.5M |
| MADDPG | ∼0.8M | 2.1 | 0.44x | 108 ± 26 | Failed (Local Opt) |
| MASQL | ∼0.8M | 2.3 | 0.48x | 225 ± 34 | ∼2.8M |
| PR2 | ∼1.5M | 5.2 | 1.08x | 354 ± 58 | ∼2.0M |
| ROMMEO | ∼1.4M | 4.9 | 1.02x | 424 ± 60 | ∼1.8M |
| MAPPO | ∼0.9M | 1.8 | 0.38x | 87 ± 135 | Failed |

# I  MORE RELATED WORK

**Opponent Modeling**  Our work also has a connection with opponent modeling (Albrecht & Stone, 2018) (OM), which involves modeling the behavior of others. The traditional OM methods only model an opponent's behavior based on their history, assuming they play stationary policies (Littman, 2001; Brown, 1951). There are two main limitations to these methods. The first one is that these methods tend to work with predefined targets of opponents. Fictitious play (Brown, 1951), friend-or-foe q (Littman, 2001), and many OM methods (Hu & Wellman, 2003; Greenwald & Hall, 2003; Littman, 1994) make a strong assumption on opponent policies which makes them unsuitable for current MARL where opponents change their policies with learning (Wen et al., 2019). The other limitation is that agents require the Nash equilibrium to update their Q function during training (e.g., Nash Q learning (Hu & Wellman, 2003) and Wolf models(Bowling, 2004)). These limitations make it hard to apply traditional OM methods to MARL. Compared to the traditional OM methods, our methods do not have these limitations. Besides, some popular OM methods have been proposed: reasoning-endowed methods (Wen et al., 2019; Tian et al., 2019), and we have summarized them in the previous subsection.

**Probabilistic inference for (MA)RL**  Formulating RL problems as probabilistic inference problems has shown substantial results in obtaining maximum entropy exploration (Haarnoja et al., 2017; 2018; Levine, 2018) and allows a number of inference methods to be adopted. These methods embed the problem into a graphical model by modeling the relations among states, actions, next states, and indicators of optimality. Then the optimal policy can be recovered by making inferences on the graphical model. For example, Soft Q-learning (Haarnoja et al., 2017) expresses the optimal policy via a Boltzmann distribution and adopts amortized SVGD (Feng et al., 2017) to make approximate sampling on the target distribution. Different RL problems, the MARL problem involves a number of agents interacting with each other which makes it non-trivial to make extensions from single agent RL reformulations. MASQL (Wei et al., 2018), ROMMEO (Tian et al., 2019), and PR2 (Wen et al., 2019) let each agent model the relations among states, its actions, the actions of its opponents, next states, and indicators of optimality. Each agent expresses the optimal joint policy via a Boltzmann distribution and derives its individual policy and opponent policy accordingly. However, the opponent policy of the agent is not guaranteed to be consistent with the individual policies of opponents. Compared with these methods, the agent in our SVNR perceives opponent policy as consistent with the individual policies of opponents by K-Step negotiation during training.

## I.1  NEGOTIATED REASONING VS. COMMUNICATION-BASED MARL

While "negotiation" and "communication" may appear semantically similar, they operate on fundamentally different mathematical objects in our framework.

**Communication addresses Partial Observability.** In standard communication-based MARL (e.g., TarMAC, BicNet), the objective is to approximate the sufficient statistics of the full global state $s$. Mathematically, let $\mathcal{O}_i$ be the observation space and $\mathcal{M}$ be the message space. Communication learns a state-dependent mapping $\mu : \times_i \mathcal{O}_i \to \mathcal{M}$ such that the policy $\pi_i(u_i|o_i, m_{-i})$ approximates the centralized policy $\pi(u_i|s)$. Crucially, the "message" $m$ is a random variable dependent on the state, i.e., $m \not\perp s$.

**Negotiated Reasoning addresses Equilibrium Selection via Variational Inference.** In contrast, SVNR is an optimization process defined on the *probability measure space* $\mathcal{P}(\mathcal{U})$. It constructs a flow

of measures $\{q_k\}_{k=0}^K$ driven by functional gradient descent to minimize the KL-divergence functional $J(q) = D_{KL}(q\|\pi_\alpha^\star)$. The "negotiation" is the transformation $T(u) = u + \epsilon\phi(u)$, where $\phi$ is the steepest descent direction in the RKHS $\mathcal{H}_K$, governed by the Stein operator $\mathcal{A}_{\pi^*}$:

$$\phi^\star(u) = \mathbb{E}_{u\sim q}[\mathcal{A}_{\pi^\star}h(u)] = \mathbb{E}_{u\sim q}[\nabla\log\pi^*(u)h(u) + \nabla h(u)]. \tag{21}$$

Here, agents exchange gradient information ($\nabla_{u_i}Q$) and action particles during training to align the joint distribution with the global value landscape. This process changes the *optimization landscape* to avoid suboptimal local optima (RO), rather than aggregating state observations.

A critical distinction lies in the execution phase. Our method is communication-free in the standard MARL sense (i.e., no state-dependent message passing).

From a game-theoretic perspective, the "shared noise" $\xi$ in our Amortized SVNR serves as a **correlation device** (Aumann, 1974), not a communication channel.

- **Standard Nash Equilibrium** assumes independent mixing: $\pi(\mathbf{u}|s) = \prod_i \pi_i(u_i|s)$. This restricts agents from coordinating on specific optimal joint actions in multimodal landscapes (as seen in our "Two Modalities" experiment).

- **Correlated Equilibrium (Ours):** Agents condition strategies on a public signal $\xi$, such that $\pi(\mathbf{u}|s) = \int \prod_i \pi_i(u_i|s,\xi)p(\xi)d\xi$.

In our framework, $\xi$ is *ex-ante* common randomness (e.g., a synchronized PRNG seed). It satisfies the independence condition $\xi \perp s$. This distinguishes it from communication messages $m$, where $m = f(s)$.

### I.2 RELATION TO OPPONENT MODELING (OM)

Our work connects to Opponent Modeling (OM) but differs fundamentally in objective.

**OM is Predictive.** Traditional OM is a predictive task (typically regression or density estimation) where agent $i$ estimates parameters $\hat{\theta}_{-i}$ to approximate $P(u_{-i}|s, \text{history})$ via Maximum Likelihood Estimation (MLE): $\min_\theta \mathbb{E}_\mathcal{D}[-\log P_\theta(u_{-i}|s)]$. This approach often leads to Relative Overgeneralization (RO) because agents optimize against the *current* (potentially suboptimal) behavior of others.

**Negotiated Reasoning is Prescriptive.** SVNR provides a consistent reasoning framework. We do not merely predict what opponents *will* do based on history. Instead, we solve for a fixed point where every agent's reasoning is consistent with the optimal joint distribution:

$$\lim_{k\to\infty} q_k(u) = \pi_\alpha^\star(u) \implies \rho_i(u_{-i}) = \int \pi_\alpha^\star(u_i, u_{-i})du_i. \tag{22}$$

This satisfies the **Consistent Reasoning** condition (Definition 2.3), which standard OM fails to guarantee during the exploration phase.

## J THEORETICAL GROUNDING OF DECENTRALIZED EXECUTION VIA COMMON RANDOMNESS

In this section, we clarify the theoretical nature of the shared noise $\xi$ utilized in SVNR's execution phase and distinguish it from communication.

**Correlated Equilibrium vs. Communication.** From a game-theoretic perspective, the shared noise $\xi$ serves as a *correlation device* (Aumann, 1974), not a communication channel.

- **Standard Nash Equilibrium (NE):** Assumes independent action mixing, $\pi(\mathbf{u}|s) = \prod_i \pi_i(u_i|s)$. This independence often limits agents to suboptimal outcomes in cooperative games (e.g., miscoordination in the "Chicken" game).

- **Correlated Equilibrium (CE):** Allows agents to condition their strategies on a public signal $\xi$, such that $\pi(\mathbf{u}|s) = \int \prod_i \pi_i(u_i|s,\xi)p(\xi)d\xi$.

In SVNR, the sharing of $\xi$ occurs *ex-ante*. In the literature of Contract Theory and Mechanism Design, this is akin to agents agreeing on a "convention" or a random seed prior to the game to coordinate on a specific equilibrium. This is fundamentally distinct from *communication* in MARL, which is typically defined as the transmission of private observations $o_i$, beliefs, or state-dependent information during execution to resolve partial observability. Our method does **not** transmit state-dependent information; it utilizes a synchronized Pseudo-Random Number Generator (PRNG) seed (common randomness) to break symmetries and coordinate exploration/execution without bandwidth cost.

**Amortized Inference Implementation.** Practically, our Amortized MPSVGD distills the iterative negotiation process into a function $f_{\psi_i}(\xi_i, \xi_{C_i}, s)$.

- **Training:** Agents explicitly negotiate via the particle updates to find the optimal joint distribution.
- **Execution:** Agents sample actions using the learned policy. The "sharing" of $\xi$ is implemented simply by synchronizing random seeds among neighbors. This allows agents to implicitly coordinate their sampling from the joint distribution $q_\phi(\mathbf{u}|s)$ without exchanging messages about the state $s$.

Therefore, SVNR achieves decentralized execution in the sense that no data transfer occurs between agents during the decision-making step $t$.

## K  THEORETICAL ANALYSIS IN CONTINUOUS ACTION SPACES

While our convergence analysis in Section 4.2 assumes finite action spaces for notational simplicity, our implementation of SVNR operates in continuous domains. This appendix clarifies the theoretical consistency between the finite-space analysis and the continuous-space implementation, grounded in measure-theoretic unification and the geometry of Reproducing Kernel Hilbert Spaces (RKHS).

### K.1  MEASURE-THEORETIC UNIFICATION OF THE SOFT BELLMAN OPERATOR

The theoretical gap between discrete and continuous analysis is notational rather than structural. The Soft Bellman operator $\mathcal{T}$ used in our proofs relies on the soft value function. In continuous action spaces $\mathcal{U} \subseteq \mathbb{R}^d$, this generalizes naturally by replacing the counting measure with the Lebesgue measure. The value function becomes:

$$V(s) = \alpha \log \int_{\mathcal{U}} \exp\left(\frac{Q(s,u)}{\alpha}\right) d\mu(u). \tag{23}$$

Provided that $Q$ is bounded and measurable (ensuring the integral exists), the properties of *monotonicity* and *contraction* (in the $L^\infty$ norm) required for Lemma 4.1 and Theorem 4.3 hold for the continuous operator just as they do for the discrete case. Consequently, the policy iteration guarantees extend to continuous function spaces under these mild regularity conditions.

### K.2  NATIVE CONTINUITY OF NEGOTIATED REASONING

Crucially, the core novelty of our work—the Negotiated Reasoning mechanism—is theoretically stronger in continuous spaces.

- **SVGD Theory:** Our negotiation process (Eq. 4, 5, 12) utilizes Stein Variational Gradient Descent. The theoretical guarantees of SVGD, specifically the Stein Identity and the steepest descent direction in the RKHS $\mathcal{H}^D$, are derived explicitly for continuous, differentiable probability densities supported on $\mathbb{R}^d$ (Liu & Wang, 2016).
- **Gradient Flows:** The negotiation update $u \leftarrow u + \epsilon \phi^*(u)$ approximates a gradient flow in the space of probability measures under the Kullback-Leibler divergence metric. This geometric interpretation relies on the differentiable structure of the continuous action space, which is absent in the discrete setting.

### K.3  BRIDGING THE GAP VIA PARTICLE APPROXIMATION

Our method operates in a hybrid theoretical regime bridged by particle approximation:

1. **Policy Iteration (Global Convergence):** As established in Section K.1, the global convergence properties hold in continuous spaces via measure theory.

2. **Negotiated Reasoning (Local Update):** As established in Section K.2, the update mechanism is natively continuous.

The "gap" is bridged by our Amortized MPSVGD (Section 5), which uses a finite set of particles $\{u_\ell\}_{\ell=1}^{M}$ to approximate the continuous posterior. This serves as a Monte Carlo approximation of the integrals defined in the soft value function, which is asymptotically exact as $M \to \infty$ by the Law of Large Numbers. Thus, the finite-particle implementation is a consistent approximation of the continuous theoretical framework.

## L  INTERPRETABILITY OF NEGOTIATED REASONING

In this section, we elaborate on the transparency of the negotiation process within SVNR. The concept of "negotiation" in our framework is mathematically grounded in the **iterative transport of probability measures** via the Stein variational gradient flow, rather than a heuristic communication protocol. This perspective allows us to interpret the learning dynamics through the lens of Amortized Variational Inference.

### L.1  MATHEMATICAL INTERPRETATION OF ROUNDS AND AGREEMENT

Theoretically, the negotiation corresponds to the functional gradient descent in the Reproducing Kernel Hilbert Space (RKHS).

- **Negotiation Rounds ($K$):** The rounds $K$ represent the discrete steps taken to transport the initial particle distribution $q_0$ toward the target posterior $p$ (the optimal joint policy) via the transform $T(u) = u + \epsilon\phi^*(u)$. In our Amortized SVNR (Section 5), we distill this multi-step transport dynamic into a parameterized function $f_\psi$. Consequently, the explicit "round count" collapses into the complexity of the learned mapping, where the network learns to approximate the cumulative effect of the transport.

- **Agreement:** The "agreement" is mathematically defined as the system reaching the fixed point where the **Stein Discrepancy** approaches zero, i.e., $\mathbb{E}_{u\sim q}[\mathcal{A}_p\phi(u)] \approx 0$, where $\mathcal{A}_p$ is the Stein operator. This implies that the empirical measure of the agents' joint policy matches the optimal Boltzmann distribution.

### L.2  VISUALIZING THE CONVERGENCE OF MEASURE

The dynamics of negotiation are explicitly visualized as the **evolution of the joint policy's support** in our experimental results.

**Evolution of Support (Figure 5):** Figure 5 illustrates the transport of the joint action measure over training steps. Initially (steps 1-1500), the probability mass is distributed over sub-optimal modes (local Nash Equilibria). As the amortized policy $f_\psi$ minimizes the KL-divergence, we observe the **concentration of measure** shifting from the local optimum to the global optimum (steps 1500-3000). This trajectory visually represents the "negotiation" resolving the Perceived Relative Over-generalization (PRO) by reshaping the energy landscape of the policy and transporting particles to the high-probability regions of the target distribution.

**Topological Comparison (Figure 7):** By comparing SVNR with SVNR-M (no negotiation) in Figure 7, we isolate the effect of the conditional dependency structure (the nested sets $C_i$). Figure 7(a) versus Figure 7(c) demonstrates that without the Stein transport (negotiation), the joint distribution remains trapped in a sub-optimal mode. The "negotiation" is interpretable as the **correction vector** applied to the joint distribution that aligns the agents' conditional policies, ensuring the joint support covers the global optimum.

## M  LIMITATIONS AND FUTURE WORK

This section discusses the limitations and outlines directions for future research.

**Computational Overhead and Scalability.** The communication complexity of our negotiation process during training is $O(N)$, where $N$ is the number of agents—comparable to standard centralized training methods. While this does not affect execution efficiency (as no communication is required during testing), scaling to environments with many agents or high-dimensional state/action spaces may require balancing RO-free guarantees with computational efficiency. Our current implementation uses automated hyperparameter tuning via TPE Sampler to optimize learning rates, entropy coefficients, and particle numbers, providing reliable default configurations across various settings.

**Theoretical Assumptions.** Our framework assumes nested negotiation/communication during training, consistent with the Centralized Training with Decentralized Execution (CTDE) paradigm widely used in MARL. This allows agents to leverage global information for improved coordination during training while maintaining fully decentralized, communication-free execution. Other assumptions (e.g., stationarity, bounded rewards) are standard in MARL literature and necessary for theoretical rigor without imposing impractical constraints.

**Environmental Complexity.** Our validation focuses on standard benchmark environments with sufficient complexity to verify our theoretical claims while maintaining tractability. Extending our approach to more complex, high-dimensional domains represents an important future direction, which will likely require additional architectural innovations to preserve our RO-free guarantees while maintaining computational efficiency.

**Partial Observability.** The current implementation leverages the Centralized Training with Decentralized Execution (CTDE) paradigm to address partial observability. As demonstrated in the MaMuJoCo experiments, our method effectively projects global guidance onto local policies during training. However, explicitly incorporating recurrent architectures (e.g., Transformers or LSTMs) to better encode long-horizon sequential observations within the negotiation policies remains a promising direction for handling complex POMDPs with severe memory dependencies.

While addressing these limitations is beyond the scope of this paper, they represent valuable avenues for future research that could significantly broaden the applicability of our RO-free MARL approach.

## M.1 Extended Theoretical Analysis on Partial Observability

In this section, we provide a deeper theoretical analysis regarding the applicability of Stein Variational Negotiated Reasoning (SVNR) to Partially Observable Stochastic Games (POSGs) and the feasibility of fully decentralized training.

### M.1.1 SVNR in Partially Observable Stochastic Games

While the main text formulates the problem using global states $s$ for clarity, SVNR naturally extends to POSGs through the lens of *projected variational inference*. In a POSG, agent $i$ observes a local history $\tau_i \in \mathcal{T}_i$, while the global state $s$ (or joint history $\boldsymbol{\tau}$) is available only during centralized training.

The objective of Maximum Entropy MARL in this setting is to learn a joint policy $\pi(\mathbf{u}|\boldsymbol{\tau})$ that minimizes the KL-divergence with the energy-based optimal policy induced by the global Q-function $Q(\boldsymbol{\tau}, \mathbf{u})$:

$$\min_{\pi} D_{\mathrm{KL}} \left( \pi(\mathbf{u}|\boldsymbol{\tau}) \, \| \, \frac{1}{Z} \exp \left( \frac{1}{\alpha} Q(\boldsymbol{\tau}, \mathbf{u}) \right) \right). \tag{24}$$

In SVNR, the negotiation policy is parameterized by amortized neural networks $f_{\psi_i}(u_i|\tau_i, \xi_i, \xi_{C_i})$ which condition only on local information $\tau_i$. The update rule in our Amortized MPSVGD (Equation 9 and 10) performs a **projection** of the global gradient onto the local parameter space. The gradient for the local policy parameters $\psi_i$ is:

$$\frac{\partial J}{\partial \psi_i} \propto \mathbb{E}_{\boldsymbol{\tau}, \xi} \left[ \Delta f_i^{\psi}(\xi; \boldsymbol{\tau}) \cdot \frac{\partial f_i^{\psi}(\xi; \tau_i)}{\partial \psi_i} \right]. \tag{25}$$

Here, $\Delta f_i^{\psi}(\xi; \boldsymbol{\tau})$ is the Stein gradient computed using the *global* critic (full observability), representing the optimal direction in the functional space. The term $\frac{\partial f_i^{\psi}(\xi; \tau_i)}{\partial \psi_i}$ is the Jacobian of the local policy given *local* history.

This update effectively solves the following projection problem:

$$\psi_i^* = \arg\min_{\psi_i} \mathbb{E}_{\boldsymbol{\tau}} \left[ D_{\mathrm{KL}} \left( q_{\mathrm{global}}(\cdot|\boldsymbol{\tau}) \parallel \pi_{\psi_i}(\cdot|\tau_i) \right) \right]. \tag{26}$$

By updating $\psi_i$ via the chain rule, the agent learns a local policy $\pi_{\psi_i}(\cdot|\tau_i)$ that is the best possible approximation (in terms of KL-divergence) of the globally optimal negotiated outcome, conditioned on its limited view $\tau_i$. This theoretical formulation explains the strong empirical performance of SVNR on partially observed benchmarks like MaMuJoCo (Table 1).

### M.1.2 Feasibility of Fully Decentralized Training

Although our implementation utilizes a centralized critic $Q(\mathbf{u}, s)$ for sample efficiency, the SVNR framework is theoretically compatible with fully decentralized training, provided the global utility function admits a factorizable structure.

Consider a scenario where the global Q-function decomposes according to a factor graph (e.g., a pairwise Markov Random Field) consistent with the agent topology:

$$Q_{\mathrm{total}}(\mathbf{u}, s) = \sum_{c \in \mathcal{C}} Q_c(\mathbf{u}_c, s_c), \tag{27}$$

where $c$ represents a local clique of agents (e.g., neighbors) and $Q_c$ is a local utility function. The core component of our method, the Stein variational update direction for agent $i$, is given by:

$$\phi_i^*(\mathbf{u}) = \mathbb{E}_{\mathbf{u} \sim q} \left[ k_i(\mathbf{u}, \cdot) \nabla_{u_i} Q_{\mathrm{total}}(\mathbf{u}, s) + \nabla_{u_i} k_i(\mathbf{u}, \cdot) \right]. \tag{28}$$

Due to the linearity of the gradient operator, the score function term decomposes locally:

$$\nabla_{u_i} Q_{\mathrm{total}}(\mathbf{u}, s) = \sum_{c:i \in c} \nabla_{u_i} Q_c(\mathbf{u}_c, s_c). \tag{29}$$

This implies that agent $i$ does not need to query a global critic. Instead, it only requires the gradients of the local utility functions from the cliques it belongs to. If we employ a decomposable kernel $k(\mathbf{u}, \mathbf{u}') = \prod_j k_j(u_j, u_j')$, the expectation term also factorizes.

Consequently, Algorithm 1 can be reformulated as a **Distributed Stein Variational Gradient Descent (DSVGD)** algorithm. In this variant, the "negotiation" during training occurs via gradient message passing between neighbors rather than querying a central oracle, extending the applicability of SVNR to scenarios where centralized training is not feasible.

