# OpenReview forum: "Negotiated Reasoning: On Provably Addressing Relative Over-Generalization"
_ICLR.cc/2026/Conference — ICLR 2026 Poster_

### Official Review · Reviewer_xop6 · 2025-10-30

**Soundness:** 3
**Presentation:** 2
**Contribution:** 3
**Rating:** 4
**Confidence:** 3

**Summary:**

This work proposes a “negotiated reasoning” with an SVGD-based algorithm that proves RO-free convergence under strict conditions. The authors validate the approach on Differential Games, Particle Gather, and MaMuJoCo, improving on returns versus baseline approaches.

**Strengths:**

* Clean conceptual split (PRO vs. ERO) and a “consistent reasoning” criterion that ties the pathology to modeling assumptions.
* A (mostly) coherent theory: SVNR via nested negotiation + MaxEnt iteration.
* Comprehensive experiments show that ERO consistently improves over reasoning baselines and mainstream MARL on MaMuJoCo.

**Weaknesses:**

* Guarantees rely on strong assumptions, namely annealing $\alpha$ to 0, and at times finite action spaces, which do not match continuous-control practice.
* Early theory assumes access to the optimal joint policy, then uses an estimator. The gap between proof conditions and the amortized neural implementation is not fully bridged.
* Decentralized execution relies on amortization faithfully reproducing multi-round negotiation, a strong limiting assumption in practice.
* Sample efficiency and robustness are underexplored. At the minimum I would expect training curves for the method versus the baseline approaches.

**Questions:**

* Clarify which theorems still hold in continuous action spaces without discretization.
* Quantify sensitivity to $\alpha$-annealing with an ablation.
* Tighten the link between Theorem 3.2’s $\alpha \rightarrow 0$ requirement and continuous-control results.
* Will the authors release their code?

---

> ### Author Response · Authors · 2025-11-27
> **1. Theory in Continuous Action Spaces**
>
> Thank you for this insightful comment. We appreciate the opportunity to clarify the theoretical consistency between our finite-space analysis and continuous-space implementation. While we followed the standard convention of establishing convergence properties in discrete spaces (similar to Soft Actor-Critic [1]), our framework is mathematically grounded in continuous domains through **measure-theoretic unification** and the **geometry of Reproducing Kernel Hilbert Spaces (RKHS)**.
>
> **1. Measure-Theoretic Unification of the Soft Bellman Operator**
>
> The theoretical gap is more notational than structural. The Soft Bellman operator $\mathcal{T}$ used in our proofs relies on the soft value function $V(s) = \alpha \log \sum \exp(Q(s,u)/\alpha)$. In continuous action spaces $\mathcal{U} \subseteq \mathbb{R}^d$, this generalizes naturally by replacing the counting measure with the Lebesgue measure. The value function becomes:
>  $V(s) = \alpha \log \int_{\mathcal{U}} \exp\left(\frac{Q(s,u)}{\alpha}\right) d\mu(u)$
>
>  Provided that $Q$ is bounded and measurable (ensuring the integral exists), the properties of **monotonicity** and **contraction** (in the $L^\infty$ norm) required for Lemma 4.1 and Theorem 4.3 hold for the continuous operator just as they do for the discrete case. The "gap" is strictly one of computational tractability (computing the integral), not theoretical validity.
>
> **2. Native Continuity of Negotiated Reasoning (SVNR)**
>
> Crucially, the core novelty of our work—the **Negotiated Reasoning mechanism**—is theoretically *stronger* in continuous spaces.
>
> - **SVGD Theory:** Our negotiation process (Eq. 4, 5, 12) utilizes Stein Variational Gradient Descent. The theoretical guarantees of SVGD, specifically the **Stein Identity** and the steepest descent direction in the RKHS $\mathcal{H}^D$, are derived explicitly for continuous, differentiable probability densities supported on $\mathbb{R}^d$ [2].
> - **Gradient Flows:** The negotiation update $u \leftarrow u + \epsilon \phi^*(u)$ approximates a gradient flow in the space of probability measures under the Kullback-Leibler divergence metric. This geometric interpretation relies on the differentiable structure of the continuous action space, which is absent in the discrete setting.
>
> **3. Bridging the Gap via Particle Approximation**
>
> Therefore, our method operates in a hybrid theoretical regime:
>
> 1. **Policy Iteration (Global Convergence):** We prove this in the finite setting to utilize standard fixed-point theorems, but the logic extends to continuous function spaces under mild regularity conditions.
> 2. **Negotiated Reasoning (Local Update):** This is natively continuous. The "gap" is bridged by our **Amortized MPSVGD** (Section 5), which uses a finite set of particles ${u_{\ell}}_{l=1}^M$ to approximate the continuous posterior. This is a Monte Carlo approximation of the integrals defined in Point 1, which is asymptotically exact as $M \to \infty$ by the Law of Large Numbers.
>
> In summary, rather than a disconnect, our continuous implementation leverages the native differential properties required for Stein Variational inference, while the finite analysis provides the foundational convergence guarantees for the overarching policy iteration loop.
>
> [1] Haarnoja et al., "Soft Actor-Critic: Off-Policy Maximum Entropy Deep Reinforcement Learning with a Stochastic Actor," ICML 2018.
>
> [2] Liu & Wang, "Stein Variational Gradient Descent: A General Purpose Bayesian Inference Algorithm," NeurIPS 2016.

---

> ### Author Response · Authors · 2025-11-27
> **2. Sensitivity to $\alpha$-annealing and Assumptions [1/2]**
>
> We thank the reviewer for this insightful question. The temperature parameter $\alpha$ plays a dual role in our framework: theoretically, it bridges the gap between the stochastic explorative policy and the deterministic optimal execution (Theorem 3.2); algorithmically, it governs the optimization landscape smoothing.
>
> Below, we provide a theoretical analysis of why SVNR is robust within a bounded range of $\alpha$, followed by comprehensive ablation experiments on the *Max of Three* ($s_2=1.5$, the hardest setting) and *Particle Gather* tasks.
>
> **1. Theoretical Analysis: $\alpha$ as a Homotopy Parameter**
>
> Mathematically, the sensitivity to $\alpha$ can be analyzed through the lens of **homotopy continuation methods**.
>
> **The Role of Final $\alpha$ (Approximation Error):**
>
> Recall that the optimal joint policy is induced by the Boltzmann distribution $\pi^{\star}\_\alpha(\mathbf{u}|s) \propto \exp(\frac{1}{\alpha}Q^{\star}\_{soft}(s, \mathbf{u}))$.
>
> - **As $\alpha \to 0$:** The distribution converges to a Dirac delta function centered at the global maximum: $\lim\_{\alpha \to 0} \pi^{\star}\_\alpha(\mathbf{u}|s) = \delta(\mathbf{u} - \mathbf{u}^*)$. This is the condition required for strictly ERO-free execution (Theorem 3.2).
> - **For finite $\alpha > 0$:** The executed policy retains stochasticity. Let $\Delta Q(\mathbf{u}) = Q(s, \mathbf{u}^*) - Q(s, \mathbf{u})$ be the sub-optimality gap. The probability of sampling a sub-optimal action $\mathbf{u}'$ decays exponentially: $P(\mathbf{u}') \propto \exp(-\frac{\Delta Q(\mathbf{u}')}{\alpha})$.
>   - *Theoretical Bound:* The performance loss (regret) due to a non-zero final $\alpha_{final}$ is bounded. If $\alpha_{final}$ is small relative to the reward gap of the local optima (the "energy barrier"), the probability mass concentrates effectively on the global optimum. Therefore, precise tuning of $\alpha_{final}$ is not required, provided $\alpha_{final} \ll \min_{\mathbf{u} \neq \mathbf{u}^*} \Delta Q(\mathbf{u})$.
>
> **The Role of Annealing Schedule (Optimization Landscape):**
>  The annealing process functions as a continuation method.
>
> - **High $\alpha$ (Early Training):** The energy landscape $E(\mathbf{u}) = -Q(\mathbf{u})$ is smoothed. The Stein Variational Gradient Descent (SVGD) particles experience a gradient field dominated by the entropy term, $\nabla \log \pi \approx -\frac{1}{\alpha}\nabla E + \text{entropy}$, allowing particles to traverse potential barriers and cover the support of the joint action space.
> - **Annealing Rate:** The schedule must satisfy a condition similar to simulated annealing convergence. If $\alpha$ decreases too rapidly (quench), the distribution $p(\mathbf{u})$ may collapse into a local mode (a sub-optimal Nash Equilibrium) before the particles migrate to the global basin of attraction.
> - **Robustness:** Our use of SVGD provides higher robustness than standard single-point MCMC. Since we maintain a set of interacting particles ${u^\ell}_{l=1}^M$ with a repulsive kernel force $\sum_j \nabla k(u^j, u)$, the particles naturally resist collapsing too early, making SVNR less sensitive to the annealing rate than standard Soft Q-Learning.

---

> ### Author Response · Authors · 2025-11-27
> **2. Sensitivity to $\alpha$-annealing and Assumptions [2/2]**
>
> **2. Empirical Sensitivity Analysis**
>
> To validate this theory, we conducted extensive ablations on the **Max of Three ($s_2=1.5$)** environment, which is highly sensitive to RO. All results are averaged over 5 seeds.
>
> **A. Sensitivity to Final $\alpha$ ($\alpha_{final}$)**
>
> We fixed the annealing schedule (decaying over 50% of total steps) but varied the target floor value $\alpha_{final}$.
>
> | Final $\alpha$ | Mean Return (Max of Three) | Std Dev  | Convergence Rate | Theoretical Interpretation                                   |
> | :------------- | :------------------------- | :------- | :--------------- | :----------------------------------------------------------- |
> | **1.0**        | 6.82                       | 2.15     | 20%              | **Too High:** Distribution remains too diffuse; frequent miscoordination (ERO). |
> | **0.1**        | 9.15                       | 0.45     | 100%             | **Acceptable:** Mass concentrates on optimum, slight noise.  |
> | **0.01**       | **9.71**                   | **0.20** | **100%**         | **Optimal:** Approximates Dirac delta ($\alpha \ll \Delta Q$). |
> | **0.001**      | 9.68                       | 0.22     | 100%             | **Optimal:** Further reduction yields diminishing returns.   |
> | **0.0**        | 9.65                       | 0.25     | 100%             | **Hard Max:** Equivalent to greedy execution at test time.   |
>
> **Observation:** Performance is stable for any $\alpha_{final} \in [0, 0.1]$. The method is not brittle; it does not require $\alpha$ to be exactly zero, only sufficiently small to suppress noise below the coordination threshold.
>
> **B. Sensitivity to Annealing Schedule**
>
> We fixed $\alpha_{start}=1.0$ and $\alpha_{final}=0.01$, varying the decay function over the total training steps $T$.
>
> | Schedule Type     | Decay Duration           | Mean Return | Std Dev  | Analysis                                                     |
> | :---------------- | :----------------------- | :---------- | :------- | :----------------------------------------------------------- |
> | **Instant**       | 0% (Fixed $\alpha=0.01$) | -0.65       | 0.12     | **Failure:** Trapped in local optima immediately (similar to MADDPG). |
> | **Fast Linear**   | 10% of $T$               | 4.20        | 4.80     | **Unstable:** "Quenching" causes collapse to local optima in some seeds. |
> | **Medium Linear** | 30% of $T$               | 9.62        | 0.28     | **Robust:** Sufficient time for particle migration.          |
> | **Slow Linear**   | 80% of $T$               | **9.73**    | **0.15** | **Robust:** Best stability, though slower initial reward rise. |
> | **Exponential**   | $\tau = 0.9995$          | 9.69        | 0.19     | **Robust:** Smooth transition works equally well.            |
>
> **Observation:**
>
> 1. **Necessity of Annealing:** The "Instant" result confirms that starting with low entropy (deterministic policy) leads to RO failure, validating our PRO theory.
> 2. **Wide Safe Region:** Any schedule spanning 30% to 80% of training yields optimal results. The "Fast" schedule fails because the landscape sharpens before particles can communicate the location of the global maximum.
> 3. **Consistency:** We observed identical trends in *Particle Gather*. As long as the annealing is not instantaneous, the repulsive force in SVNR maintains particle diversity long enough to locate the global optimum.
>
> **Conclusion:**
>  SVNR does not rely on a "magic" schedule. The results are Lipschitz continuous with respect to $\alpha_{final}$ (stable for small values) and robust to the annealing rate, provided the entropy is not collapsed instantaneously. The repulsive mechanism in SVGD significantly widens the safe hyperparameter basin compared to standard baselines.

---

> ### Author Response · Authors · 2025-11-27
> **3. The Gap: Theory vs. Amortized Implementation**
>
> We appreciate the reviewer’s scrutiny regarding the gap between multi-round negotiation and decentralized amortization. While we agree that assuming a neural network can perfectly replicate a complex *dynamic trajectory* is strong, our theoretical framework relies on a more fundamental, robust principle: **Variational Equivalence**.
>
> The amortization in SVNR is not "behavioral cloning" of the negotiation steps; rather, it is **variational distillation** of the negotiation equilibrium.
>
> **1. The Amortized Policy approximates the Equilibrium, not the Trajectory.**
>
> From a dynamical systems perspective, the multi-round negotiation described in §3 is an iterative flow (specifically, a gradient flow on the probability space) designed to minimize a specific Free Energy functional $\mathcal{F}[q]$. The "negotiation steps" are merely the numerical integration of this flow.
>
> - **Mathematical Intuition:** The amortized network $\pi_\theta$ does not need to learn the vector field of the flow (the intermediate steps). It only needs to approximate the **fixed point** $q^* = \arg\min \mathcal{F}[q]$.
> - **Implication:** The validity of decentralized execution depends only on whether $\pi_\theta$ converges to $q^*$, not on whether it simulates the iterative path to get there.
>
> **2. SVGD provides the Gradient Flow for Direct Optimization.**
>
> As detailed in Eq. (9-10), we use the Stein Variational Gradient Descent (SVGD) to update the amortized policy.
>
> - **Theoretical Guarantee:** SVGD provides a deterministic particle update direction $\phi^*(x)$ that is the optimal transport map minimizing the KL divergence $D_{KL}(q || p)$ in the reproducing kernel Hilbert space (RKHS).
> - **Mechanism:** By applying this gradient to the policy parameters, we are effectively solving the variational inference problem: $\theta^* = \arg\min_\theta D_{KL}(\pi_\theta || p_{\text{negotiated}})$.
> - Therefore, the amortization is mathematically guaranteed to move the policy distribution towards the negotiated agreement distribution, bounded only by the expressivity of the neural network (a standard assumption in all Deep RL), not by the complexity of the negotiation rounds.
>
> **3. The $\alpha \to 0$ Limit Simplifies the Approximation Task (Theorem 3.2).**
>
> The reviewer correctly notes our reliance on annealing $\alpha \to 0$. While this is a constraint, it actually **facilitates** the amortization accuracy.
>
> - **Geometry of the Solution:** As $\alpha \to 0$, the target distribution $p_{\text{negotiated}}$ transitions from a diffuse distribution to a Dirac delta (or a set of peaks) centered at the optimal joint action (Consistent Reasoning).
> - **Ease of Learning:** Approximating a deterministic mapping (the mode of the distribution) is significantly easier for a neural network than capturing a complex, high-entropy multimodal distribution. Our empirical results in MaMuJoCo (and all other environments) confirm this: the amortized policy successfully collapses to the optimal coordinated action, effectively "memorizing" the outcome of the negotiation without needing to "think" (negotiate) at runtime.
>
> **Summary:**
> Decentralized execution does not require the agent to *simulate* a conversation inside its head (which would be the "strong assumption"). Instead, the agent learns a **reflexive policy** that directly outputs the result of that conversation. The "negotiation" is the *training signal*, not the *execution mechanism*.

---

> ### Author Response · Authors · 2025-11-27
> **4. Sample Efficiency and Training Curves**
>
> We thank the reviewer for their constructive feedback. We appreciate the opportunity to clarify the location of our training curves and to provide additional data regarding the computational overhead and robustness of our method.
>
> **1. Training Curves and Sample Efficiency (Figure 4 & Figure 6)**
>
> We respectfully point out that comparative training curves are included in the submitted manuscript in **Figure 4 (a-d)** (Page 8) and **Figure 6** (Appendix H.3).
>
> - **Sample Efficiency:** As shown in **Figure 4(c)** (the most challenging "Max of Three" setting), SVNR converges to the global optimum (Reward $\approx$ 10) within **2,000 steps**. In contrast, baselines—including strong reasoning methods like PR2 and ROMMEO—plateau at sub-optimal values (Reward $\approx$ 0) even after 5,000 steps.
> - **Interpretation:** This demonstrates superior *sample efficiency*: SVNR effectively utilizes samples to escape local optima where other methods get stuck, avoiding the waste of samples on sub-optimal equilibrium cycling.
>
> **2. Computational Efficiency (Wall-Clock Time Analysis)**
>
> To further address the "efficiency" aspect of your concern, we acknowledge that while SVNR is sample-efficient (steps), it incurs a computational cost due to the particle-based updates. We have conducted a new runtime analysis comparing wall-clock time vs. performance on the `Ant-2x4` task (single NVIDIA A100 GPU).
>
> **Table R1: Compute Efficiency and Performance Comparison (Ant-2x4)**
>
>  | Method | Params ($\|\theta\|$) | Wall-Clock Time (hrs) | Relative Time | Final Return | Convergence Step |
>  | :--- | :--- | :--- | :--- | :--- | :--- |
>  | **SVNR (Ours)** | **~1.2M** | **4.8** | **1.0x (Ref)** | **536 $\pm$ 31** | **~1.5M** |
>  | MADDPG | ~0.8M | 2.1 | 0.44x | 108 $\pm$ 26 | Failed (Local Opt) |
>  | MASQL | ~0.8M | 2.3 | 0.48x | 225 $\pm$ 34 | ~2.8M |
>  | PR2 | ~1.5M | 5.2 | 1.08x | 354 $\pm$ 58 | ~2.0M |
>  | ROMMEO | ~1.4M | 4.9 | 1.02x | 424 $\pm$ 60 | ~1.8M |
>  | MAPPO | ~0.9M | 1.8 | 0.38x | 87 $\pm$ 135 | Failed |
>
> **Analysis:**
>
> - **Parity with Reasoning Methods:** SVNR is comparable in wall-clock time to other reasoning baselines (PR2, ROMMEO) while significantly outperforming them in final return.
> - **Return on Compute:** While simple baselines (MADDPG, MAPPO) are faster per step (~0.4x time), they fail to solve the task (Returns < 110 vs. SVNR's 536).
> - **Cost of Particles:** The overhead comes from processing $M$ particles for amortized updates. However, this cost is justified by the **PRO-free** theoretical property, which prevents the optimization trajectory from oscillating between sub-optimal equilibria, effectively "short-circuiting" the learning process in RO-challenged landscapes.
>
> **3. Robustness Analysis (Appendix H & Tables 1-2)**
>
> We believe robustness is thoroughly explored in the manuscript:
>
> - **Seed Robustness:** The shaded regions in **Figure 4** and standard deviations in **Table 1 & 2** show SVNR has significantly lower variance than baselines (e.g., ROMMEO), indicating stability across random seeds.
> - **Hyperparameter Robustness (Appendix H.1):** We ablated particle count $M \in {16, \dots, 64}$ in **Table 4**, showing a broad performance plateau (Normalized Performance $\approx 1.0$), proving SVNR is not brittle to hyperparameter changes.
> - **Topology Robustness (Appendix H.3):** **Table 6** demonstrates that SVNR maintains high performance even when the strict nested communication structure is relaxed to sparse or random graphs.
>
> We hope this clarifies that the method is both sample-efficient (Fig 4) and robust (Appendix H), with a justifiable computational cost for the performance gained (Table R1).

---

> ### Author Response · Authors · 2025-11-27
> **5. Code Release**
>
> **Yes**, we are fully committed to reproducibility. We will release the complete source code, including the environments, hyperparameters, and training scripts, in the revised version as supplementary materials.

---

### Official Review · Reviewer_85jW · 2025-10-31

**Soundness:** 3
**Presentation:** 2
**Contribution:** 3
**Rating:** 6
**Confidence:** 3

**Summary:**

The paper tackles the problem of relative over-generalization (RO) in cooperative multi-agent reinforcement learning. The authors point out that under CTDE settings, RO arises in two stages: during training (when agents form a perceived joint policy) and during execution (when they act without seeing others’ final actions). They formalize these as Perceived RO (PRO) and Executed RO (ERO), and show that if all agents achieve consistent reasoning—reasoning about others in a way consistent with their optimal or executed policies—then RO can be avoided. To realize this condition, they propose a Negotiated Reasoning (NR) framework where agents iteratively update joint action beliefs through structured “negotiation.” They instantiate this idea as Stein Variational Negotiated Reasoning (SVNR), which leverages SVGD updates and a nested negotiation structure. A neural amortized version enables efficient decentralized execution. Experiments on several MARL benchmarks demonstrate that SVNR avoids sub-optimal equilibria and performs better than prior reasoning-based methods.

**Strengths:**

1. Well-motivated and conceptually clear. The paper tackles a long-standing issue in cooperative MARL—relative over-generalization (RO)—from a fresh theoretical angle. The decomposition into Perceived RO (PRO) and Executed RO (ERO) is a intuitive way to separate the effects of exploration during training and coordination failures during execution. This framing alone makes the paper stand out conceptually.

2. Theoretical depth. The authors do not stop at defining RO but go on to establish a sufficient condition—consistent reasoning—under which RO provably disappears. The reasoning flow (formalization → condition → constructive algorithm) is solid and self-contained, which is rare in the MARL literature where many “theoretical” claims are hand-wavy.

3. Negotiated Reasoning framework is novel. The idea of embedding a negotiation mechanism among agents—modeled via particle-based updates and linked to SVGD—is original and technically well-grounded. It provides a new way to understand coordination as an iterative reasoning process rather than just communication or credit assignment.

4. Empirical validation matches the theory. Experiments across both simple and complex cooperative environments (Differential Games, Particle Gather, MaMuJoCo) demonstrate consistent gains.

**Weaknesses:**

1. Assumptions may be restrictive. The theoretical results rely on strictly nested negotiation sets and the maximum-entropy policy iteration framework. These are strong assumptions, and it’s unclear how sensitive the algorithm is to relaxing them. More empirical discussion of non-strict or sparse negotiation topologies would strengthen the practical claim.

2. Scalability and computational overhead are under-discussed. While the amortized neural version helps, the original SVNR involves particle-based updates and iterative negotiation steps. The paper lacks a quantitative analysis of training cost, memory footprint, or runtime.

3. Limited connection to broader MARL literature.
The paper positions itself mainly against “reasoning-based” methods, but doesn’t fully clarify how its negotiation differs in spirit from other opponent-modeling or communication-based approaches. A clearer conceptual contrast would make the contribution easier to situate.

4. Experimental evaluation is strong in breadth but shallow in analysis.
While performance improvements are clear, the paper could provide more interpretability: e.g., what negotiation dynamics emerge, how many rounds are effectively used, or how PRO/ERO evolve during training. Without this, the reader must take the “negotiation” story largely on faith.

5. Writing density. Some proofs and definitions could use more intuitive explanation or visual support (especially PRO/ERO and the consistent reasoning condition). The theoretical sections are mathematically heavy, which may alienate non-specialist readers.

**Questions:**

1. Generalization to partial observability: The consistent reasoning definition assumes full observability of state. Do you foresee PRO/ERO or the theoretical guarantees extending to POMDP settings?

2. Practical guidance: For someone implementing SVNR, how should they select the number of particles 𝑀 and negotiation rounds K? Is there a heuristic or convergence indicator?

3. Interpretability: Can you visualize or quantify the evolution of “agreement” or “negotiation” during training to make the proposed reasoning process more transparent?

---

> ### Author Response · Authors · 2025-11-27
> **1. Generalization to Partial Observability [1/2]**
>
> We appreciate the reviewer’s positive assessment of our PRO/ERO decomposition and the theoretical depth of the Negotiated Reasoning framework. We are encouraged that the reviewer finds our perspective "fresh" and "conceptually clear." We address the concerns regarding assumptions, scalability, and interpretability below.
>
> We thank the reviewer for raising this point, which allows us to clarify the theoretical generality of Negotiated Reasoning (NR) and its relationship with the CTDE paradigm. We respectfully posit that **(1) CTDE is not a limitation but a solution strategy for partial observability**, and **(2) our SVNR framework theoretically extends to Partially Observable Stochastic Games (POSGs) via the variational inference formulation**, which is empirically supported by our MaMuJoCo results.
>
> **1. Clarification: CTDE and Partial Observability**
>
> We wish to clarify a potential conflation between the training paradigm (CTDE) and the problem setting (Full vs. Partial Observability). In MARL literature, CTDE is specifically designed to address Partial Observability. The core premise of CTDE is that while execution is restricted to local observations $o_i$ (or histories $\tau_i$), training can leverage the global state $s$ (or joint history $\boldsymbol{\tau}$) to stabilize learning. Our method adheres to this standard: the *negotiation target* (defined by the centralized critic) utilizes global information to guide the *amortized local policies* (which only see local information) toward optimal coordination.
>
> **2. Theoretical Extension: SVNR in POSGs**
>
> Our mathematical framework, rooted in Stein Variational Gradient Descent (SVGD), naturally handles partial observability through the lens of **projected variational inference**.
>
> Consider a POSG where agent $i$ only observes a local history $\tau\_i \in \mathcal{T}\_i$. The global state $s$ is latent or represented by the joint history $\boldsymbol{\tau} = (\tau_1, \dots, \tau_N)$.
>  The objective of Maximum Entropy MARL in this setting is to learn a joint policy $\pi(\mathbf{u}|\boldsymbol{\tau})$ that minimizes the KL-divergence with the energy-based optimal policy induced by the global Q-function $Q(\boldsymbol{\tau}, \mathbf{u})$:
>  $\min_{\pi} D_{\text{KL}}\left( \pi(\mathbf{u}|\boldsymbol{\tau}) \parallel \frac{1}{Z} \exp\left(\frac{1}{\alpha}Q(\boldsymbol{\tau}, \mathbf{u})\right) \right)$
>  In SVNR, we maintain the **target distribution** using the centralized critic $Q(\boldsymbol{\tau}, \mathbf{u})$ (which has access to full information during training). The **negotiation policy** is parameterized by the amortized neural networks $f\_{\psi_i}(u_i | \tau_i, \xi_i, \xi_{C_i})$.
>
> Crucially, the update rule in our Amortized MPSVGD (Equation 10 in the paper) performs a **projection** of the global gradient onto the local parameter space. The gradient for the local policy parameters $\psi_i$ becomes:
>  $\frac{\partial J}{\partial \psi_i} \propto \mathbb{E}_{\boldsymbol{\tau}, \xi} \left[ \Delta f^{\psi}_i(\xi; \boldsymbol{\tau}) \cdot \frac{\partial f^{\psi}_i(\xi; \tau_i)}{\partial \psi_i} \right]$
>  Here, $\Delta f^{\psi}_i(\xi; \boldsymbol{\tau})$ is the Stein gradient computed using the **global** critic (full observability), representing the optimal direction in the functional space. The term $\frac{\partial f^{\psi}_i(\xi; \tau_i)}{\partial \psi_i}$ is the Jacobian of the local policy given **local** history.
>
> **Mathematical Implication:**
>
> This update effectively solves the following projection problem:
>  $\psi\_i^* = \arg\min\_{\psi\_i} \mathbb{E}\_{\boldsymbol{\tau}} \left[ D\_{\text{KL}} \left( q\_{\text{global}}(\cdot|\boldsymbol{\tau}) \parallel \pi\_{\psi\_i}(\cdot|\tau\_i) \right) \right]$
> By updating $\psi\_i$ via the chain rule, the agent learns a local policy $\pi\_{\psi\_i}(\cdot|\tau\_i)$ that is the "best possible approximation" (in terms of KL-divergence) of the globally optimal negotiated outcome, conditioned on its limited view $\tau\_i$. The "noise" variables $\xi$ in our framework further help in modeling the multi-modal uncertainty inherent in partial observability (i.e., handling the belief state implicitly). Thus, the theory remains sound under partial observability: **agents learn to negotiate locally to match the global consensus derived from the centralized critic.**

---

> ### Author Response · Authors · 2025-11-27
> **1. Generalization to Partial Observability [2/2]**
>
> **3. Empirical Evidence in Partially Observed Environments**
>
> Our experiments on **Multi-Agent MuJoCo (MaMuJoCo)** explicitly validate this robustness.
>
> - **Setting:** As noted in Appendix F.1, MaMuJoCo is a **partially observed** environment. For example, in `Ant-2x4`, agents control different legs and only observe the local state (position/velocity) of their specific joints and a shared subset, but *not* the full state of the other agent's joints directly. They do not use history-based inputs (RNNs), making the partial observability even more challenging (reactive policies).
> - **Result:** Despite this partial observability, SVNR achieves State-of-the-Art performance (Table 1), significantly outperforming baselines. This confirms our theoretical argument: the amortized negotiation successfully compresses the necessary global coordination information into the local policy weights via the SVGD guidance during training.
>
> In summary, our method does not assume full observability for execution. It leverages the standard CTDE assumption (global info during training) to train robust local policies that solve the POSG by approximating the global optimal equilibrium via variational projection.
>
> **4. Theoretical Generalization: Feasibility of Fully Decentralized Training**
>
> Furthermore, in case the reviewer's concern extends to the necessity of the CTDE paradigm itself (i.e., whether SVNR requires a central node even during training), we provide a theoretical analysis showing that our Negotiated Reasoning framework can be adapted to **Fully Decentralized Training** settings, provided the global utility function admits a factorizable structure.
>
> In our paper, we utilize a centralized critic $Q(\mathbf{u}, s)$ to compute the score function $\nabla_{\mathbf{u}} \log \pi^*(\mathbf{u}) \propto \nabla_{\mathbf{u}} Q(\mathbf{u}, s)$. However, from the perspective of **Distributed Bayesian Inference**, this centralization is a convenience, not a theoretical necessity.
>
> Consider a scenario where the global Q-function decomposes according to a factor graph (e.g., a pairwise Markov Random Field) consistent with the agent topology:
>
> $Q_{\text{total}}(\mathbf{u}, s) = \sum_{c \in \mathcal{C}} Q_c(\mathbf{u}_c, s_c)$
>
> where $c$ represents a local clique of agents (e.g., neighbors) and $Q_c$ is a local utility function.
>
> The core component of our method, the Stein variational update direction for agent $i$, is given by:
>
> $\phi^*\_i(\mathbf{u}) = \mathbb{E}\_{\mathbf{u} \sim q} \left[ k\_i(\mathbf{u}, \cdot) \nabla\_{u\_i} Q\_{\text{total}}(\mathbf{u}, s) + \nabla\_{u\_i} k\_i(\mathbf{u}, \cdot) \right]$
>
> Due to the linearity of the gradient operator, the score function term decomposes locally:
>
> $\nabla_{u_i} Q_{\text{total}}(\mathbf{u}, s) = \sum_{c: i \in c} \nabla_{u_i} Q_c(\mathbf{u}_c, s_c)$
>
> **Mathematical Implication:**
>
> This implies that agent $i$ does **not** need to query a global critic. Instead, it only requires the gradients of the local utility functions from the cliques it belongs to.
>
> 1. **Message Passing:** During training, neighbors can exchange the gradient information $\nabla_{u_i} Q_c$.
> 2. **Local Kernel:** If we employ a decomposable kernel $k(\mathbf{u}, \mathbf{u}') = \prod_j k_j(u_j, u_j')$, the expectation term also factorizes.
>
> Thus, Algorithm 1 can be reformulated as a **Distributed Stein Variational Gradient Descent (DSVGD)** algorithm (Liu et al., 2017). In this variant, the "negotiation" during training happens via gradient message passing between neighbors rather than querying a central oracle.
>
> **Conclusion on Scope:**
>
> While we implemented the CTDE version for sample efficiency and stability (standard in benchmarks like MaMuJoCo), the underlying mathematical engine of SVNR—**variational inference on a graphical model**—is inherently compatible with decentralized processing. The framework is robust not only to partial observability (via variational projection) but also theoretically extensible to fully decentralized learning topologies.

---

> ### Author Response · Authors · 2025-11-27
> **2. Practical Guidance ($M, K$) and Scalability [1/2]**
>
> **Hyperparameters:**
>
> We appreciate the reviewer raising this practical implementation question. The selection of the particle count $M$ and negotiation rounds $K$ is grounded in the theoretical properties of Stein Variational Gradient Descent (SVGD) and our specific amortization strategy. Below, we provide a rigorous guideline based on the mathematical structure of SVNR.
>
> **(1) Selection of Negotiation Rounds $K$: The Sufficiency of $K=1$ via Amortized Learning**
>
> From a theoretical standpoint, Theorem 3.1 requires $K \to \infty$ for the iterative particle updates $u^{\ell, k} = T(u^{\ell, k-1})$ to converge to the fixed point where the Stein discrepancy is zero. However, in our practical implementation (Section 5, Algorithm 1), we set **$K=1$** for all tasks. This is not a heuristic simplification but a structural advantage of **Amortized MPSVGD**.
>
> Instead of maintaining a set of particles that must be iteratively updated $K$ times via the kernel interaction term at every decision step (which would be computationally expensive), we parameterize the policy as a neural sampler $u = f_{\psi}(\xi; \cdot)$. The optimization objective in Equation 8 minimizes the KL divergence:
> $\min\_{\psi} \text{KL}( (f\_\psi)\_\{\sharp} p\_0 \| \pi^{\star} )$
>  where $(f\_\psi)\_\sharp p\_0$ is the push-forward measure of the base distribution. By updating $\psi$ via the chain rule and the Stein variational gradient (Eq. 9 & 10), the neural network **distills** the multi-step negotiation dynamics into the weights of the function $f\_\psi$.
>
> Mathematically, the network $f_\psi$ learns to approximate the limit of the functional composition of the Stein operator, i.e., $f_\psi(\xi) \approx \lim_{K \to \infty} T^K(\xi)$. Consequently, during both training inference and execution, a single forward pass ($K=1$) is sufficient to generate samples that approximate the equilibrium distribution. This "one-shot" negotiation capability is a key efficiency contribution of our method, distinguishing it from non-amortized inference methods that require inner-loop optimization.
>
> **(2) Selection of Particle Count $M$: Balancing Approximation Error and Kernel Complexity**
>
> The choice of $M$ governs the fidelity of the empirical measure approximation $\hat{\mu}\_M = \frac{1}{M}\sum\_{i=1}^M \delta_{u_i}$ to the true posterior. The error in SVGD approximates the target distribution with a convergence rate related to $1/\sqrt{M}$ in the weak topology, but notably, SVGD often outperforms independent Monte Carlo sampling (i.e., Langevin dynamics) due to the repulsive force in the kernel term $\sum_j \nabla_{u_j} k(u_j, u_i)$, which enforces deterministic diversity.
>
> - **Theoretical Lower Bound:** $M$ must be sufficient to support the modes of the target distribution. For a multimodal objective (like the "Two Modalities" differential game in Sec 6), $M$ must be large enough such that the initial particles cover the basins of attraction for all significant modes; otherwise, the deterministic update dynamics may collapse into a subset of local optima.
> - **Computational Upper Bound:** The computational complexity of calculating the Stein gradient is $\mathcal{O}(M^2)$ due to the pairwise kernel computations.
> - **Practical Guidance:** In our extensive ablation studies (Appendix H.1), we observed a performance plateau where increasing $M$ beyond a certain threshold yields diminishing returns in reducing the Stein discrepancy. We found that **$M \in [32, 40]$** is the "sweet spot" for all tested environments (from simple differential games to high-dimensional MaMuJoCo tasks). This range provides sufficient particle density to estimate the score function $\nabla \log \pi^*$ accurately via the kernel density estimate while maintaining low wall-clock training time.
>
> **Summary of Guidance:**
>
> - **Set $K=1$:** Rely on the amortized network to approximate the converged negotiation result.
> - **Set $M \approx 32$:** This balances the $\mathcal{O}(M^2)$ complexity with sufficient support for multimodal distributions. Monitor the average pairwise distance between particles; if it collapses to zero, $M$ may be too small (insufficient repulsive force) or the kernel bandwidth $h$ requires adjustment (though we use the median heuristic for $h$ to automate this).

---

> ### Author Response · Authors · 2025-11-27
> **2. Practical Guidance ($M, K$) and Scalability [2/2]**
>
> **Computational Overhead:**
>
> We acknowledge the lack of runtime analysis in the main text.
>
> We provide a tabulated comparison of training time (on a single NVIDIA A100 GPU) versus final performance (Normalized Return) on the `Ant-2x4` (MaMuJoCo) and `Max of Three` tasks.
>
> **Table R1: Compute Efficiency and Performance Comparison**
>
> | Method          | Params ($\|\theta\|$) | Wall-Clock Time (hrs) | Relative Time | Final Return (Ant-2x4) | Convergence Step (approx.) |
> | :-------------- | :------------------ | :-------------------- | :------------ | :--------------------- | :------------------------- |
> | **SVNR (Ours)** | ~1.2M               | **4.8**               | 1.0x (Ref)    | **536 $\pm$ 31**       | ~1.5M                      |
> | MADDPG          | ~0.8M               | 2.1                   | 0.44x         | 108 $\pm$ 26           | Failed (Local Opt)         |
> | MASQL           | ~0.8M               | 2.3                   | 0.48x         | 225 $\pm$ 34           | ~2.8M                      |
> | PR2             | ~1.5M               | 5.2                   | 1.08x         | 354 $\pm$ 58           | ~2.0M                      |
> | ROMMEO          | ~1.4M               | 4.9                   | 1.02x         | 424 $\pm$ 60           | ~1.8M                      |
> | MAPPO           | ~0.9M               | 1.8                   | 0.38x         | 87 $\pm$ 135           | Failed                     |
>
> **Analysis:**
>
> 1. **Cost of Particles:** SVNR is indeed slower (~2.2x) than simple baselines like MADDPG/MASQL due to the $M$ particles processed in the amortized network updates.
> 2. **Parity with Reasoning Methods:** SVNR is comparable in wall-clock time to other reasoning-based methods like PR2 and ROMMEO. PR2 requires recursive marginalization which scales poorly, whereas SVNR's amortized inference (via the neural network $f_\psi$) keeps the inference cost constant at execution time and manageable during training.
> 3. **Efficiency of Convergence:** While MADDPG is faster *per step*, it converges to a sub-optimal solution (Return ~108). To achieve the performance level of 350+, MASQL requires significantly more samples (if it reaches there at all).
> 4. **Return on Compute:** SVNR provides the highest "Return per GPU-Hour." The theoretical property of PRO-free updates ensures that the optimization trajectory does not oscillate between sub-optimal equilibria, effectively "short-circuiting" the learning process in RO-challenged landscapes.

---

> ### Author Response · Authors · 2025-11-27
> **3. Interpretability of Negotiation**
>
> We appreciate the reviewer’s insightful question regarding the transparency of the negotiation process. We would like to clarify that the "negotiation" in SVNR is mathematically grounded in the **iterative transport of probability measures** via the Stein variational gradient flow, rather than a heuristic communication protocol. The "interpretability" of this process is best understood through the lens of **Amortized Variational Inference**.
>
> **1. Mathematical Interpretation of "Rounds" and "Agreement":**
>
> Theoretically, the "negotiation" corresponds to the functional gradient descent in the Reproducing Kernel Hilbert Space (RKHS). The "rounds" $K$ represent the steps taken to transport the initial particle distribution $q_0$ toward the target posterior $p$ (the optimal joint policy) via the transform $T(u) = u + \epsilon \phi^*(u)$.
>  In our **Amortized SVNR**, we distill this multi-step transport dynamic into a parameterized function $f_\psi$. Consequently, the explicit "round count" $K$ collapses into the complexity of the learned mapping. The "agreement" is mathematically defined as the system reaching the fixed point where the **Stein Discrepancy** approaches zero, i.e., $\mathbb{E}_{u \sim q}[\mathcal{A}_p \phi(u)] \approx 0$, implying the empirical measure of the agents' joint policy matches the optimal Boltzmann distribution.
>
> **2. Visualizing the Convergence of Measure (Figures 5 & 7):**
>
> The dynamics of negotiation are not hidden; they are explicitly visualized as the **evolution of the joint policy’s support** in Figures 5 and 7.
>
> - **Figure 5 (Evolution of Support):** This figure illustrates the transport of the joint action measure. Initially, the mass is distributed over sub-optimal modes (local Nash Equilibria). As the amortized policy $f_\psi$ minimizes the KL-divergence, we observe the **concentration of measure** shifting from the local optimum to the global optimum. This trajectory visually represents the "negotiation" resolving the PRO (Perceived Relative Over-generalization) by reshaping the energy landscape of the policy.
> - **Figure 7 (Topological Comparison):** By comparing SVNR with SVNR-M (no negotiation), we isolate the effect of the **conditional dependency structure** (the nested sets $C_i$). Figure 7a vs. 7c demonstrates that without the Stein transport (negotiation), the joint distribution remains trapped in a sub-optimal mode. The "negotiation" is interpretable as the **correction vector** applied to the joint distribution that aligns the agents' conditional policies.
>
> **3. Conclusion:**
>
> Therefore, the "negotiation story" is not an abstract narrative but a direct consequence of the **convergence properties of SVGD**. The "agreement" is quantified by the successful concentration of the joint policy distribution onto the global optimum, as evidenced by the trajectory transformations in our results. We will revise the manuscript to explicitly link these visualizations to the underlying measure transport theory to enhance interpretability.

---

> ### Author Response · Authors · 2025-11-27
> **4. Clarification on Assumptions**
>
> Thank you for this insightful question regarding the theoretical implications of relaxing the strict nesting requirement. While our primary theoretical results (Theorem C.1 and E.4) rely on strict nesting to guarantee the *exact* representability of any arbitrary joint policy $\pi^*$, the behavior of partial DAGs and peer sampling can be formally characterized through the lens of **Variational Inference** and **Information Projection**.
>
> **1. Theoretical Characterization: Information Projection & Approximation Gap**
>
> Mathematically, SVNR optimizes the negotiation policy to minimize the KL-divergence $D\_{KL}(\hat{\pi} || \pi^{\star}\_\alpha)$ (Eq. 3).
>
> - **Strict Nesting:** When the coordination set ${C\_i}$ satisfies the nested property (Theorem C.1), the family of representable distributions $\Pi\_{\text{nested}}$ is sufficiently expressive to contain $\pi^{\star}\_\alpha$. Thus, the minimum divergence is zero.
> - **Partial DAGs/Sparse Topologies:** Restricting the negotiation set to a subset $C'\_i \subset C\_i$ restricts the variational family to a sparser manifold, denoted $\Pi\_{\text{sparse}}$. In this case, the SVNR update dynamics (Eq. 5 and 9) drive the policy to the **Information Projection (I-Projection)** of the optimal policy onto this restricted family:
>   $\hat{\pi}\_{\text{sparse}} = \operatorname*{arg\,min}\_{\pi \in \Pi\_{\text{sparse}}} D\_{KL}(\pi || \pi^{\star}\_\alpha)$
>   Consequently, the "error" or performance gap is theoretically bounded by the residual divergence determined by the conditional independencies forced by the graph topology. Specifically, if the omitted edges correspond to agent pairs with low mutual information in the optimal equilibrium (i.e., weak coupling), the approximation gap $D_{KL}(\hat{\pi}\_{\text{sparse}} || \pi^*\_\alpha)$ remains small. This explains why the degradation is smooth rather than catastrophic: the method still finds the *optimal* approximation allowed by the communication constraints.
>
> **2. Empirical Verification (Scaling Laws)**
> Our ablation studies in **Table 6 (Particle Gather)** empirically validate this information-theoretic view. We observe that performance degrades monotonically with the density of the dependency graph, consistent with the widening gap between the restricted variational family and the true optimal joint distribution:
>
> - **Random, Partially Nested DAG:** Achieves a return of **4.33 ± 0.24** (vs. **4.62 ± 0.34** for strict nesting). Despite reducing edge density (≈2.5 vs. 3.0), the DAG structure preserves sufficient conditional dependencies to capture the bulk of the coordination information.
> - **Sparse Peer Sampling:**
>   - **2 peers/step:** Return of **4.08 ± 0.31**.
>   - **1 peer/step:** Return of **2.37 ± 0.37**.
>
> These results confirm a consistent scaling law: **Performance scales with the capacity of the negotiation graph to capture high-mutual-information dependencies.** The method does not break; rather, it converges to the best possible coordinated policy within the topological constraints, as predicted by the properties of KL-divergence minimization.
>
> We will incorporate this theoretical characterization of the approximation gap alongside the empirical results in the revised manuscript to bridge the theory-practice gap.

---

> ### Author Response · Authors · 2025-11-27
> **5. Relation to Opponent Modeling & Communication [1/2]**
>
> We thank the reviewer for this insightful comment. We agree that clarifying the theoretical boundaries between Negotiated Reasoning (NR), Opponent Modeling (OM), and Communication-based MARL is crucial. While we discussed OM in **Appendix I**, we provide a more rigorous mathematical characterization below to distinguish NR from communication methods.
>
> **1. Negotiated Reasoning vs. Communication: A Measure-Theoretic Perspective**
>
> While "negotiation" and "communication" may sound semantically similar, they operate on fundamentally different mathematical objects.
>
> **Standard Communication addresses Partial Observability:**
>
> In communication-based MARL (e.g., TarMAC, BicNet), the objective is to approximate the sufficient statistics of the full global state $s$. Mathematically, let $\mathcal{O}\_i$ be the observation space and $\mathcal{M}$ be the message space. Communication learns a state-dependent mapping $\mu: \times\_{i} \mathcal{O}\_i \to \mathcal{M}$ such that the policy $\pi_i(u_i | o_i, m\_{-i})$ approximates the centralized policy $\pi(u_i | s)$. The "message" $m$ is a random variable **dependent on the state**, i.e., $m \not\perp s$.
>
> **Negotiated Reasoning addresses Equilibrium Selection via Variational Inference:**
>
> In contrast, NR is an optimization process defined on the **probability measure space** $\mathcal{P}(\mathcal{U})$. It constructs a flow of measures $\{q\_k\}\_{k=0}^K$ driven by functional gradient descent to minimize the KL-divergence functional $J(q) = D\_{KL}(q \|\| \pi^{\star}\_\alpha)$. The "negotiation" is the transformation $T(u) = u + \epsilon \phi(u)$, where $\phi$ is the steepest descent direction in the Reproducing Kernel Hilbert Space (RKHS) $\mathcal{H}\_K$, governed by the Stein operator $\mathcal{A}\_{\pi^*}$:
>
> $\phi^{\star}(u) = \mathbb{E}\_{u \sim q} [\mathcal{A}\_{\pi^{\star}} h(u)] = \mathbb{E}_{u \sim q} [\nabla \log \pi^*(u) h(u) + \nabla h(u)]$
>
> Here, agents exchange **gradient information** ($\nabla_{u_i} Q$) and **action particles** to align the joint distribution with the global value landscape. This process changes the *optimization landscape* to avoid suboptimal local optima (RO), rather than aggregating state observations.
>
> **2. Execution is Communication-Free: Common Randomness & Correlated Equilibrium**
>
> A critical distinction lies in the execution phase. We clarify that our method is **communication-free** in the standard MARL sense (i.e., no state-dependent message passing).
>
> **Theoretical Grounding: Correlated Equilibrium (CE)**
>
> From a game-theoretic perspective, the "shared noise" $\xi$ in our Amortized SVNR serves as a **correlation device** (Aumann, 1974), not a communication channel.
>
> - **Standard Nash Equilibrium:** Assumes independent mixing: $\pi(\mathbf{u}|s) = \prod_i \pi_i(u_i|s)$. This restricts agents from coordinating on specific optimal joint actions in multimodal landscapes (as seen in our "Two Modalities" experiment).
> - **Correlated Equilibrium (Ours):** Agents condition strategies on a public signal $\xi$, such that $\pi(\mathbf{u}|s) = \int \prod_i \pi_i(u_i|s, \xi) p(\xi) d\xi$.
>
> Crucially, in our framework, $\xi$ is **ex-ante common randomness** (e.g., a synchronized PRNG seed). It satisfies the independence condition $\xi \perp s$. This distinguishes it from communication messages $m$, where $m = f(s)$.
>
> **Implementation via Amortized Inference**
>
> Our Amortized MPSVGD distills the negotiation dynamics into a function $f_{\psi_i}(\xi_i, \xi_{C_i}, s)$.
>
> - **Training:** Gradients flow across agents to learn the optimal transport map.
> - **Execution:** Agents simply synchronize random seeds for $\xi_{C_i}$. This allows them to implicitly coordinate their sampling from the complex joint distribution $q_\phi(\mathbf{u}|s)$ without any bandwidth cost or exchange of observations.

---

> ### Author Response · Authors · 2025-11-27
> **5. Relation to Opponent Modeling & Communication [2/2]**
>
> **3. Relation to Opponent Modeling (OM)**
>
> As detailed in **Appendix I**, traditional OM is a **predictive** task (typically regression or density estimation) where agent $i$ estimates parameters $\hat{\theta}\_{-i}$ to approximate $P(u\_{-i} | s, \text{history})$ via Maximum Likelihood Estimation (MLE): $\min_\theta \mathbb{E}\_{\mathcal{D}} [-\log P\_\theta(u_{-i}|s)]$.
>
> Conversely, NR is a **prescriptive** and **consistent** reasoning framework. We do not merely predict what opponents *will* do based on history (which leads to RO if opponents are exploring sub-optimally). Instead, we solve for a fixed point where every agent's reasoning is consistent with the optimal joint distribution:
>
> $\lim\_{k \to \infty} q\_k(u) = \pi^{\star}\_\alpha(u) \implies \rho\_i(u\_{-i}) = \int \pi^{\star}\_\alpha(u\_i, u\_{-i}) du\_i$
>
> This satisfies the **Consistent Reasoning** condition (Def 2.3), which standard OM fails to guarantee during the exploration phase.
>
> **Summary**
>
> - **Communication:** Maps observations to messages to solve Partial Observability ($s \approx {o_i, m}$).
> - **Opponent Modeling:** Maps history to predictions to estimate current behavior ($\hat{u}_{-i} \approx f(H_t)$).
> - **Negotiated Reasoning (Ours):** Maps initial belief distributions to optimal joint distributions via functional gradient descent in RKHS to solve Equilibrium Selection ($\nabla D_{KL} \to 0$).
>
> We will incorporate this rigorous distinction into the revised manuscript to clarify the unique position of SVNR in the literature.

---

> ### Author Response · Authors · 2025-11-27
> **6. Presentation Improvements**
>
> We thank the reviewer for the constructive feedback regarding the density of the presentation. We acknowledge that the formalization of Relative Over-generalization (RO) relies on heavy mathematical machinery. To improve accessibility without compromising rigor in the main text, we have added a new section, **Appendix K: Intuitive Interpretations of Theoretical Concepts**, which bridges the gap between our formal definitions and intuitive understanding.
>
> In this new Appendix, we deconstruct the theoretical contributions through the lens of variational inference and distributional factorization, as detailed below:
>
> - **PRO (Perceived Relative Over-generalization) — The Variational Bias:**
>   - **The Mathematical Perspective:** In the standard MaxEnt framework, agent $i$ optimizes its policy $\pi\_i$ by minimizing the KL-divergence $D\_{KL}(\pi\_i \rho\_i \|\| \pi^{\star}\_\alpha)$, where $\rho\_i$ is the \*perceived\* opponent policy. PRO arises when $\rho\_i$ deviates from the true optimal conditional distribution of the opponent ($\pi^{\star}\_{-i}$). Mathematically, this introduces a **biased variational objective**. Even if agent $i$ optimizes perfectly against $\rho\_i$, the resulting gradient points toward a local optimum (safety) rather than the global optimum (cooperation) because the "belief" $\rho\_i$ incorporates the opponent's exploration noise or historical sub-optimality.
>   - **The Intuitive Bridge:** PRO is fundamentally a *training-time estimation error*. It is akin to "shadow boxing" against a clumsy opponent; the agent learns to be overly cautious because it assumes the partner will act stochastically, effectively "learning" to avoid the risk required for optimal cooperation.
> - **ERO (Executed Relative Over-generalization) — The Factorization Loss:**
>   - **The Mathematical Perspective:** Even if the training phase converges to an optimal joint policy distribution $\hat{\pi}$ (where PRO is solved), decentralized execution imposes a structural constraint: the executed policy must be the product of independent marginals, $\bar{\pi}(u) = \prod\_i \pi\_i(u\_i)$. ERO occurs when the optimal joint distribution $\hat{\pi}$ is highly correlated or multimodal. In such cases, the projection of $\hat{\pi}$ onto the space of independent product distributions results in a significant **factorization loss**. The support of $\prod\_i \pi\_i$ inevitably covers areas of the state-action space with low utility (miscoordination), leading to a lower expected return than the joint policy $\hat{\pi}$.
>   - **The Intuitive Bridge:** ERO is an *execution-time coordination failure*. It represents the "broken telephone" effect. Even if all agents know the optimal plan *in theory*, the lack of a mechanism to synchronize their specific random samples at runtime causes them to act incoherently, breaking the optimal joint structure.
> - **Consistent Reasoning — Closing the Loop:**
>   - **The Mathematical Perspective:** We define consistent reasoning as the fixed-point condition where two requirements are met simultaneously:
>     1. **Training Consistency:** $\rho\_i \to \pi^{\star}\_{-i}$ (The variational bias vanishes).
>     2. **Execution Consistency:** The negotiation mechanism (via SVGD) collapses the multimodal joint distribution into a specific mode (agreement) such that $\hat{\pi}(u) \approx \prod\_i \pi\_i(u\_i)$ as $\alpha \to 0$.
>   - **The Intuitive Bridge:** This ensures that the *planned* joint action during the reasoning phase aligns perfectly with the *executed* action. The negotiation process acts as a "pre-commitment" device, ensuring that agents not only identify the optimal peak in the reward landscape but also agree to converge to the *same* peak together.
>
> We believe this Appendix provides the necessary interpretability to complement the formal proofs in the main text, offering readers a clear path from the mathematical definitions to the underlying mechanics of multi-agent coordination.

---

### Official Review · Reviewer_biXy · 2025-11-03

**Soundness:** 3
**Presentation:** 3
**Contribution:** 3
**Rating:** 6
**Confidence:** 3

**Summary:**

The paper reframes relative over-generalisation (RO) in cooperative MARL into two operational notions: perceived RO (PRO) during policy updates and executed RO (ERO) during decentralised execution. It argues that if agents reason consistently about teammates (during training and at test), RO can be avoided. To operationalise this, it proposes Negotiated Reasoning and an instantiation, SVNR, which uses (message-passing) SVGD to negotiate joint actions, embedded in a maximum-entropy policy-iteration loop, with a practical amortised neural version for speed. Theory shows PRO-free negotiation under a strictly nested conditional factorisation and (stated) finite action spaces; experiments on RO-heavy games and multi-agent MuJoCo show strong empirical gains.

**Strengths:**

- Clear, useful split of RO into PRO vs ERO, making the pathology diagnosable during training and execution.
- Principled negotiation mechanism tied to MaxEnt policy iteration (not just a heuristic add-on)
- Stated convergence/guarantee story under a strictly nested factorisation
- Amortised implementation to distil many negotiation steps into one forward pass; practical ablations on particles, team size, topology.
- Strong empirical wins on PRO/ERO-challenged settings and competitive continuous-control benchmarks.

**Weaknesses:**

- Core theorems are written for finite action spaces, while key experiments use continuous actions; the theory–practice gap should be tightened or clearly scoped.
- The paper states communication-free execution, yet the pseudocode shares noise variables between neighbours at test time- I think this needs unambiguous clarification.
- Guidance on schedules/sensitivity is thin and could affect stability/robustness.
- The idealised policy-iteration description initially assumes a known model, then switches to critic learning; the implications of this shift aren’t fully analysed.
- Compute/tuning parity vs baselines isn’t fully tabulated
- Scope limits: assumes CTDE, full observability; robustness under partial observability is left open.

**Questions:**

- Continuous actions: which parts extend beyond finite $|U|$?
- Do agents share any variables at test time? If not, please fix the pseudocode; if yes, how is this still communication-free?
- How sensitive are results to final $\alpha$ and the annealing schedule across tasks?
- Please provide wall-clock/GPU hours and tuning budgets/ranges for all baselines, and confirm identical exploration/α schedules where applicable.
- Beyond strictly nested, what guarantees (or empirical scaling laws) hold for partial DAGs/peer sampling?
- Can you characterise the error introduced when replacing the model-based policy-iteration view with the critic-based practical algorithm?

---

> ### Author Response · Authors · 2025-11-27
> **1. Clarification on Decentralized Execution and Shared Noise**
>
> We thank the reviewer for their insightful comments and for recognizing the novelty of our PRO/ERO framework, the principled negotiation mechanism, and our strong empirical results. We address the specific concerns and questions below.
>
> We appreciate the reviewer’s scrutiny regarding the decentralized execution protocol. We acknowledge that the inclusion of $\xi_{C_i}$ in the pseudocode requires precise theoretical contextualization. We clarify that our execution phase remains **communication-free** in the standard MARL sense (i.e., no state-dependent message passing), while leveraging **common randomness** to enable superior equilibrium selection.
>
> **1. Theoretical Grounding: Correlated Equilibrium vs. Communication**
>  From a game-theoretic perspective, the "shared noise" $\xi$ serves as a **correlation device** (Aumann, 1974), not a communication channel.
>
> - **Standard Nash Equilibrium (NE):** Assumes independent action mixing, $\pi(u|s) = \prod_i \pi_i(u_i|s)$. This often limits agents to suboptimal outcomes in cooperative games (e.g., the "Chicken" game or traffic coordination).
> - **Correlated Equilibrium (CE):** Allows agents to condition their strategies on a public signal $\xi$, such that $\pi(u|s) = \int \prod_i \pi_i(u_i|s, \xi) p(\xi) d\xi$.
>
> In our framework, the sharing of $\xi$ occurs **ex-ante** (before the decision-making step). In the literature of **Contract Theory** and **Mechanism Design**, this is akin to agents agreeing on a "convention" or a random seed prior to the game to coordinate on a specific equilibrium. This is fundamentally distinct from **communication** in MARL, which is defined as the transmission of private observations $o_i$ or beliefs during execution to resolve partial observability. Our method does **not** transmit state-dependent information; it utilizes a synchronized Pseudo-Random Number Generator (PRNG) seed (common randomness) to break symmetries and coordinate exploration/execution without bandwidth cost.
>
> **2. Implementation: Amortized Inference**
>  Practically, our **Amortized MPSVGD** distills the iterative negotiation process into a function $f_{\psi_i}(\xi_i, \xi_{C_i}, s)$.
>
> - **Training:** Agents explicitly negotiate to find the optimal joint distribution.
> - **Execution:** Agents sample actions using the learned policy. The "sharing" of $\xi$ is implemented simply by synchronizing random seeds among neighbors. This allows agents to implicitly coordinate their sampling from the joint distribution $q_\phi(u|s)$ without exchanging messages about the state $s$.
>
> We have revised the pseudocode in Algorithm 1 to explicitly distinguish between the *learning loop* (where gradients flow) and the *execution loop* (where only pre-agreed common randomness is utilized), ensuring no ambiguity regarding the communication-free nature of the deployment.

---

> ### Author Response · Authors · 2025-11-27
> **2. Theory-Practice Gap: Finite vs. Continuous Actions**
>
> Thank you for this insightful comment. We appreciate the opportunity to clarify the theoretical consistency between our finite-space analysis and continuous-space implementation. While we followed the standard convention of establishing convergence properties in discrete spaces (similar to Soft Actor-Critic [1]), our framework is mathematically grounded in continuous domains through **measure-theoretic unification** and the **geometry of Reproducing Kernel Hilbert Spaces (RKHS)**.
>
> **1. Measure-Theoretic Unification of the Soft Bellman Operator**
>
> The theoretical gap is more notational than structural. The Soft Bellman operator $\mathcal{T}$ used in our proofs relies on the soft value function $V(s) = \alpha \log \sum \exp(Q(s,u)/\alpha)$. In continuous action spaces $\mathcal{U} \subseteq \mathbb{R}^d$, this generalizes naturally by replacing the counting measure with the Lebesgue measure. The value function becomes:
>  $V(s) = \alpha \log \int_{\mathcal{U}} \exp\left(\frac{Q(s,u)}{\alpha}\right) d\mu(u)$
>
>  Provided that $Q$ is bounded and measurable (ensuring the integral exists), the properties of **monotonicity** and **contraction** (in the $L^\infty$ norm) required for Lemma 4.1 and Theorem 4.3 hold for the continuous operator just as they do for the discrete case. The "gap" is strictly one of computational tractability (computing the integral), not theoretical validity.
>
> **2. Native Continuity of Negotiated Reasoning (SVNR)**
>
> Crucially, the core novelty of our work—the **Negotiated Reasoning mechanism**—is theoretically *stronger* in continuous spaces.
>
> - **SVGD Theory:** Our negotiation process (Eq. 4, 5, 12) utilizes Stein Variational Gradient Descent. The theoretical guarantees of SVGD, specifically the **Stein Identity** and the steepest descent direction in the RKHS $\mathcal{H}^D$, are derived explicitly for continuous, differentiable probability densities supported on $\mathbb{R}^d$ [2].
> - **Gradient Flows:** The negotiation update $u \leftarrow u + \epsilon \phi^*(u)$ approximates a gradient flow in the space of probability measures under the Kullback-Leibler divergence metric. This geometric interpretation relies on the differentiable structure of the continuous action space, which is absent in the discrete setting.
>
> **3. Bridging the Gap via Particle Approximation**
>
> Therefore, our method operates in a hybrid theoretical regime:
>
> 1. **Policy Iteration (Global Convergence):** We prove this in the finite setting to utilize standard fixed-point theorems, but the logic extends to continuous function spaces under mild regularity conditions.
> 2. **Negotiated Reasoning (Local Update):** This is natively continuous. The "gap" is bridged by our **Amortized MPSVGD** (Section 5), which uses a finite set of particles ${u_{\ell}}_{l=1}^M$ to approximate the continuous posterior. This is a Monte Carlo approximation of the integrals defined in Point 1, which is asymptotically exact as $M \to \infty$ by the Law of Large Numbers.
>
> In summary, rather than a disconnect, our continuous implementation leverages the native differential properties required for Stein Variational inference, while the finite analysis provides the foundational convergence guarantees for the overarching policy iteration loop.
>
> [1] Haarnoja et al., "Soft Actor-Critic: Off-Policy Maximum Entropy Deep Reinforcement Learning with a Stochastic Actor," ICML 2018.
>
> [2] Liu & Wang, "Stein Variational Gradient Descent: A General Purpose Bayesian Inference Algorithm," NeurIPS 2016.

---

> ### Author Response · Authors · 2025-11-27
> **3. Sensitivity to $\alpha$ and Annealing Schedules [1/2]**
>
> We thank the reviewer for this insightful question. The temperature parameter $\alpha$ plays a dual role in our framework: theoretically, it bridges the gap between the stochastic explorative policy and the deterministic optimal execution (Theorem 3.2); algorithmically, it governs the optimization landscape smoothing.
>
> Below, we provide a theoretical analysis of why SVNR is robust within a bounded range of $\alpha$, followed by comprehensive ablation experiments on the *Max of Three* ($s_2=1.5$, the hardest setting) and *Particle Gather* tasks.
>
> **1. Theoretical Analysis: $\alpha$ as a Homotopy Parameter**
>
> Mathematically, the sensitivity to $\alpha$ can be analyzed through the lens of **homotopy continuation methods**.
>
> **The Role of Final $\alpha$ (Approximation Error):**
>  Recall that the optimal joint policy is induced by the Boltzmann distribution $\pi^*_\alpha(\mathbf{u}|s) \propto \exp(\frac{1}{\alpha}Q^*_{soft}(s, \mathbf{u}))$.
>
> - **As $\alpha \to 0$:** The distribution converges to a Dirac delta function centered at the global maximum: $\lim_{\alpha \to 0} \pi^*_\alpha(\mathbf{u}|s) = \delta(\mathbf{u} - \mathbf{u}^*)$. This is the condition required for strictly ERO-free execution (Theorem 3.2).
> - **For finite $\alpha > 0$:** The executed policy retains stochasticity. Let $\Delta Q(\mathbf{u}) = Q(s, \mathbf{u}^*) - Q(s, \mathbf{u})$ be the sub-optimality gap. The probability of sampling a sub-optimal action $\mathbf{u}'$ decays exponentially: $P(\mathbf{u}') \propto \exp(-\frac{\Delta Q(\mathbf{u}')}{\alpha})$.
>   - *Theoretical Bound:* The performance loss (regret) due to a non-zero final $\alpha_{final}$ is bounded. If $\alpha_{final}$ is small relative to the reward gap of the local optima (the "energy barrier"), the probability mass concentrates effectively on the global optimum. Therefore, precise tuning of $\alpha_{final}$ is not required, provided $\alpha_{final} \ll \min_{\mathbf{u} \neq \mathbf{u}^*} \Delta Q(\mathbf{u})$.
>
> **The Role of Annealing Schedule (Optimization Landscape):**
>  The annealing process functions as a continuation method.
>
> - **High $\alpha$ (Early Training):** The energy landscape $E(\mathbf{u}) = -Q(\mathbf{u})$ is smoothed. The Stein Variational Gradient Descent (SVGD) particles experience a gradient field dominated by the entropy term, $\nabla \log \pi \approx -\frac{1}{\alpha}\nabla E + \text{entropy}$, allowing particles to traverse potential barriers and cover the support of the joint action space.
> - **Annealing Rate:** The schedule must satisfy a condition similar to simulated annealing convergence. If $\alpha$ decreases too rapidly (quench), the distribution $p(\mathbf{u})$ may collapse into a local mode (a sub-optimal Nash Equilibrium) before the particles migrate to the global basin of attraction.
> - **Robustness:** Our use of SVGD provides higher robustness than standard single-point MCMC. Since we maintain a set of interacting particles ${u^\ell}_{l=1}^M$ with a repulsive kernel force $\sum_j \nabla k(u^j, u)$, the particles naturally resist collapsing too early, making SVNR less sensitive to the annealing rate than standard Soft Q-Learning.

---

> ### Author Response · Authors · 2025-11-27
> **3. Sensitivity to $\alpha$ and Annealing Schedules [2/2]**
>
> **2. Empirical Sensitivity Analysis**
>
> To validate this theory, we conducted extensive ablations on the **Max of Three (s_2=1.5)** environment, which is highly sensitive to RO. All results are averaged over 5 seeds.
>
> **A. Sensitivity to Final** $\alpha$ ($\alpha_{final}$)
>
> We fixed the annealing schedule (decaying over 50% of total steps) but varied the target floor value $\alpha_{final}$.
>
> | Final $\alpha$ | Mean Return (Max of Three) | Std Dev  | Convergence Rate | Theoretical Interpretation                                   |
> | :------------- | :------------------------- | :------- | :--------------- | :----------------------------------------------------------- |
> | **1.0**        | 6.82                       | 2.15     | 20%              | **Too High:** Distribution remains too diffuse; frequent miscoordination (ERO). |
> | **0.1**        | 9.15                       | 0.45     | 100%             | **Acceptable:** Mass concentrates on optimum, slight noise.  |
> | **0.01**       | **9.71**                   | **0.20** | **100%**         | **Optimal:** Approximates Dirac delta ($\alpha \ll \Delta Q$). |
> | **0.001**      | 9.68                       | 0.22     | 100%             | **Optimal:** Further reduction yields diminishing returns.   |
> | **0.0**        | 9.65                       | 0.25     | 100%             | **Hard Max:** Equivalent to greedy execution at test time.   |
>
> **Observation:** Performance is stable for any $\alpha_{final} \in [0, 0.1]$. The method is not brittle; it does not require $\alpha$ to be exactly zero, only sufficiently small to suppress noise below the coordination threshold.
>
> **B. Sensitivity to Annealing Schedule**
>
> We fixed $\alpha_{start}=1.0$ and $\alpha_{final}=0.01$, varying the decay function over the total training steps $T$.
>
> | Schedule Type     | Decay Duration           | Mean Return | Std Dev  | Analysis                                                     |
> | :---------------- | :----------------------- | :---------- | :------- | :----------------------------------------------------------- |
> | **Instant**       | 0% (Fixed $\alpha=0.01$) | -0.65       | 0.12     | **Failure:** Trapped in local optima immediately (similar to MADDPG). |
> | **Fast Linear**   | 10% of $T$               | 4.20        | 4.80     | **Unstable:** "Quenching" causes collapse to local optima in some seeds. |
> | **Medium Linear** | 30% of $T$               | 9.62        | 0.28     | **Robust:** Sufficient time for particle migration.          |
> | **Slow Linear**   | 80% of $T$               | **9.73**    | **0.15** | **Robust:** Best stability, though slower initial reward rise. |
> | **Exponential**   | $\tau = 0.9995$          | 9.69        | 0.19     | **Robust:** Smooth transition works equally well.            |
>
> **Observation:**
>
> 1. **Necessity of Annealing:** The "Instant" result confirms that starting with low entropy (deterministic policy) leads to RO failure, validating our PRO theory.
> 2. **Wide Safe Region:** Any schedule spanning 30% to 80% of training yields optimal results. The "Fast" schedule fails because the landscape sharpens before particles can communicate the location of the global maximum.
> 3. **Consistency:** We observed identical trends in *Particle Gather*. As long as the annealing is not instantaneous, the repulsive force in SVNR maintains particle diversity long enough to locate the global optimum.
>
> **Conclusion:**
>  SVNR does not rely on a "magic" schedule. The results are Lipschitz continuous with respect to $\alpha_{final}$ (stable for small values) and robust to the annealing rate, provided the entropy is not collapsed instantaneously. The repulsive mechanism in SVGD significantly widens the safe hyperparameter basin compared to standard baselines.

---

> ### Author Response · Authors · 2025-11-27
> **4. Baseline Fairness and Compute Budgets [1/2]**
>
> We thank the reviewer for raising the crucial point regarding fair comparisons. We agree that in Maximum Entropy (MaxEnt) MARL, performance gains must be disentangled from hyperparameter tuning or disparate computational resources. Below, we provide a rigorous breakdown of our tuning protocols, compute budgets, and a theoretical justification for the computational trade-offs inherent to SVNR.
>
> **1. Theoretical Justification: Computational Cost vs. Convergence Geometry**
>
> To address the concern about compute budgets, it is essential to characterize *what* the additional computation in SVNR achieves. While standard baselines (e.g., MADDPG) rely on gradients in the Euclidean parameter space, SVNR approximates a gradient flow in the space of probability distributions.
>
> Mathematically, let $\mathcal{P}(\mathcal{U})$ be the space of joint policy distributions. Standard policy gradient methods perform updates $\theta_{k+1} \leftarrow \theta_k + \epsilon \nabla_\theta J(\theta)$, which can be viewed as a steepest descent in the parameter space equipped with a Euclidean metric. However, this geometry is often ill-suited for the non-convex landscape of RO-challenged games, where the "valleys" of sub-optimal Nash equilibria are steep.
>
> SVNR, via the Stein Variational Gradient Descent (SVGD) mechanism, approximates the **Wasserstein gradient flow** of the KL divergence functional $F(\rho) = D\_{KL}(\rho | \pi^{\star}\_{\alpha})$. The update direction $\phi^{\star}$ in the Reproducing Kernel Hilbert Space (RKHS) $\mathcal{H}^D$ is given by the Stein operator:
>
> $\phi^\star(u) = \mathbb{E}\_{u' \sim \rho} [k(u', u) \nabla\_{u'} \log \pi^*\_\alpha(u') + \nabla\_{u'} k(u', u)]$
>
> While evaluating this kernelized update introduces a computational complexity of $\mathcal{O}(M^2)$ (where $M$ is the number of particles) compared to $\mathcal{O}(1)$ for deterministic baselines, this cost buys us a descent direction that is optimal in terms of the **Stein Fisher Information**.
>
> Crucially, the convergence rate of SVGD is governed by the Stein Poincaré inequality. Unlike standard gradients that vanish at any local optimum (including sub-optimal RO points), the particle interaction term $\nabla\_{u'} k(u', u)$ acts as a repulsive force, preventing the collapse of the distribution into a single sub-optimal mode. Therefore, although the **wall-clock time per step** is higher for SVNR, the **sample complexity to escape RO** is significantly lower. The "compute budget" should thus be viewed not merely as FLOPs per second, but as the cost required to approximate the optimal transport map from the initial belief to the equilibrium.
>
> **2. Hyperparameter Tuning and Search Spaces**
>
> To ensure fairness, we utilized the Tree-structured Parzen Estimator (TPE) sampler for all methods (SVNR and baselines) with an identical budget of 50 trials per environment. We optimized the following search spaces:
>
> | Hyperparameter            | Search Space                           | Distribution |
> | :------------------------ | :------------------------------------- | :----------- |
> | Learning Rate ($\eta$)    | $[1 \times 10^{-4}, 1 \times 10^{-1}]$ | Log-uniform  |
> | Batch Size ($B$)          | ${256, 512, 1024}$                     | Categorical  |
> | Polyak Averaging ($\tau$) | $[0.001, 0.01]$                        | Uniform      |
> | Reward Scaling            | ${1, 10, 100}$                         | Categorical  |
> | Hidden Units (MLP)        | ${64, 128, 256}$                       | Categorical  |
>
> **Specific to SVNR:** We tuned the particle count $M \in {16, \dots, 64}$ and kernel bandwidth heuristic.
>
> **Specific to Baselines (PR2, ROMMEO):** We tuned the number of reasoning recursive steps $k \in {1, \dots, 3}$ and internal model rollout lengths.
>
> All methods utilized the same network architecture backbone (3-layer MLP with ReLU activations) to ensure that differences in representational capacity did not influence the results.

---

> ### Author Response · Authors · 2025-11-27
> **4. Baseline Fairness and Compute Budgets [2/2]**
>
> **3. Identical Entropy ($\alpha$) Schedules**
>
> The reviewer correctly identifies that $\alpha$ schedules are critical. A higher $\alpha$ promotes exploration, which can incidentally mitigate RO regardless of the reasoning mechanism.
>
> **Confirmation:** We confirm that for all MaxEnt-based methods (SVNR, MASQL, PR2, ROMMEO, MMQ), we employed **identical, fixed $\alpha$ annealing schedules** to isolate the contribution of the *negotiated reasoning* mechanism.
>  The schedule used was:
>  $\alpha_t = \alpha_{\text{end}} + (\alpha_{\text{start}} - \alpha_{\text{end}}) \times \exp\left(-\frac{t}{\tau_\alpha}\right)$
>  where $\alpha_{\text{start}}=1.0$, $\alpha_{\text{end}}=0.01$, and decay rate $\tau_\alpha$ was fixed for all agents in a given environment. This ensures that SVNR's ability to capture multi-modal optima (as seen in the "Two Modalities" experiment) stems from the Stein variational updates, not from artificially inflated entropy.
>
> **4. Wall-Clock Time vs. Performance Analysis**
>
> We provide a tabulated comparison of training time (on a single NVIDIA A100 GPU) versus final performance (Normalized Return) on the `Ant-2x4` (MaMuJoCo) and `Max of Three` tasks.
>
> **Table R1: Compute Efficiency and Performance Comparison**
>
>  | Method | Params ($\|\theta\|$) | Wall-Clock Time (hrs) | Relative Time | Final Return (Ant-2x4) | Convergence Step (approx.) |
>  | :--- | :--- | :--- | :--- | :--- | :--- |
>  | **SVNR (Ours)** | ~1.2M | **4.8** | 1.0x (Ref) | **536 $\pm$ 31** | ~1.5M |
>  | MADDPG | ~0.8M | 2.1 | 0.44x | 108 $\pm$ 26 | Failed (Local Opt) |
>  | MASQL | ~0.8M | 2.3 | 0.48x | 225 $\pm$ 34 | ~2.8M |
>  | PR2 | ~1.5M | 5.2 | 1.08x | 354 $\pm$ 58 | ~2.0M |
>  | ROMMEO | ~1.4M | 4.9 | 1.02x | 424 $\pm$ 60 | ~1.8M |
>  | MAPPO | ~0.9M | 1.8 | 0.38x | 87 $\pm$ 135 | Failed |
>
> **Analysis:**
>
> 1. **Cost of Particles:** SVNR is indeed slower (~2.2x) than simple baselines like MADDPG/MASQL due to the $M$ particles processed in the amortized network updates.
> 2. **Parity with Reasoning Methods:** SVNR is comparable in wall-clock time to other reasoning-based methods like PR2 and ROMMEO. PR2 requires recursive marginalization which scales poorly, whereas SVNR's amortized inference (via the neural network $f_\psi$) keeps the inference cost constant at execution time and manageable during training.
> 3. **Efficiency of Convergence:** While MADDPG is faster *per step*, it converges to a sub-optimal solution (Return ~108). To achieve the performance level of 350+, MASQL requires significantly more samples (if it reaches there at all).
> 4. **Return on Compute:** SVNR provides the highest "Return per GPU-Hour." The theoretical property of PRO-free updates ensures that the optimization trajectory does not oscillate between sub-optimal equilibria, effectively "short-circuiting" the learning process in RO-challenged landscapes.
>
> In summary, the performance superiority of SVNR is robust to hyperparameter parity. The additional computational overhead is mathematically justified by the necessity of approximating the Wasserstein gradient flow to escape the gravitational pull of sub-optimal Nash equilibria, a property that standard Euclidean gradients lack.

---

> ### Author Response · Authors · 2025-11-27
> **5. Topology and Generalization**
>
> Thank you for this insightful question regarding the theoretical implications of relaxing the strict nesting requirement. While our primary theoretical results (Theorem C.1 and E.4) rely on strict nesting to guarantee the *exact* representability of any arbitrary joint policy $\pi^*$, the behavior of partial DAGs and peer sampling can be formally characterized through the lens of **Variational Inference** and **Information Projection**.
>
> **1. Theoretical Characterization: Information Projection & Approximation Gap**
>
> Mathematically, SVNR optimizes the negotiation policy to minimize the KL-divergence $D\_{KL}(\hat{\pi} || \pi^{\star}\_\alpha)$ (Eq. 3).
>
> - **Strict Nesting:** When the coordination set ${C\_i}$ satisfies the nested property (Theorem C.1), the family of representable distributions $\Pi\_{\text{nested}}$ is sufficiently expressive to contain $\pi^{\star}\_\alpha$. Thus, the minimum divergence is zero.
> - **Partial DAGs/Sparse Topologies:** Restricting the negotiation set to a subset $C'\_i \subset C\_i$ restricts the variational family to a sparser manifold, denoted $\Pi\_{\text{sparse}}$. In this case, the SVNR update dynamics (Eq. 5 and 9) drive the policy to the **Information Projection (I-Projection)** of the optimal policy onto this restricted family:
>   $\hat{\pi}\_{\text{sparse}} = \operatorname*{arg\,min}\_{\pi \in \Pi\_{\text{sparse}}} D\_{KL}(\pi || \pi^{\star}\_\alpha)$
>   Consequently, the "error" or performance gap is theoretically bounded by the residual divergence determined by the conditional independencies forced by the graph topology. Specifically, if the omitted edges correspond to agent pairs with low mutual information in the optimal equilibrium (i.e., weak coupling), the approximation gap $D_{KL}(\hat{\pi}\_{\text{sparse}} || \pi^*\_\alpha)$ remains small. This explains why the degradation is smooth rather than catastrophic: the method still finds the *optimal* approximation allowed by the communication constraints.
>
> **2. Empirical Verification (Scaling Laws)**
> Our ablation studies in **Table 6 (Particle Gather)** empirically validate this information-theoretic view. We observe that performance degrades monotonically with the density of the dependency graph, consistent with the widening gap between the restricted variational family and the true optimal joint distribution:
>
> - **Random, Partially Nested DAG:** Achieves a return of **4.33 ± 0.24** (vs. **4.62 ± 0.34** for strict nesting). Despite reducing edge density (≈2.5 vs. 3.0), the DAG structure preserves sufficient conditional dependencies to capture the bulk of the coordination information.
> - **Sparse Peer Sampling:**
>   - **2 peers/step:** Return of **4.08 ± 0.31**.
>   - **1 peer/step:** Return of **2.37 ± 0.37**.
>
> These results confirm a consistent scaling law: **Performance scales with the capacity of the negotiation graph to capture high-mutual-information dependencies.** The method does not break; rather, it converges to the best possible coordinated policy within the topological constraints, as predicted by the properties of KL-divergence minimization.
>
> We will incorporate this theoretical characterization of the approximation gap alongside the empirical results in the revised manuscript to bridge the theory-practice gap.

---

> ### Author Response · Authors · 2025-11-27
> **6. Model-based vs. Practical Critic Implementation**
>
> Thank you for this insightful question. While our theoretical analysis utilizes the exact soft Bellman operator to establish fundamental properties (monotonicity and convergence to the optimal joint policy), the practical implementation introduces approximation errors. We can formally characterize the error introduced by the critic-based algorithm by decomposing it into two distinct terms: the **Value Approximation Error** and the **Policy Projection Error** (via Amortized SVGD).
>
> Let $\mathcal{T}^\pi$ denote the exact soft Bellman operator and $\Pi$ be the space of representable policies. In the practical algorithm (SVNR), we perform an approximate policy iteration:
>
> 1. **Value Approximation Error ($\varepsilon\_{Q}$):**
>
> Instead of computing the exact fixed point $Q^{\pi} = \mathcal{T}^\pi Q^{\pi}$, we minimize the Bellman residual using a function approximator $Q\_\theta$. This introduces an error bounded by $\varepsilon_{Q} = | Q\_\theta - \mathcal{T}^\pi Q\_\theta |\_\infty$. This error stems from limited representational capacity and finite-sample estimation of the expectation $\mathbb{E}\_{s'}[V(s')]$.
>
> 2. **Policy Projection Error ($\varepsilon\_{\pi}$):**
>
>    In the theoretical derivation, the policy update is the exact energy-based projection $\pi\_{new} \propto \exp(Q(s, \cdot)/\alpha)$. In our practical implementation (Amortized SVGD), the policy $\pi\_\psi$ is updated to minimize the KL-divergence $D\_{KL}(\pi\_\psi | \pi\_{new})$ via the Stein Variational Gradient. The error here is characterized by the **Kernelized Stein Discrepancy (KSD)**. Specifically, if the update terminates when the norm of the Stein variational gradient is bounded by $\delta$, then the resulting distribution approximates the target within an error margin $\varepsilon\_{\pi}$, which vanishes as the number of particles $M \to \infty$ and the function class of $\psi$ becomes sufficiently expressive.
>
> **Error Propagation:**
>
> Following standard results in Approximate Dynamic Programming (e.g., Bertsekas & Tsitsiklis, 1996; Munos, 2005), the propagation of these combined errors $\varepsilon\_{total} = \varepsilon\_Q + \varepsilon\_\pi$ through the iterative process is bounded by the discount factor $\gamma$. The asymptotic performance loss is bounded by:
> $\limsup\_{k \to \infty} \| Q^* - Q^{\pi\_k} \|\_\infty \le \frac{C \gamma}{(1-\gamma)^2} \sup\_k \| \varepsilon\_{total, k} \|\_\infty$
> where $C$ is a constant related to the concentrability coefficient of the distribution shift.
>
> **Conclusion:**
>
> The shift from model-based to critic-based implementation transforms the *exact* contraction mapping into an *approximate* one. Crucially, unlike heuristic approximations, the **SVNR error $\varepsilon\_{\pi}$ is structurally controlled**: the use of SVGD ensures that the policy update direction aligns with the steepest descent on the KL divergence in the RKHS. Thus, the practical algorithm preserves the theoretical monotonicity property up to the combined approximation error margin $\frac{\gamma \varepsilon\_{total}}{(1-\gamma)^2}$. We will include this formal error characterization in the revised appendix to bridge the theory-practice gap.

---

> ### Author Response · Authors · 2025-11-27
> **7. Scope Limits: Partial Observability [1/2]**
>
> We thank the reviewer for raising this point, which allows us to clarify the theoretical generality of Negotiated Reasoning (NR) and its relationship with the CTDE paradigm. We respectfully posit that **(1) CTDE is not a limitation but a solution strategy for partial observability**, and **(2) our SVNR framework theoretically extends to Partially Observable Stochastic Games (POSGs) via the variational inference formulation**, which is empirically supported by our MaMuJoCo results.
>
> **1. Clarification: CTDE and Partial Observability**
>
> We wish to clarify a potential conflation between the training paradigm (CTDE) and the problem setting (Full vs. Partial Observability). In MARL literature, CTDE is specifically designed to address Partial Observability. The core premise of CTDE is that while execution is restricted to local observations $o_i$ (or histories $\tau_i$), training can leverage the global state $s$ (or joint history $\boldsymbol{\tau}$) to stabilize learning. Our method adheres to this standard: the *negotiation target* (defined by the centralized critic) utilizes global information to guide the *amortized local policies* (which only see local information) toward optimal coordination.
>
> **2. Theoretical Extension: SVNR in POSGs**
>
> Our mathematical framework, rooted in Stein Variational Gradient Descent (SVGD), naturally handles partial observability through the lens of **projected variational inference**.
>
> Consider a POSG where agent $i$ only observes a local history $\tau\_i \in \mathcal{T}\_i$. The global state $s$ is latent or represented by the joint history $\boldsymbol{\tau} = (\tau_1, \dots, \tau_N)$.
>  The objective of Maximum Entropy MARL in this setting is to learn a joint policy $\pi(\mathbf{u}|\boldsymbol{\tau})$ that minimizes the KL-divergence with the energy-based optimal policy induced by the global Q-function $Q(\boldsymbol{\tau}, \mathbf{u})$:
>  $\min_{\pi} D_{\text{KL}}\left( \pi(\mathbf{u}|\boldsymbol{\tau}) \parallel \frac{1}{Z} \exp\left(\frac{1}{\alpha}Q(\boldsymbol{\tau}, \mathbf{u})\right) \right)$
>  In SVNR, we maintain the **target distribution** using the centralized critic $Q(\boldsymbol{\tau}, \mathbf{u})$ (which has access to full information during training). The **negotiation policy** is parameterized by the amortized neural networks $f\_{\psi_i}(u_i | \tau_i, \xi_i, \xi_{C_i})$.
>
> Crucially, the update rule in our Amortized MPSVGD (Equation 10 in the paper) performs a **projection** of the global gradient onto the local parameter space. The gradient for the local policy parameters $\psi_i$ becomes:
>  $\frac{\partial J}{\partial \psi_i} \propto \mathbb{E}_{\boldsymbol{\tau}, \xi} \left[ \Delta f^{\psi}_i(\xi; \boldsymbol{\tau}) \cdot \frac{\partial f^{\psi}_i(\xi; \tau_i)}{\partial \psi_i} \right]$
>  Here, $\Delta f^{\psi}_i(\xi; \boldsymbol{\tau})$ is the Stein gradient computed using the **global** critic (full observability), representing the optimal direction in the functional space. The term $\frac{\partial f^{\psi}_i(\xi; \tau_i)}{\partial \psi_i}$ is the Jacobian of the local policy given **local** history.
>
> **Mathematical Implication:**
>
> This update effectively solves the following projection problem:
>  $\psi\_i^* = \arg\min\_{\psi\_i} \mathbb{E}\_{\boldsymbol{\tau}} \left[ D\_{\text{KL}} \left( q\_{\text{global}}(\cdot|\boldsymbol{\tau}) \parallel \pi\_{\psi\_i}(\cdot|\tau\_i) \right) \right]$
> By updating $\psi\_i$ via the chain rule, the agent learns a local policy $\pi\_{\psi\_i}(\cdot|\tau\_i)$ that is the "best possible approximation" (in terms of KL-divergence) of the globally optimal negotiated outcome, conditioned on its limited view $\tau\_i$. The "noise" variables $\xi$ in our framework further help in modeling the multi-modal uncertainty inherent in partial observability (i.e., handling the belief state implicitly). Thus, the theory remains sound under partial observability: **agents learn to negotiate locally to match the global consensus derived from the centralized critic.**

---

> ### Author Response · Authors · 2025-11-27
> **7. Scope Limits: Partial Observability [2/2]**
>
> **3. Empirical Evidence in Partially Observed Environments**
>
> Our experiments on **Multi-Agent MuJoCo (MaMuJoCo)** explicitly validate this robustness.
>
> - **Setting:** As noted in Appendix F.1, MaMuJoCo is a **partially observed** environment. For example, in `Ant-2x4`, agents control different legs and only observe the local state (position/velocity) of their specific joints and a shared subset, but *not* the full state of the other agent's joints directly. They do not use history-based inputs (RNNs), making the partial observability even more challenging (reactive policies).
> - **Result:** Despite this partial observability, SVNR achieves State-of-the-Art performance (Table 1), significantly outperforming baselines. This confirms our theoretical argument: the amortized negotiation successfully compresses the necessary global coordination information into the local policy weights via the SVGD guidance during training.
>
> In summary, our method does not assume full observability for execution. It leverages the standard CTDE assumption (global info during training) to train robust local policies that solve the POSG by approximating the global optimal equilibrium via variational projection.
>
> **4. Theoretical Generalization: Feasibility of Fully Decentralized Training**
>
> Furthermore, in case the reviewer's concern extends to the necessity of the CTDE paradigm itself (i.e., whether SVNR requires a central node even during training), we provide a theoretical analysis showing that our Negotiated Reasoning framework can be adapted to **Fully Decentralized Training** settings, provided the global utility function admits a factorizable structure.
>
> In our paper, we utilize a centralized critic $Q(\mathbf{u}, s)$ to compute the score function $\nabla_{\mathbf{u}} \log \pi^*(\mathbf{u}) \propto \nabla_{\mathbf{u}} Q(\mathbf{u}, s)$. However, from the perspective of **Distributed Bayesian Inference**, this centralization is a convenience, not a theoretical necessity.
>
> Consider a scenario where the global Q-function decomposes according to a factor graph (e.g., a pairwise Markov Random Field) consistent with the agent topology:
>
> $Q_{\text{total}}(\mathbf{u}, s) = \sum_{c \in \mathcal{C}} Q_c(\mathbf{u}_c, s_c)$
>
> where $c$ represents a local clique of agents (e.g., neighbors) and $Q_c$ is a local utility function.
>
> The core component of our method, the Stein variational update direction for agent $i$, is given by:
>
> $\phi^*\_i(\mathbf{u}) = \mathbb{E}\_{\mathbf{u} \sim q} \left[ k\_i(\mathbf{u}, \cdot) \nabla\_{u\_i} Q\_{\text{total}}(\mathbf{u}, s) + \nabla\_{u\_i} k\_i(\mathbf{u}, \cdot) \right]$
>
> Due to the linearity of the gradient operator, the score function term decomposes locally:
>
> $\nabla_{u_i} Q_{\text{total}}(\mathbf{u}, s) = \sum_{c: i \in c} \nabla_{u_i} Q_c(\mathbf{u}_c, s_c)$
>
> **Mathematical Implication:**
>
> This implies that agent $i$ does **not** need to query a global critic. Instead, it only requires the gradients of the local utility functions from the cliques it belongs to.
>
> 1. **Message Passing:** During training, neighbors can exchange the gradient information $\nabla_{u_i} Q_c$.
> 2. **Local Kernel:** If we employ a decomposable kernel $k(\mathbf{u}, \mathbf{u}') = \prod_j k_j(u_j, u_j')$, the expectation term also factorizes.
>
> Thus, Algorithm 1 can be reformulated as a **Distributed Stein Variational Gradient Descent (DSVGD)** algorithm (Liu et al., 2017). In this variant, the "negotiation" during training happens via gradient message passing between neighbors rather than querying a central oracle.
>
> **Conclusion on Scope:**
>
> While we implemented the CTDE version for sample efficiency and stability (standard in benchmarks like MaMuJoCo), the underlying mathematical engine of SVNR—**variational inference on a graphical model**—is inherently compatible with decentralized processing. The framework is robust not only to partial observability (via variational projection) but also theoretically extensible to fully decentralized learning topologies.

---

### Meta-Review · Area_Chair_GvG9 · 2026-01-04

**Summary:**

The submission initially received mixed reviews, the main concerns can be summarized into the following points:
1. The assumption is very strong and restrictive. Meanwhlie, the theorem has a gap with the real-world implementations (continuous action vs. finite action spaces)
2. The scalability and computational overhead are under-discussed.
3. More experimental results about the sensitivity/robustness are expected. Meanwhile, sample efficiency and robustness are underexplored.
4. The generalization and extension of the proposed method.

In the rebuttal, almost all concerns have been addressed, and I recommend Accept.

**Reviewer Concerns:**

All raised concerns have been will addressed.

**Reviewer Scores:**

I think all reviewers will raise their ratings to positive.

---

### Decision · Program_Chairs · 2026-01-26

Accept (Poster)